# Signal-induced enhancer activation requires Ku70 to read topoisomerase1–DNA covalent complexes

Yuliang Tan ●[1] ✉, Lu Yao ●[2], Amir Gamliel[1], Sreejith J. Nair[1,4], Havilah Taylor[1], Kenny Ohgi[1], Aneel K. Aggarwal ●[3] & Michael G. Rosenfeld ●[1] ✉

Enhancer activation serves as the main mechanism regulating signal-dependent transcriptional programs, ensuring cellular plasticity, yet central questions persist regarding their mechanism of activation. Here, by successfully mapping topoisomerase I–DNA covalent complexes genome-wide, we find that most, if not all, acutely activated enhancers, including those induced by 17β-estradiol, dihydrotestosterone, tumor necrosis factor alpha and neuronal depolarization, are hotspots for topoisomerase I–DNA covalent complexes, functioning as epigenomic signatures read by the classic DNA damage sensor protein, Ku70. Ku70 in turn nucleates a heterochromatin protein 1 gamma (HP1γ)–mediator subunit Med26 complex to facilitate acute, but not chronic, transcriptional activation programs. Together, our data uncover a broad, unappreciated transcriptional code, required for most, if not all, acute signal-dependent enhancer activation events in both mitotic and postmitotic cells.

Forty years removed from their initial discovery[1,2], gene control by transcriptional enhancers is considered one of the dominant molecular mechanisms underlying cell-type and signal-specific transcriptional diversity in metazoans. Thus, in addition to their critical functions in cell-type determination and differentiation, transcriptional enhancers serve as the predominant regulators of the precise, rapidly altered patterns of gene regulation in response to diverse acute signaling pathways[3–7], reflecting the preferential binding of many of these signal-dependent transcription factors to enhancers, rather than promoters. Interestingly, signal-dependent enhancer activation generally temporally precedes activation of its cognate promoter[8,9], and the robust, signal-regulated enhancer exhibit increased eRNA transcription and concomitant condensation of RNA-dependent ribonucleoprotein (RNP) complexes upon acute stimulation but maturing to a more gel-like state upon chronic stimulation[10–13]. However, whether there might be a universal strategy required for signal-dependent

enhancer activation is largely unknown. Intriguingly, recent reports have revealed that active enhancers serve as the hotspots for DNA single-strand breaks, or DNA nicks[14,15], which are the most common form of DNA damage. Unfortunately, the role of these putative DNA nicks in the signal-dependent transcriptional activation of the functional enhancers remains unclear.

In this article, we interrogate the potential role of the 'nicked' covalent intermediate that topoisomerase I (Top1) makes with genomic DNA (TOP1cc) in response to acute activating signals. Although Top1 has emerged as a critical component of the transcriptional machinery on promoters, few experiments have been directed at examining its role in robustly activated enhancers. This is because detection of the genomic landscape of the TOP1cc in mammalian cells has remained a main challenge. Firstly, TOP1cc is transient and not readily detectable[16]. Secondly, traditional chromatin immunoprecipitation (ChIP) assays induce very high artificial formation of poly ADP-ribose[17], which is involved

[1]Department of Medicine, University of California San Diego, La Jolla, CA, USA. [2]Familial and Hereditary Cancer Center, Key Laboratory of Carcinogenesis and Translational Research (Ministry of Education/Beijing), Peking University Cancer Hospital and Institute, Beijing, China. [3]Department of Pharmacological Sciences, Icahn School of Medicine at Mount Sinai, New York, NY, USA. [4]Present address: Department of Oncology Lombardi Comprehensive Cancer Center, Georgetown University, Washington DC, USA. ✉e-mail: yut020@health.ucsd.edu; mrosenfeld@ucsd.edu

in the formation and release of TOP1cc[16,18]. Third, current assays for DNA–protein intermediates such as an immunocomplex of enzyme and trapped in agarose DNA immunostaining assays are low throughput[19,20], while chaotropic salts isolation[21], fluorescein isothiocyanate-labeling[22], rapid approach to DNA adduct recovery[23] approaches and even the recently developed repair–seq[14], synthesis-associated with repair sequencing and S1-END–seq[15] have failed in identifying the DNA-bound sites of the specific proteins interrogated. Therefore, the function of TOP1cc in translating activation signals into context-specific responses remains poorly understood.

Here, by taking advantage of a specific antibody recognizing TOP1cc, in conjunction with CUT&RUN assays, we unexpectedly find that TOP1cc serves as an epigenomic signature recognized by the DNA damage sensor protein Ku70, which has been classically linked to the double-strand DNA damage repair pathway. We provide evidence that Ku70 is an invariant requirement for acute signal-dependent enhancer activation, acting to tether HP1γ–Med26 to facilitate the phosphorylation of serine 5 at RNA polymerase II (Pol II) to promote the transcriptional elongation of the enhancers. Remarkably, we find that not only 17β-estradiol (E$_2$) but also dihydrotestosterone (DHT), tumor necrosis factor alpha (TNFα) and even neuronal depolarization also require the actions of TOP1cc and the reading of the TOP1cc by Ku70 for acute enhancer activation. The discovery of TOP1cc as an epigenomic signature in a broad swath of acute signal-dependent enhancer activation events uncovers a conceptually new dimension in the regulation of enhancer function in mammalian cells.

## Results

### TOP1cc is detected at signal-dependent active enhancers

An ideal system to explore any potential link between the actions of Top1 at enhancers and signal-dependent enhancer activation is afforded by E$_2$-caused rapid assembly of a megadalton-sized enhancer complex (MegaTrans complex), on robustly activated estrogen receptor alpha (ERα)-bound MegaTrans enhancers, controlling E$_2$-regulated transcriptional programs[24–26]. To avoid formaldehyde fixation-induced high artificial formation of poly ADP-ribose[17], which is involved in the formation and release of TOP1cc[18], we employed CUT&RUN assays to uncover the genomic landscape of the TOP1cc by utilizing a monoclonal antibody with specificity for TOP1cc. We found 18,308 TOP1cc-enriched regions in E$_2$ (1 h)-treated human breast cancer MCF7 cells, of which 55.9% were localized to intronic and intergenic regions and around 31.6% were localized at promoters (Fig. 1a). Indeed, TOP1cc was associated strongly with active enhancers, as measured by H3K27Ac and H3K4me2, but not with condensed chromatin, as identified by H3K9me3 (Fig. 1b,c). To verify the authenticity of the TOP1cc detected in the genome, the requirement for enzymatically active Top1 was tested by employing CRISPR interference (CRISPRi; dCas9 fused with KRAB protein)[27] to suppress the endogenous Top1 gene promoter in MCF7 cells expressing tetracycline-inducible wild type (wt) or the catalytically dead mutant (Y723F)[16,28,29] Top1 (Fig. 1d). Our results showed that the appearance of TOP1cc was strikingly diminished in Top1-Y723F-expressing cells (Fig. 1e), confirming the authenticity of the TOP1cc loci detected in the genome.

This approach affords a dramatic improvement in detection of the specific TOP1cc actions compared with previous TOP1–seq approaches[30], which, for example, in human colon cancer HCT116 cells, detected only 508 peaks of Top1-dependent single-stranded DNA nicks in the genome. We also compared TOP1cc signals with traditional ChIP–seq data targeting Top1 in human prostate cancer LNCAP cells (Extended Data Fig. 1a) and found that, although 62.3% TOP1cc proved to locate with Top1-enriched regions (Extended Data Fig. 1b–d), TOP1cc could be detected at only around 14.1% of the Top1-enriched regions detected with ChIP–seq[29], which might reflect assay noise due to the formaldehyde fixation[17].

### Top1 is required for acute transcriptional activation

To explore the function of TOP1cc at active enhancers, we found that bromodomain-containing protein 4 (BRD4), a well-known transcriptional activator[31], was correlated strongly and positively with TOP1cc at the enhancers in E$_2$-treated MCF7 cells (Extended Data Fig. 2a,b). We thus employed hypergeometric optimization of motif enrichment (HOMER)[4] to investigate motifs enriched at enhancers marked with TOP1cc in E$_2$-treated MCF7 cells, finding estrogen response elements (ERE), Forkhead Box A1 (FOXA1) and Grainyhead-like protein 2 (GRHL2) motifs were highly represented (Fig. 2a), all of which are enriched in ERα transcriptionally activated enhancers[32,33]. Strikingly, ERα was detected in 68.5% of TOP1cc-enriched enhancers in E$_2$-treated MCF7 cells by ChIP–seq (Fig. 2b and Extended Data Fig. 2c), and TOP1cc signals were increased (Extended Data Fig. 2d) following the induction of ERα signals (Extended Data Fig. 2e) at TOP1cc-enriched, robustly active MegaTrans enhancers, indicating that the recruitment of ERα to these signal-dependent active enhancers might promote the formation of TOP1cc.

Further, we found that knockdown of Top1 (Extended Data Fig. 3a) impaired the E$_2$-dependent eRNA transcription at TOP1cc-enriched, robustly active MegaTrans enhancers (Extended Data Fig. 3b,c). In contrast, enhancers at which TOP1cc were not enriched were minimally impaired by Top1 knockdown (Extended Data Fig. 3d). Validating the importance of TOP1cc for signal-dependent enhancer activation, we found that the robustly E$_2$-activated Tff1 and Greb1 enhancers could be rescued in Top1 wt, but not by the catalytically dead Top1-Y723F-expressing cells (Fig. 2c).

We next asked whether TOP1cc is also required for the chronic activation of the ERα-marked active enhancers, characterized by similar ERα binding but less robust eRNA transcription[13]. We found that TOP1cc was, at best, minimally detected at the MegaTrans enhancers following chronic (around 14 h) E$_2$ stimulation (Fig. 2d and Extended Data Fig. 4a–d). In concert with these findings, the acute E$_2$-dependent activation of the cognate target genes of the MegaTrans enhancers was inhibited by Top1 knockdown (Fig. 2e), while the activation of these cognate target genes following chronic E$_2$ treatment was not impaired (Fig. 2f). Lastly, because of the role of eRNA in the assembly and physical properties of MegaTrans enhancer condensates[12,13], we found that the eRNA expression level at the TOP1cc-enriched, acutely activated MegaTrans enhancers was also much higher than that of non-TOP1cc-enriched MegaTrans enhancers (Extended Data Fig. 4e,f),

**Fig. 1 | TOP1cc is presented at signal-induced active enhancers. a**, Pie chart shows the genome-wide distribution of TOP1cc CUT&RUN signals in E$_2$-treated MCF7 cells. **b**, Numbers of TOP1cc-enriched regions at ATAC–seq peaks, ChIP–seq peaks of enhancer-related marks (H3K4me2 and H3K27Ac), additional chromatin marks at accessible regions (H3K4me3, H4K16Ac, H2A.Z, SMC1, RAD21 and CTCF), chromatin silencing marks (H4K20me1, H3K27me3 and H3K9me3) and 179,220 random selected genomic regions in E$_2$-treated MCF7 cells. **c**, Genomic browser images show the TOP1cc CUT&RUN signals, Top1 and H3K27Ac ChIP–seq signals at selected gene loci. IgG and H3K27me3 CUT&RUN signals are serving as the experimental controls for the CUT&RUN assays. Enhancers are highlighted with light-brown boxes. **d**, Western blots show the expression of BLRP-tagged wt and Y723F enzymatic dead mutant Top1 in MCF7 cells. Endogenous Top1 was inhibited with dCas9-KRAB, and wt and Y723F enzymatic dead mutant Top1 were expressed using DOX; three biological replicates. **e**, TOP1cc signals (n = 18,308) in E$_2$-treated BLRP-tagged wt- and enzymatic dead mutant Top1-Y723F-expressing MCF7 cells detected by CUT&RUN assays are shown with heatmaps. An additional 5 kb from the center of the peaks is shown, and the color scale shows the normalized tag numbers. Center lines show the medians, box limits indicate the 25th and 75th percentiles and whiskers extend 1.5× the interquartile range from the 25th and 75th percentiles. P value generated from unpaired two-tailed t-test denotes statistical differences between wt and Y723F conditions. Uncropped images for **d** and data for graphs in **b** are available as Source data.

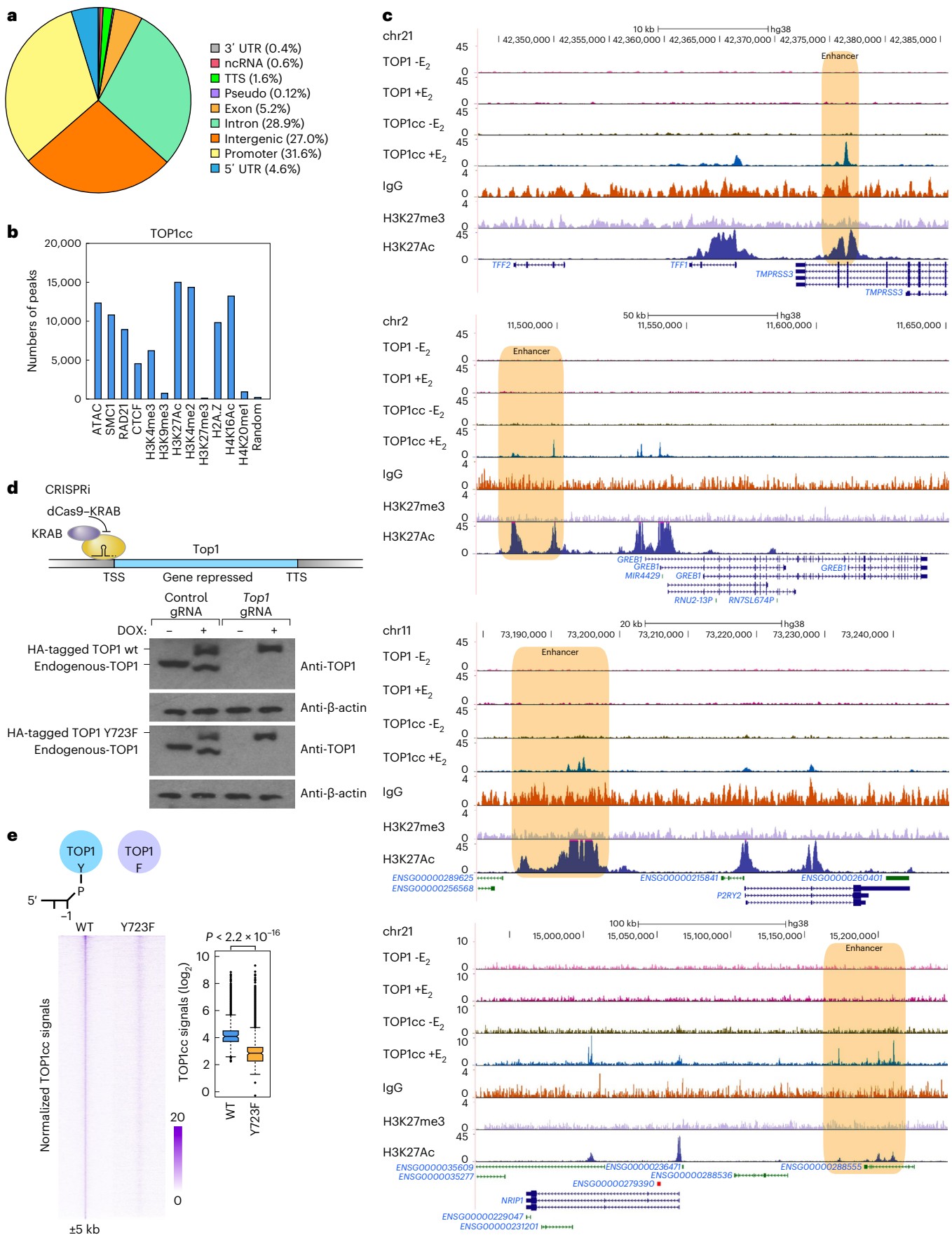

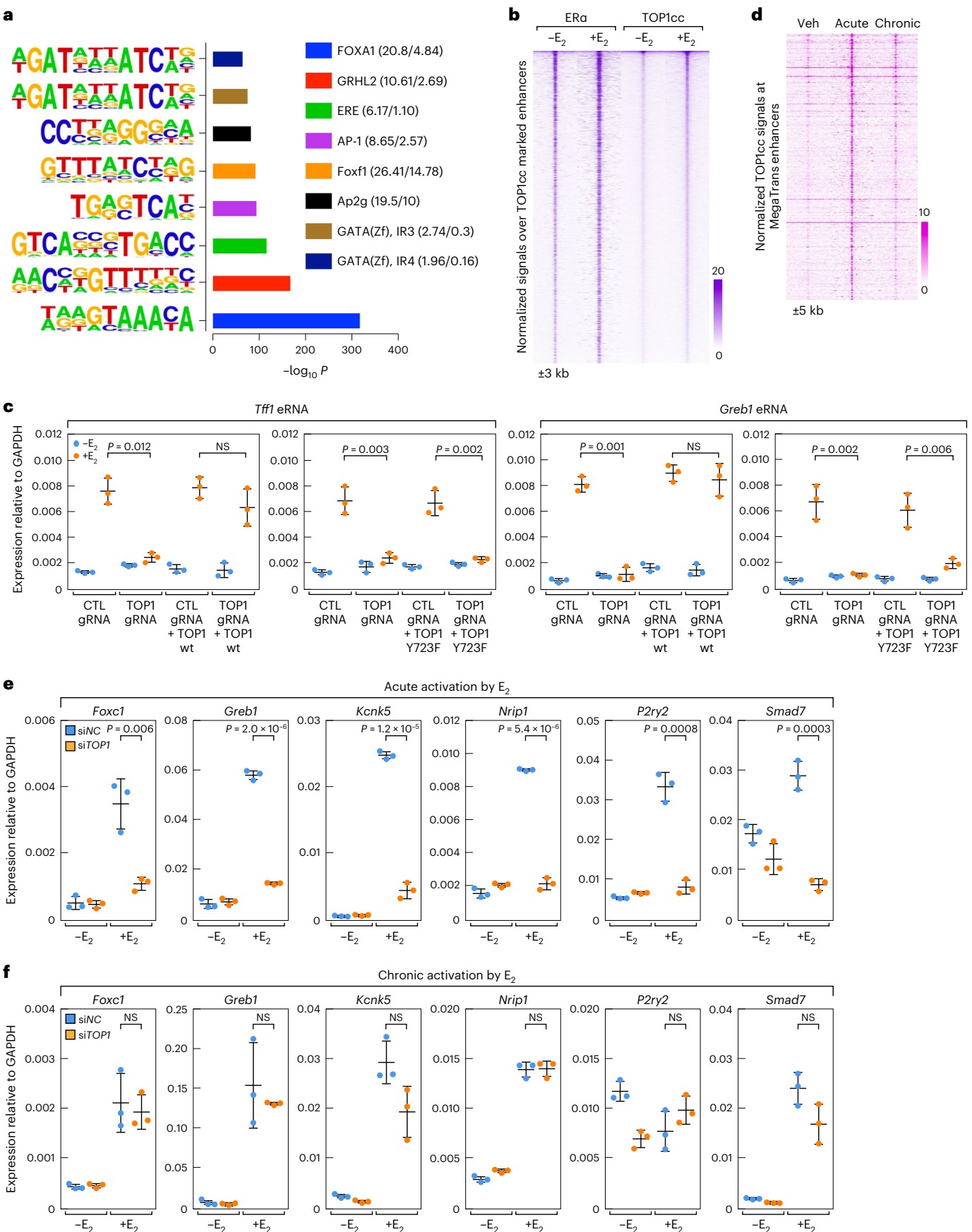

**Fig. 2 | TOP1cc is required for signal-dependent acutely enhancer activation.**
**a**, Motifs for sequences within 200 bp of the summit of the TOP1cc peaks at the enhancers in $E_2$-treated MCF7 cells are presented by bar graph. *P* values generated from two-tailed binomial test denote statistical differences between the target and background sequences for enrichment. **b**, ERα ChIP–seq signals and TOP1cc CUT&RUN signals and at the enhancers in MCF7 cells are shown with heatmaps. Additional 3 kb from the center of the peaks are shown. **c**, RT–qPCR results show that the transcriptional activation of *Tff1* and *Greb1* enhancers could be recused by DOX-induced expression of wt Top1, but not enzymatic dead mutant Top1Y723F. **d**, Heatmaps show TOP1cc CUT&RUN signals at MegaTrans

enhancers upon Veh (-$E_2$), acute (1 h) and chronic (around 14 h) $E_2$ treatment. An additional 5 kb from the center of the peaks is shown, and the color scale shows the normalized tag numbers. **e**, RT–PCR shows that Top1 is required for the acute activation of $E_2$-induced transcriptional programs. **f**, RT–qPCR results show that the chronical activation of MegaTrans enhancers cognate target genes is not Top1-dependent. For **c**, **e** and **f**, data are shown as mean ± s.d. (*n* = 3 (three independent biological replicates), two-tailed Student's *t*-test); NS, nonstatistically significant. Raw data for graphs in **c**, **e** and **f** are available as Source data.

consistent with the coordinate role of acutely activated eRNAs and ERα in promoting the transient formation of the TOP1cc at MegaTrans enhancers.

### Ku70 is tethered to TOP1cc-enriched active enhancers

To explore the mechanisms underlying TOP1cc-mediated transcriptional activation, proteins interacting with Top1 were identified by reversible crosslink immunoprecipitation (ReCLIP)[34] and mass spectrometry (MS). We found that the components of the nonhomologous end joining (NHEJ) pathway such as catalytic subunit of DNA protein kinase (DNA-PKcs), paralogues of XRCC4 and XLF (PAXX), Ku70 and Ku80 were the main interactants that increased in $E_2$-treated MCF7 cells (Fig. 3a and Supplementary Tables 1–3), and the reciprocal immunoprecipitation (IP) confirmed that these components interact with Top1 (Fig. 3b).

We then investigated the genomic landscape of the DNA-PKcs, Ku70 and Ku80 by ChIP–seq experiments. Both DNA-PKcs and Ku70 were induced at TOP1cc-enriched enhancers in response to $E_2$ treatment (Fig. 3c). Furthermore, the recruitment of Ku70 was correlated positively with TOP1cc at the TOP1cc-enriched MegaTrans enhancers (Fig. 3d). We next tested whether TOP1cc could function as the molecular signature that could be 'read' by Ku70, finding that both the Top1-dependent DNA nicking (Top1-DN) activities, which is abolished by the TOP1-Y723F mutant, and the presence of DNA were required for the interaction between Top1 and Ku70 by IP (Fig. 3e,f), suggesting that Ku70 is recruited to TOP1cc-enriched genomic regions resulting from Top1-induced single-strand DNA nicking at enhancers upon acute signal stimulation.

In contrast, Ku80, which is required for double-strand DNA damage repair by forming a heterodimer with Ku70 (ref. [35]), was not at all, or only minimally, increased following the induction of TOP1cc signals at these TOP1cc-enriched MegaTrans enhancers (Fig. 3g,h). Interestingly, in genomic regions that were cobound with Ku70 and Ku80, and without any transcriptional activation as indicated by diminished RNA Pol II induction upon $E_2$ stimulation, Ku70 was not induced; in contrast, in regions cobound with Ku70, Ku80 and exhibiting Pol II induction, Ku70 was induced upon acute $E_2$ stimulation (Fig. 3i). Finally, while the DNA damage sensor proteins including DNA-PKcs and Ku70 were required for the transcriptional activation of $E_2$-regulated enhancers, such as the *Greb1* and *Tff1* enhancers, knockdown of DNA damage

repair proteins, such as XRCC4, was not required (Fig. 3j and Extended Data Fig. 5), confirming the role of the classic DNA damage sensor protein Ku70, in TOP1cc-induced acute transcriptional regulation at signal-dependent active enhancers.

### HP1γ and Med26 are recruited to facilitate transcription

To explore how the classic DNA damage sensor protein[36], Ku70, can induce transcriptional activation at enhancers, and noting that HP1γ, one of the main factors pulled down by TOP1cc, has been reported to be tethered to euchromatin by Ku70 (ref. [37]), we tested the possibility that Ku70 functions to assemble the phosphorylated HP1γ on TOP1cc-enriched enhancers. Strikingly, our data indicated that HP1γ localization was highly correlated with the genomic landscape of TOP1cc and Ku70 (Fig. 4a). Moreover, HP1γ was induced at Ku70 and TOP1cc-enriched MegaTrans enhancers upon $E_2$ treatment, but not at other active enhancers (Fig. 4b). The interactions between HP1γ with Ku70 and Top1 were detected upon $E_2$ treatment by IP (Fig. 4c). Knockdown of *Ku70* greatly decreased the enrichment of HP1γ at the TOP1cc-enriched MegaTrans enhancers (Fig. 4d). Consistently, knockdown of *Top1* decreased the enrichment of HP1γ at those enhancers that exhibited TOP1cc and HP1γ colocalized/overlap by ChIP–seq experiments (Fig. 4e), as exemplified by the *Tff1* enhancer (Fig. 4f). In contrast, knockdown of *Top1* had little, or no, effect on the enrichment of HP1γ at the nonoverlapped regions (Fig. 4g).

Indeed, minimal HP1γ binding was observed in regions harboring heterochromatin-associated lysine 9 trimethylated histone H3 (H3K9me3), which is the main marker directing the localization of HP1 family members in the genome (Extended Data Fig. 6a,b). The correlation between H3K9me3 and HP1γ at heterochromatin is high (Extended Data Fig. 6c), but the genome-wide correlation between H3K9me3 and HP1γ is quite low (Extended Data Fig. 6d), while the correlation between TOP1cc and HP1γ is high (Extended Data Fig. 6e), indicating the role of TOP1cc in tethering HP1γ to the euchromatin regions. Then, to confirm whether phosphorylation of HP1γ[37] was important for the enrichment of HP1γ at enhancers, rescue experiments were performed. We found that mutation of Ser83 to alanine (S83A), which prevented the phosphorylation of HP1γ at Ser83, decreased HP1γ recruitment to enhancers. In contrast, mutation of Ser83 to the phosphomimic glutamine (S83D), resulted in increased HP1γ recruitment to the enhancers (Fig. 4h).

**Fig. 3 | Ku70 functions as the 'reader' Top1-DN at acutely activated enhancers in $E_2$-treated MCF7 cells. a**, ReCLIP assays show that the components of NHEJ complexes such as DNA-PKcs, Ku70, PAXX and HP1γ interact with TOP1 in MCF7 cells. **b**, Interactions between DNA-PKcs, Ku70, PAXX and TOP1. **c**, Genomic browsers show DNA-PKcs (GSE60270)[24], Ku70 and H3K27Ac ChIP–seq signals at selected gene loci. Enhancers are highlighted with light-brown boxes. **d**, Scatter plots showing the correlation between TOP1cc CUT&RUN signals and Ku70 ChIP–seq signals. Pearson correlation coefficient is indicated by *P* value. **e**, The interaction between Ku70 with TOP1 and TOP1Y723F are shown. **f**, The interaction between Ku70 with TOP1 are shown with coimmunoprecipitation. **g**, Heatmaps show Ku70 and Ku80 ChIP–seq signals at MegaTrans enhancers treatment. An additional 5 kb from the center of the peaks is shown, and the color scale shows the normalized tag numbers. **h**, Violin plots show normalized Ku70 and Ku80

ChIP–seq tags ($\log_2$) at TOP1cc-enriched MegaTrans enhancers (*n* = 481). **i**, Violin plots show normalized Ku70 and Ku80 ChIP–seq tags ($\log_2$) at Pol II unchanged genomic regions (*n* = 3,466) and $E_2$-dependent Pol II increased genomic regions (*n* = 3,907). **j**, RT–qPCR results show that the $E_2$-dependent transcriptional activation of *Greb1* and *Tff1* enhancer RNAs are impaired. Data are shown as mean ± s.d. (*n* = 3 (three independent biological replicates), two-tailed Student's *t*-test). For **h** and **i**, center lines show the medians, box limits indicate the 25th and 75th percentiles and whiskers extend 1.5× the interquartile range from the 25th and 75th percentiles. *P* values generated from unpaired two-tailed *t*-test denote statistical differences between −$E_2$ and +$E_2$ conditions, and the median value of normalized Pol II ChIP–seq tags ($\log_2$) are listed under the boxplots. Uncropped images for **b**, **e** and **f** and data for graphs in **j** are available as Source data.

Assessing the function of the HP1γ in mediating the transcriptional activation at enhancers, we found that, whereas Pol II exhibited only modest changes at other active enhancers upon the depletion of the *Hp1γ*, Pol II and other coactivators including BRD4, CBP and GATA3 were decreased dramatically at TOP1cc-enriched MegaTrans enhancers (Fig. 4i and Extended Data Fig. 7). In exploring why HP1γ, a protein

primarily associated with transcriptional silencing, was involved in the transcriptional activation at enhancers, we were cognizant of the fact that an alternative mediator subunit, Med26, harbors a canonical PXVXL motif at the extreme C-terminus that is sufficient to bind HP1γ[38]. Accordingly, we tested this possibility experimentally, finding that Med26 was highly recruited to the MegaTrans enhancers in response

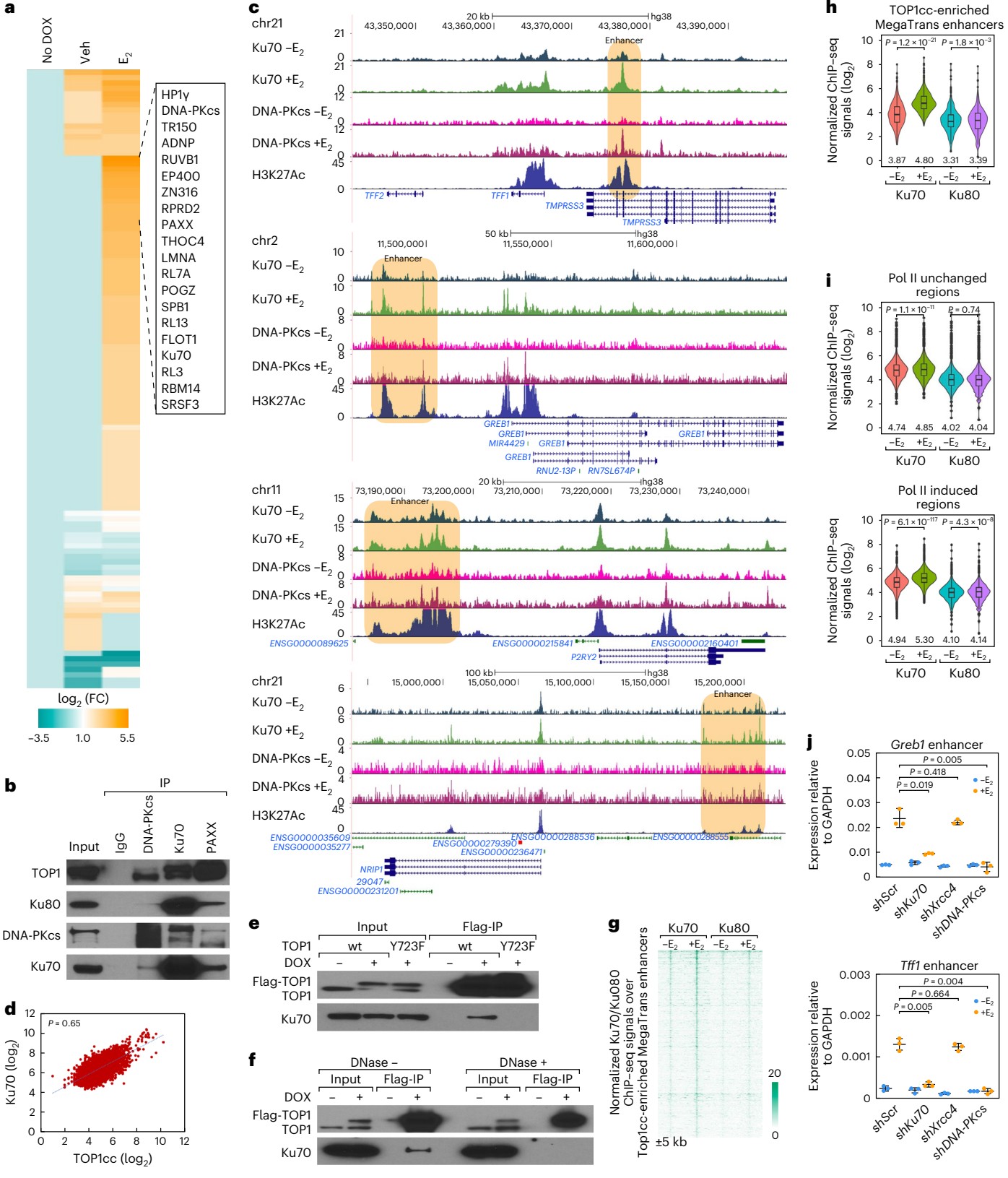

to E₂, and that recruitment of Med26 was decreased by knockdown of *Hp1γ* (Extended Data Fig. 8). Further, knockdown of *Med26* strikingly impaired the Pol II enrichment at TOP1cc-enriched MegaTrans enhancers (Extended Data Fig. 9a,b), while Pol II exhibited only modest changes at other active enhancers (Extended Data Fig. 9c). We further ascertained that the knockdown of *Med26* decreased Ser2 phosphorylated Pol II (PolIISer2P) dramatically at TOP1cc-enriched, acute activated MegaTrans enhancers (Extended Data Fig. 9d), consistent with reports that Med26 is required for transcriptional elongation by promoting the phosphorylation of Pol II Ser2[39,40].

### General requirement for TOP1cc–Ku70–HP1γ for transcription

Based on the discovery of TOP1cc–Ku70–HP1γ in E₂-dependent enhancer activation, we investigated whether this strategy might represent a common or even general mechanism underlying acute signal-induced enhancer activation. We tested whether TOP1cc–Ku70–HP1γ are employed in other acute signal-induced transcriptional activation events, consequent to the induction of TOP1cc at their corresponding enhancers. Strikingly, TOP1cc–Ku70–HP1γ were present at the acute TNFα-induced p65-bound active proinflammatory enhancers (Fig. 5a) and acute DHT-induced androgen receptor (AR)-bound active enhancers (Fig. 5b), wherein induction of the target genes mediated by the acute treatment of the TNFα or DHT was, in each case, diminished by the knockdown of the *Top1*, *Ku70* or *Hp1γ* (Fig. 5c).

To further generalize the importance of TOP1cc, we assessed enhancer activation during the depolarization of primary murine neuronal cultures, using a standard KCl-mediated depolarization protocol to mimic neuronal activity stimulation (Extended Data Fig. 10a)[41]. We identified 1,344 active enhancers in primary cortical neurons, of which 737 exhibited the induction of TOP1cc signals at KCl-activated enhancers (Fig. 5d). Specifically, the presence of corresponding binding motifs, transcription factors such as MEF2a, MEF2b, MEF2c and MEF2d, which are known to be crucial for neuronal activity-regulated gene transcription[42], highly represented at TOP1cc-enriched enhancers (Extended Data Fig. 10b), but not at non-TOP1cc enhancers (Extended Data Fig. 10c). TOP1cc at the enhancers adjacent to the neuronal activity-regulated genes, such as *Npas4* and *Fos*, was upregulated following KCl treatment in primary cortical neuronal cultures (Fig. 5e). Ku70–HP1γ was also induced at these TOP1cc-enriched neuronal enhancers upon KCl-mediated depolarization (Extended Data Fig. 10d).

## Discussion

### TOP1cc serves as a universal epigenomic signature

Because most DNA-binding factors activated in response to diverse signals bind primarily to cognate DNA sites in enhancers, rather than promoters, activation of enhancers serves as the main mechanism regulating acute signal-dependent modulation of transcriptional programs in virtually all cell types. Here, we address whether there is an as yet unappreciated molecular strategy that underlies most, if not all, signal-dependent enhancer activation events, despite the diversity of primary sequence, cell type and activating signal. Specifically, by employing a specific antibody recognizing Top1–DNA transient

intermediate with CUT&RUN assays to detect TOP1cc, we established a powerful technology that permitted us to provide evidence that TOP1cc is required to achieve robust enhancer activation following acute signals in all the systems that we have examined. We provide evidence that TOP1cc and the subsequent downstream events are required for diverse types of signal-induced enhancer activation, including not only estrogen, androgen and TNFα-activated enhancers (regulating proinflammatory genes), but also for neuronal depolarization-induced enhancer activation.

The role of Top1 in mediating acute, but not chronic, signal-dependent enhancer activation correlated with the distinct physiochemical properties of condensates established on acute versus chronically stimulated enhancers. That is, whereas the E₂-dependent assembly of RNP condensates on acute stimulated enhancers displays properties of liquid–liquid phase separation, chronically activated enhancers show progressive maturation of RNPs to a distinct, perhaps more gel-like, state[13]. Importantly, from our studies, eRNA appears to be capable not only of promoting liquid–liquid phase separation but also of augmenting the formation of TOP1cc. However, these two effects are probably interrelated, consistent with the ability of condensates to increase enzymatic activity[11].

Notably, the assay we describe here, utilizing a monoclonal antibody for Top1 covalently bound to DNA, distinguishes Top1-mediated nicks from single-strand DNA nicks associated with long patch DNA damage repair on neuronal specific enhancers[15], which are also enriched at active enhancers, but are not linked with transcriptional activation. Collectively, we uncover here an unappreciated mechanism underlying a broad range of signal-dependent enhancer activation events, wherein signal-dependent binding of regulated transcription factors to responsive enhancers elicits TOP1cc as an epigenomic signature for subsequent transcriptional activation.

### Ku70 functions as the 'reader' of the Top1-DN

Remarkably, the acute signal-induced TOP1cc licenses the recruitment of Ku70, which functions as a 'reader' of TOP1cc. It has long been assumed that the key to Top1 actions is to relieve torsional stress arising from robust transcription at coding gene bodies and promoters. However, we note that dynamic supercoiled DNA has been undetected at the regulatory enhancers[43], and the stabilized TOP1cc generated by Top1 inhibitors, which is well known to inhibit the relief of torsional stress[44], could induce the transcription of nascent RNA[45]. We find here that TOP1cc can be co-opted as an epigenetic signature for the recruitment of Ku70, hinting at a noncanonical role for TOP1cc in the mobilization of transcription factors rather than relief of torsional stress per se.

Indeed, in the traditional model for NHEJ, the initial recognition and binding of Ku70–Ku80 heterodimer to the double-strand breaks is followed by the recruitment of DNA-PKcs and formation of the DNA-PK holoenzyme, resulting in the formation of a DNA-PK dimer mediated by the conserved C-terminal helix of Ku80. This DNA-PK dimer then acts as a platform for binding of other protein and brings the broken DNA ends close together, allowing for completion of the process of double-strand break repair through NHEJ[46,47]. Here, we find that

---

**Fig. 4 | HP1γ is recruited to signal-dependent acutely activated enhancers by Ku70. a**, Venn diagram illustrating the overlap between TOP1cc CUT&RUN signals, Ku70 and HP1γ ChIP–seq signals. **b**, Box-and-whisker plots show HP1γ tags (log₂) at TOP1cc-enriched MegaTrans enhancers (*n* = 481) and other active enhancers (*n* = 3,533). *P* values denote statistical differences between −E₂ and +E₂ conditions. **c**, The interactions between Ku70 and HP1γ in E₂-treated MCF7 cells are shown. **d**, Histogram plots show the HP1γ signals on TOP1cc-enriched MegaTrans enhancers. An additional 5 kb from the center of peaks are shown. **e**, Box-and-whisker plots show HP1γ tags (log₂) at TOP1cc and HP1γ overlapped regions (*n* = 12,351). **f**, Genomic browser images show TOP1cc CUT&RUN signals, HP1γ and H3K27Ac ChIP–seq signals at selected gene loci. Enhancers are highlighted with light-brown boxes. **g**, Box-and-whisker plots show normalized HP1γ tags (log₂) at non-TOP1cc-enriched HP1γ enriched regions (*n* = 18,960). For **e**

and **g**, *P* values denote statistical differences between si*NC* and si*Top1* conditions. **h**, Heatmaps show the HP1γ signals at TOP1cc-enriched MegaTrans enhancers in E₂-treated wt, HP1γS83A and HP1γS83D mutant MCF7 cells. An additional 3 kb from the center of the peaks is shown. **i**, Box-and-whisker plots and heatmaps show Pol II ChIP–seq signals at other active enhancers (*n* = 3,533) and TOP1cc-enriched MegaTrans enhancers (*n* = 481). *P* values denote statistical differences between si*NC* and si*Hp1γ*. The median values of normalized Pol II ChIP–seq tags (log₂) are listed under the boxplots. An additional 3 kb from the center of peaks is shown, and the color scale shows the normalized tag numbers. For **b**, **e**, **g** and **i**, center lines show the medians, box limits indicate the 25th and 75th percentiles, whiskers extend 1.5× the interquartile range from the 25th and 75th percentiles and *P* values were generated from unpaired two-tailed *t*-test. Uncropped images for **c** are available as Source data.

Top1-induced single-stand DNA nicks do not require the presence of Ku80, and that Ku80 binding is not particularly induced at the enhancers upon acute signal-dependent stimulation, which is consistent with previous reports showing that Ku80 is not presented at Ku70–HP1γ complexes[37]. Thus, in addition to its evolutionarily important functions in double-strand DNA damage repair events as a heterodimer with Ku80, we propose that Ku70 has acquired an independent function

as a transcriptional coactivator in signal-induced enhancer activation, serving as a 'reader' of TOP1cc.

## Ku70 facilitates the recruitment of HP1γ–Med26

HP1γ is a member of the heterochromatin protein 1 family that reads H3K9 methylation via a conserved chromodomain, but can associate with the active gene promoters[48], and interacts with Med26 through

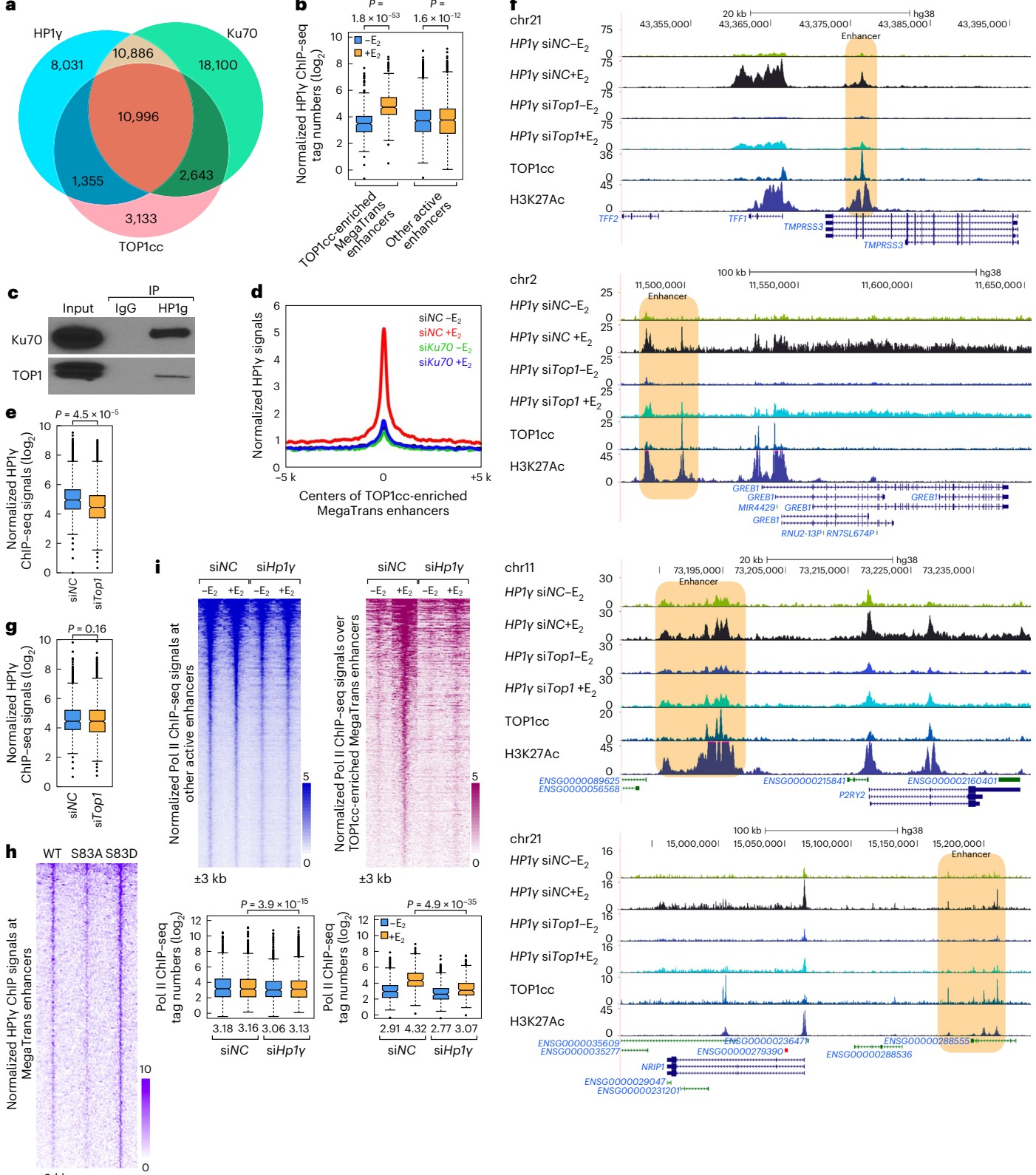

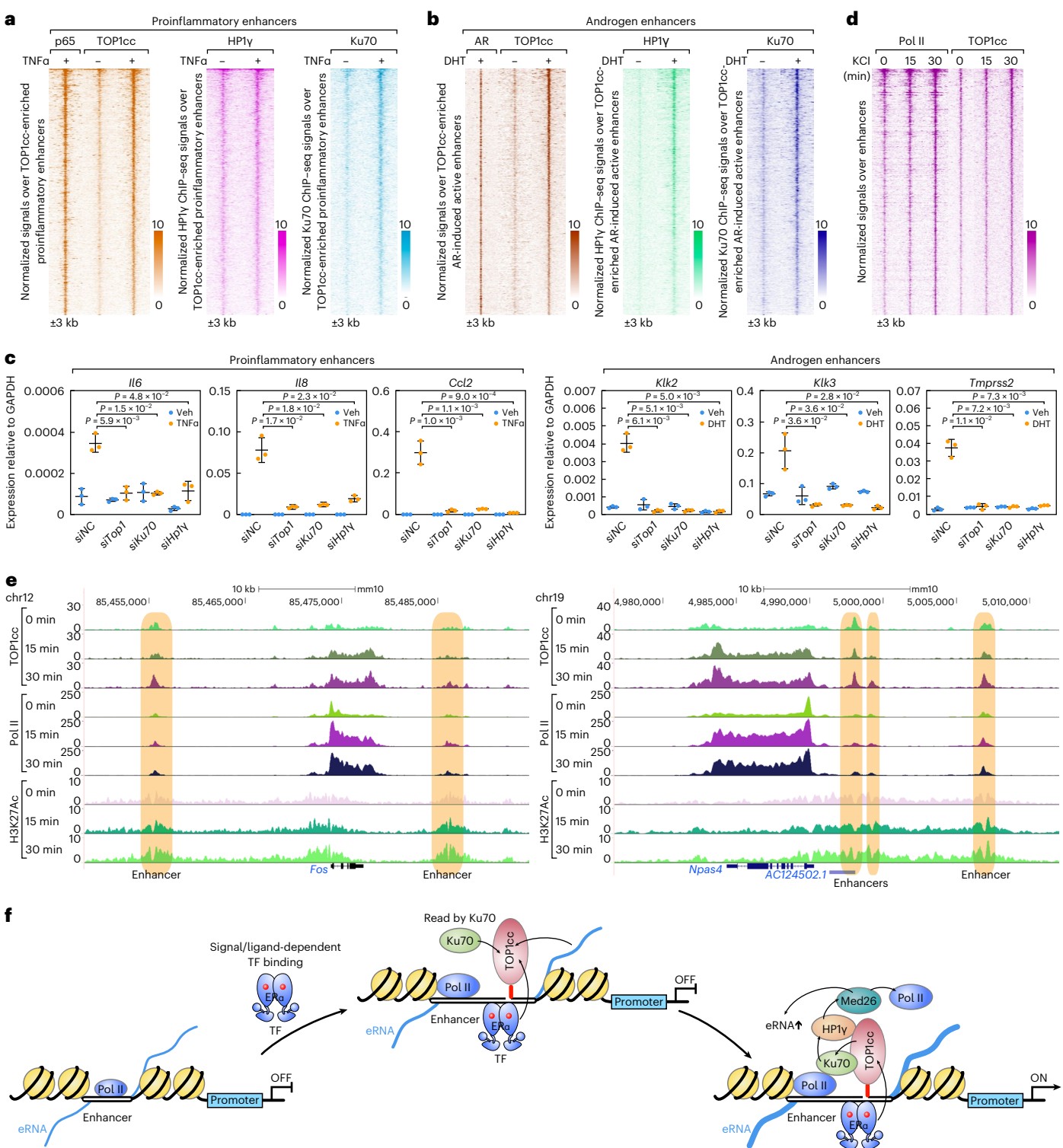

**Fig. 5 | TOP1cc–Ku70–HP1γ serves as a general transcriptional code underlying acute transcriptional activation. a**, Heatmaps show TOP1cc CUT&RUN signals and ChIP–seq signals (p65, HP1γ and Ku70) at TOP1cc-enriched acute TNFα activated proinflammatory enhancers (n = 589). **b**, Heatmaps show TOP1cc CUT&RUN signals and ChIP–seq signals (AR, HP1γ and Ku70) at TOP1cc-enriched acute DHT activated androgen enhancers (n = 691). An additional 3 kb from the center of the peaks are shown. **c**, RT–qPCR results show that the TNFα induced *Il6*, *Il8* and *Ccl2* genes, and DHT-induced *Klk2*, *Klk3* and *Tmprss2* genes are impaired by knockdown of *Top1*, *Ku70* or *Hp1γ*. Data are shown as mean ± s.d. (n = 3 (three independent biological replicates), two-tailed Student's t-test). **d**, Heatmaps show TOP1cc CUT&RUN signals and Pol II ChIP–seq signals at

TOP1cc-enriched KCl-induced active neuronal enhancers (n = 731). **e**, Genome browser images show Top1 CUT&RUN signals and ChIP–seq signals (Pol II and H3K27Ac) at the selected neuronal gene loci upon acute KCl treatment. Enhancers are highlighted with light-brown boxes. **f**, Working model: the signal-dependent acutely activated enhancers requires the signal-dependent transcriptional factor and enhancer RNA-dependent formation of TOP1cc, functioning as a broadly required transcriptional code read by Ku70, enabling the recruitment of HP1γ–Med26 for signal-dependent transcriptional activation. For **a**, **b** and **d**, an additional 3 kb from the center of the peaks are shown in the heatmaps, and the color scale shows the normalized tag numbers. Raw data for graphs in **c** are available as Source data.

its canonical PXVXL interaction motif[38,49]. Our data suggest that the acute augmentation of HP1γ–Med26 at the functional signal-dependent regulatory enhancers is a critical determinant of transcriptional activation in mammalian cells.

Mediator has been demonstrated to be required for the robust transcriptional activation[50,51], and our data strongly suggest that Med26 at acutely activated enhancers facilitates the transition of Pol II from initiation condensates with unphosphorylated Ser2 to RNA processing or elongation condensates with phosphorylated Ser2. The functional importance of Med26 proposed in this article is consistent with a recent report showing that the knock-out of *Med26* affects a larger gene expression program than knock-out of any other mediator subunit evaluated, including Med1, in mammalian cells[52].

Taken together, we conclude that TOP1cc serves in effect as an epigenomic signature recognized by Ku70, leading to the nucleation of a HP1γ–Med26 complex required for robust transcriptional activation at signal-dependent acute activated enhancers (Fig. 5f). Strikingly, TOP1cc-dependent Ku70–HP1γ recruitment emerges from our studies as a general requirement for acute signal-dependent enhancer activation events in mammalian cells and is consistent with the model that the interaction of HP1γ with Med26 is probably to augment its functions to activate the elongation complex and increase eRNA transcription and transcriptional factor condensation at acute activated enhancers.

## Online content

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

## Methods

### Cell culture

MCF7 cells (ATCC, HTB-22) were maintained at 37 °C and 5% $CO_2$ in DMEM (Gibco/Invitrogen), supplemented with 10% fetal bovine serum (FBS, Gibco/Invitrogen). LNCAP cells (ATCC, CRL1740) were maintained at 37 °C and 5% $CO_2$ in Advanced RPMI 1640 medium (Gibco/Invitrogen), supplemented with 10% FBS (GIBCO/Invitrogen). Mycoplasma contamination test was performed every 3 months for MCF7, LNCAP and 293T (ATCC, CRL3216) cells used in our laboratory. Cortical neurons were prepared as previously described[53], E15.5 C57BL/6 embryonic mouse cortices were dissected and then dissociated for 10 min in 1× Hank's balanced salt solution containing 20 mg ml$^{-1}$ trypsin (Life Technologies). Trypsin treatment was terminated by washing dissociated cells three times for 2 min each in dissociation medium consisting of 1× Hank's balanced salt solution containing a few drops of FBS (Life Technologies). Cells were then triturated using a flame-narrowed Pasteur pipette to fully dissociate cells. After dissociation, neurons in were kept on ice in dissociation medium until plating. Cell culture plates were precoated with a solution containing 20 µg ml$^{-1}$ poly-D-lysine (Sigma) in water. Before plating neurons, cell culture plates were washed three times with sterile distilled water. Neurons were grown in neuronal medium consisting of Neurobasal medium containing B27 supplement (2%; Thermo Fisher), penicillin-streptomycin (50 g ml$^{-1}$) and Glutamax (1×; Thermo Fisher). Neurons were subsequently plated and placed in a cell culture incubator that maintained a temperature of 37 °C and a $CO_2$ concentration of 5%; 10 min after plating neurons, the medium was aspirated completely from cells and replaced with fresh warm neuronal medium. Neurons were grown in vitro until day 7.

### Drug treatment

For hormone treatments, cells were incubated at 37 °C and 5% $CO_2$ for at least 3 days in phenol red-free DMEM (GIBCO/Invitrogen) supplemented with 5% charcoal dextran-stripped FBS (GIBCO/Invitrogen). For MCF7 cells, 17-β-estradiol (Steraloids, Inc.) was added to a final concentration of 100 nM. TNFα (R&D systems) was added to a final concentration of 20 ng ml$^{-1}$. For LNCAP cells, 5α-DHT (Sigma) was added to a final concentration of 100 nM. The ethanol vehicle control was 0.05% in all samples.

For the acute activation, cells were treated with drugs (ethanol or $E_2$ or TNFα or DHT) for 1 h for ChIP–seq or PRO-seq assays. To detect the changes of messenger RNA (mRNA) expression levels for the acute activation, cells were treated with drugs (ethanol or $E_2$ or TNFα or DHT) for 4 h and harvested for RNA isolation and subsequent RT–qPCR experiments. For the chronic activation, cells were treated with drugs (ethanol or $E_2$) for 14 h and harvested for RNA isolation and subsequent RT–qPCR experiments.

For ChIP–seq experiments, mouse cortical neurons were plated at an approximate density of $2 \times 10^6$ on 35-mm dishes. Neurons were plated in 2 ml neuronal medium. A 1 ml aliquot of the medium was replaced with 1 ml fresh warm medium on days 3 and 6. Before KCl depolarization, neurons were silenced with 1 µM tetrodotoxin (TTX; Fisher) on day 6. Neurons were subsequently stimulated on day 7 by adding warmed KCl depolarization buffer (170 mM KCl, 2 mM $CaCl_2$, 1 mM $MgCl_2$ and 10 mM 4-(2-hydroxyethyl)-1-piperazineethanesulfonic acid (HEPES)) directly to the neuronal culture to a final concentration of 31% in the neuronal culture medium within the culture plate or well for the indicated time.

### Transfection of small interfering RNA (siRNA)

For knockdown of *Top1* genes, the si*Top1_1* (SASI_Hs01_00047440, Sigma, CACAAAGAGAGGAAUGCUA[dT][dT], UAGCAAUCCUCUCUUUGUG[dT][dT]) and si*Top1_3* (SASI_Hs02_00335354, Sigma, GACAAGAUCCGGAACCAGU[dT][dT], ACUGGUUCCGGAUCUUGUC[dT][dT]) were employed. For knockdown of *Hp1γ* genes, the si*Hp1γ_1* (SASI_Hs01_00170890, Sigma, CCAAGAGGAUUUGCCAGAG[dT]

[dT], CUCUGGCAAAAUCCUCUUGG[dT][dT]) and si*Hp1γ_2* (SASI_Hs01_00196532, Sigma, CCAAGAGGAUUUGCCAGAG[dT][dT], CUCUGGCAAAAUCCUCUUGG[dT][dT]) were employed. For ChIP–seq and PRO-seq experiments, cells were transfected in 10-cm plates in regular DMEM without antibiotics. Then, 20 µl of 20 µM siRNAs and Lipofectamine 2000 reagent (Invitrogen catalog no. 11668-019) was diluted in Opti-MEM I Reduced Serum Medium (Invitrogen catalog no. 11058-021), and incubated for 6 h, and then changed to phenol red-free medium. For quantitative PCR (qPCR) or western blots, cells were transfected in six-well plates in regular DMEM without antibiotics and 4 µl of 20 µM siRNAs were transfected with Lipofectamine 2000 reagent (Invitrogen catalog no. 11668-019). In both conditions, cells were transfected 2 days later with siRNA again in phenol red-free medium. At 3 days following transfection, cells were treated with ethanol or $E_2$ for 4 h and harvested for RNA isolation for acute activation, treated with ethanol or $E_2$ for 14 h and harvested for RNA isolation for chronic activation or treated with ethanol or $E_2$ for 1 h for PRO-seq assay.

### Short hairpin RNA (shRNA) lentivirus package and infection

pLKO lentiviral shRNA constructs and control shRNA constructs were purchased from Addgene. The sequences of the shRNAs used in this article are shown in Supplementary Table 4. Knockdown experiments with lentivirus shRNAs were conducted according to the standard lentivirus package and transduction protocols from Addgene. pLKO-based lentiviral shRNA plasmids were cotransfected with packaging plasmids (psPAX2 and pMD2.G) into 293 T cells. Lentiviruses were harvested, concentrated and used for MCF7 cell infection. Stable knockdown MCF7 cells were selected with 1 µg ml$^{-1}$ puromycin and collected for experiments within 5 days. Before collection, the cells were grown for 3 days in phenol red-free DMEM (GIBCO/Invitrogen) supplemented with 5% charcoal dextran-stripped FBS (GIBCO/Invitrogen) and 0.5 µg ml$^{-1}$ puromycin for continued selection to achieve better knockdown.

### RNA isolation and RT–qPCR

RNA was isolated using Quick-RNA Miniprep Kit (Zymo Research, catalog no. R1054), and genomic DNA was removed from 1 µg of total RNA with DNA-free DNA removal kit (Invitrogen, catalog no. AM1609) and then total RNA reversed transcribed using the SuperScript III first-strand synthesis system with random hexamers (Invitrogen, catalog no. 18080-051). qPCR was performed with StepOne Plus (Applied Biosystems). Primer sequences used for the different gene targets are shown in Supplementary Table 4. All primers were checked on a standard curve, and it was verified that efficiencies were near 100%. β-Actin mRNA or glyceraldehyde-3-phosphate dehydrogenase (GAPDH) was used as the internal control. Relative mRNA levels were calculated by the ΔΔCt method with the vehicle (ethanol) used as the calibrator. The experiments were repeated at least three times, and one representative plot was shown in all figures; most $P$ values were obtained using a two-tailed Student's $t$-test.

### MCF7 Tet-On stable cell line construction

For BLRP-2XHA-Top1 wt or Y723F mutant Top1 cell line, MCF7 Tet-On Advanced cell line (catalog no. 631153) was bought from Takara Clontech. pRev-TRE-Flag-2XHA-Top1 or pRev-TRE-Flag-2XHA-Top1-Y723F was transfected into MCF7 Tet-On cells with the Tet-free serum supplied. At 2 days after transfections, cells were selected with hygromycin for 2 weeks. Colonies were picked and verified by western blot after doxycycline (DOX) induction. After titration, 5 µg ml$^{-1}$ DOX was chosen to achieve similar level of expression of HA-tagged Top1 compared with endogenous Top1.

### CRISPRi mediated *Top1* depletion

A dCas9-KRAB stable MCF7 cell line was constructed by lentivirus infection. The dCas9-KRAB lentivirus was prepared by cotransfecting pMD2.G, psPAX2 and Lenti-dCas9-KRAB-blast vector (Addgene,

Plasmid 89567) into HEK-293T cells using lipofectamine 2000 and harvesting lentivirus 48 h and 72 h after transfection. Then, MCF7 cells were infected with dCas9-KRAB lentivirus; 2 days after infection, blasticidin (working concentration 3 µg ml) was used to select positive infected cells. Then, the Alt-R CRISPR–Cas9 sgRNAs CD.Cas9.LPSZ5063. BO (mC*mG*mA* rGrArC rUrCrC rArGrA rArArC rGrGrC rUrGrG rUrUrU rUrArG rArGrC rUrArG rArArA rUrArG rCrArA rGrUrU rArArA rArUrA rArGrG rCrUrA rGrUrC rCrGrU rUrArU rCrArA rCrUrU rGrArA rArArA rGrUrG rGrCrA rCrCrG rArGrU rCrGrG rUrUrC mU*mU*mU* rU) and CD.Cas9.LPSZ5063.BN (mA*mG*mG* rCrUrG rUrUrA rCrArC rArArC rUrGrC rUrGrG rUrUrU rUrArG rArGrC rUrArG rArArA rUrArG rCrArA rGrUrU rArArA rArUrA rArGrG rCrUrA rGrUrC rCrGrU rUrArU rCrArA rCrUrU rGrArA rArArA rGrUrG rGrCrA rCrCrG rArGrU rCrGrG rUrGrC mU*mU*mU* rU) targeting the endogenous *Top1* gene promoter and 5′ terminal untranslated regions (5′ UTR) were transfected into the cells with Lipofectamine RNAiMAX transfection reagent (Thermo Fisher Scientific, catalog no. 13778100), and then cells were cultured using phenol red-free DMEM (GIBCO/Invitrogen) supplemented with 5% charcoal dextran-stripped FBS (GIBCO/Invitrogen). After 3 days, the cells were harvested, and RNA was extracted to analyze the expression of eRNAs and coding genes. Alt-R CRISPR–Cas9 control sgRNAs were made with 1:1 ratio of Alt-R CRISPR–Cas9 control crRNAs and Alt-R CRISPR–Cas9 control tarcrRNAs from Integrated DNA Technologies.

## Immunoblotting and coimmunoprecipitions

Cells were washed with ice-cold PBS and harvested in cold PBS. For the preparation of whole cell extracts, pellets were resuspended in lysis buffer containing 20 mM Tris-Cl (pH 8.0), 137 mM NaCl, 10% glycerol, 1% Nonidet P-40 and a mixture of protease inhibitors. Samples were sonicated by using a Bioruptor (Diagenode) for 16 min at medium power, with an interval of 30 s between pulses. Following sonication, samples were centrifuged for 2 × 5 min at 21,000$g$. After preclearing, IP was performed overnight at 4 °C by using the indicated antibodies and protein G-Sepharose. IP was followed by five washes in 1 ml lysis buffer performed at 4 °C, and protein complexes were denatured in Laemmli sample buffer (2% SDS, 10% glycerol, 60 mM Tris-Cl (pH 6.8), 0.01% bromophenol blue, 100 mM dithiothreitol (DTT)) for 5 min at 95 °C and resolved by NuPAGE Novex 4–12% Bis-Tris Protein Gels (Invitrogen catalog no. NP0336PK2). After electronic transfer, the PVDF membrane was blocked by incubation at room temperature for 1 h in Blocker Casein in TBS (Thermo Scientific, catalog no. 37532). Complexes were revealed by Clarity Western ECL substrate (Bio-Rad catalog no. 170-5061), as recommended by the manufacturer.

## ReCLIP and MS

ReCLIP was performed using dithiobis(succinimidyl propionate) (DSP, Thermo Scientific) following previous reports with minor changes[34,54]. Briefly, 4 × 10⁹ MCF7 cells were incubated in PBS containing 0.6 mM DSP for 30 min at 4 °C with mild rotation. After removing PBS, the remaining DSP was quenched by incubating the cells in TBS (50 mM Tris-Cl pH 7.4, 150 mM NaCl, 1 mM EDTA) for 15 min on ice. The cell nuclei were isolated through hypotonic lysis, and then lysed in lysis buffer (50 mM Tris-Cl pH 7.5, 150 mM NaCl, 0.25% Triton X-100, 0.25% Na-deoxycholate, 0.05% SDS, 1 mM EDTA, 5% glycerol) containing 5 mM MgCl₂, 300 mM NaCl and protease inhibitor cocktail (Roche). The nuclear extracts were then sonicated for 10 min, followed by rotation for 1 h at 4 °C and centrifugation at 20,000$g$ for 10 min. For immunopurification of the TOP1 protein complexes, nuclear extracts were incubated overnight with 100 µl streptavidin beads (Thermo Fisher Scientific) at 4 °C. After binding of the protein complexes, the streptavidin beads were washed sequentially with TBS, lysis buffer, high-salt lysis buffer (500 mM NaCl) and TBS. Finally, the beads were resuspended in 100 ml 2× SDS-sample buffer containing β-mercaptoethanol and heated for 10 min at 95 °C to elute the bound proteins. Protein complexes pulled down with streptavidin beads from precleared cell nuclei were employed for MS

analysis at the University of California San Diego (UCSD) molecular mass spectrometry facility. Briefly, protein samples were diluted in TNE (50 mM Tris pH 8.0, 100 mM NaCl, 1 mM EDTA) buffer. RapiGest SF reagent (Waters Corporation) was added to the mix to a final concentration of 0.1% and samples were boiled for 5 min. Tris (2-carboxyethyl) phosphine (TCEP) was added to 1 mM (final concentration) and the samples were incubated at 37 °C for 30 min. Subsequently, the samples were carboxymethylated with 0.5 mg ml⁻¹ of iodoacetamide for 30 min at 37 °C followed by neutralization with 2 mM TCEP (final concentration). Proteins samples prepared as above were digested with trypsin (trypsin:protein ratio, 1:50) overnight at 37 °C. RapiGest was degraded and removed by treating the samples with 250 mM HCl at 37 °C for 1 h followed by centrifugation at 21,130$g$ for 30 min at 4 °C. The soluble fraction was then added to a new tube and the peptides were extracted and desalted using C18 desalting columns (Thermo Scientific, catalog no. PI-87782). Peptides were quantified using BCA assay and a total of 1 µg of peptides was injected for LC–MS analysis.

## Ultra-high-pressure liquid chromatography coupled with tandem MS

Trypsin-digested peptides were analyzed by ultra-high-pressure liquid chromatography coupled with tandem MS (LC–MS/MS) using nanospray ionization. The nanospray ionization experiments were performed using an Orbitrap Fusion Lumos hybrid mass spectrometer (Thermo) interfaced with nanoscale reversed-phase UPLC (Thermo Dionex UltiMate 3000 RSLCnano System) using a 25 cm, 75-micron ID glass capillary packed with 1.7-µm C18 (130) BEHTM beads (Waters Corporation). Peptides were eluted from the C18 column into the mass spectrometer using a linear gradient (5–80%) of acetonitrile (ACN) at a flow rate of 375 µl min⁻¹ for 1.5 h. The buffers used to create the ACN gradient were Buffer A (98% H₂O, 2% ACN, 0.1% formic acid) and Buffer B (100% ACN, 0.1% formic acid). MS parameters were as follows: an MS1 survey scan using the orbitrap detector (mass range ($m/z$): 400–1,500 (using quadrupole isolation), 120,000 resolution setting, spray voltage of 2,200 V, ion transfer tube temperature of 275 °C, AGC target of 400,000 and maximum injection time of 50 ms) was followed by data-dependent scans (top speed for most intense ions, with charge state set to include only +2–5 ions, and 5 s exclusion time, while selecting ions with minimal intensities of 50,000 in which the collision event was carried out in the high-energy collision cell (higher energy collision dissociation energy of 30%), and the fragment masses were analyzed in an ion trap mass analyzer (with ion trap scan rate of turbo, first mass $m/z$ was 100, AGC target of 5,000 and maximum injection time of 35 ms). Protein identification was carried out using Peaks Studio v.8.5 (Bioinformatics Solutions Inc.). The proteins with the following standards were selected: peptide −log₁₀($P$) ($P$ value generated from two-tailed $t$-test) ≥15, protein −log₁₀($P$) ($P$ value generated from two-tailed $t$-test) ≥25, proteins unique peptides ≥2, de novo ALC score ≥50% and false discovery rate < 1%. All the selected proteins are shown in Supplementary Tables 1–3.

## Antibodies

The antibodies used were ERα (HC-20, sc-543, Santa Cruz for ChIP, 15 µl per sample; HC-20 and H-184, Santa Cruz for western blot, 1:1,000 dilution); ANTI-FLAG M2 Affinity Gel (A2220) (Sigma, 20 µl per sample for IP); RNA Pol II (N-20, sc-899, Santa Cruz, 15 µl per sample for ChIP); PolIISer2P (ab5095, Abcam, 3 µg per sample); HA (ab9110, Abcam, 3 µg per sample for ChIP); AP2γ (H-77, sc-8977, Santa Cruz, 15 µl per sample for ChIP); GATA3 (HG3-31, sc-268, Santa Cruz, 15 µl per sample for ChIP); CBP (C15410224, Diagenode, 3 µg per sample for ChIP); H3K27Ac (ab4729, Abcam, 3 µg per sample for ChIP); H3K9me3 (ab8898, Abcam, 3 µg per sample for ChIP); Top1 (A302-589A, Bethyl, 1:1,000 dilution for western blot, 0.5 µg per sample for CUT&RUN assays); TOP1cc (TG2017-2, TopoGEN, Lot no. 17AG15, 0.5 µg per sample for CUT&RUN assays); BRD4 (C15410337, Diagenode, 3 µg per sample for ChIP); FOXA1

(C15410231, Diagenode, 3 µg per sample for ChIP); MED26 (13641S, Cell Signaling, 15 µl per sample for ChIP; Bethyl A302-371A, 1:1,000 dilution for western blot).

## CUT&RUN assays

CUT&RUN was performed using the CUTANA CUT&RUN Protocol (www.epicypher.com) which is an optimized version of that previously described[55,56]. For each sample, $5 \times 10^5$ cells were immobilized onto Concanavalin-A beads (EpiCypher catalog no. 21-1401) and incubated overnight (4 °C with gentle rocking) with 0.5 µg of antibody. CUT&RUN-enriched DNA was purified and 10 ng used to prepare sequencing libraries with the KAPA HTP/LTP Library Preparation Kits (Roche catalog no. 07961880001). Libraries were sequenced with Illumina HiSeq 4000 or NovaSeq 6000 system according to the manufacturer's instructions. Paired end fastq files were aligned to the hg38 or mm10 reference genome using the Bowtie v.2 algorithm. Only uniquely aligned reads were retained for subsequent analyses.

## ChIP and ChIP–seq

ChIP was performed as described previously[57]. Briefly, cells were crosslinked with 1% formaldehyde at room temperature for 10 min. The cross-linking was then quenched with 0.125 M glycine for 5 min. Chromatin was fragmented using a Bioruptor Pico (Diagenode) for 10 min at high power, with an interval of 30 s between pulses to get around 200 bp fragments and precleared using 20 µl Protein G Dynabeads (Life Technologies, catalog no. 10009D). Subsequently, the soluble chromatin was incubated with 2–5 µg antibodies at 4 °C overnight. Immunoprecipitated complexes were collected using 30 µl Protein G Dynabeads, which have been saturated with PBS/1% BSA overnight at 4 °C, per reaction. For all ChIPs, after decrosslinking overnight at 65 °C, final ChIP DNA was extracted and purified using QIAquick spin columns (Qiagen). ChIP–seq libraries were constructed following Illumina's ChIP–seq sample prep kit. The library was amplified by 14 cycles of PCR.

## PRO-seq

PRO-seq experiments were performed as previously described with a few modifications[57,58]. Briefly, around 2 million MCF7 cells treated with $E_2$ for 1 h were washed three times with cold PBS and then swelled sequentially in swelling buffer (10 mM Tris-HCl pH 7.5, 2 mM MgCl$_2$, 3 mM CaCl$_2$) for 10 min on ice, harvested and lysed in lysis buffer (swelling buffer plus 0.5% Nonidet P-40, 20 U of SUPERase-In and 10% glycerol). The resultant nuclei were washed two more times with 10 ml lysis buffer and finally resuspended in 100 µl freezing buffer (50 mM Tris-HCl pH 8.3, 40% glycerol, 5 mM MgCl$_2$, 0.1 mM EDTA). For the run-on assay, resuspended nuclei were mixed with an equal volume of reaction buffer (10 mM Tris-HCl pH 8.0, 5 mM MgCl$_2$, 1 mM dithiothreitol, 300 mM KCl, 20 units of SUPERase-In, 1% sarkosyl, 250 µM A/GTP, 50 µM biotin-11-C/UTP (Perkin-Elmer)) and incubated for 5 min at 30 °C. The resultant nuclear-run-on RNA was then extracted with TRIzol LS reagent (Life Technologies, catalog no. 10296-028) following the manufacturer's instructions. Nuclear-run-on RNA was fragmented to around 200–500 nt by alkaline base hydrolysis on ice for 30 min and neutralized by adding one volume of 1 M Tris-HCl pH 6.8. Excessive salt and residual NTPs were removed by using a P-30 column (Bio-Rad, catalog no. 732-6250), followed by treatment with DNase I (Promega catalog no. M6101) and Antarctic phosphatase (NEB catalog no. M0289L). Fragmented nascent RNA was bound to 10 µl of MyOne Streptavidin C1 dynabeads (Invitrogen, catalog no. 65001) following the manufacturer's instructions. The beads were washed twice in high salt (2 M NaCl, 50 mM Tris-HCl pH 7.5, 0.5% Triton X-100, 0.5 mM EDTA), once in medium salt (1 M NaCl, 5 mM Tris-HCl pH 7.5, 0.1% Triton X-100, 0.5 mM EDTA) and once in low salt (5 mM Tris-HCl pH 7.5, 0.1% Triton X-100). Bound RNA was extracted from the beads using Trizol (Invitrogen, catalog no. 15596-018) in two consecutive extractions, and

the RNA fractions were pooled, followed by ethanol precipitation, and PRO-seq libraries were prepared with NEBNext Small RNA Library Prep Kit (NEB, catalog no. E7330).

## Deep sequencing

For all high-throughput sequencing, the extracted DNA libraries were sequenced with an Illumina HiSeq 4000 or NovaSeq 6000 system according to the manufacturer's instructions. DNA sequences generated by the Illumina Pipeline were aligned to the human (hg38) or mouse (mm10) genome assembly using Bowtie v.2 (ref. [59]). The data were visualized by preparing custom tracks on the University of California Santa Cruz genome browser using the HOMER software package[4]. For each experiment presented in this study, the total number of mappable reads was normalized to $10^7$.

## Identification of ChIP–seq peaks and TOP1cc-enriched regions

ChIP–seq peak identification, quality control and motif analysis were performed using Samtools[60] and HOMER[4] as described in our previously published methods[25,61]. Briefly, we created tag directories for each individual sample, allowing no more than two tags per base pair and the combined replicates of each treatment, and then normalized each directory by the total number of mapped tags such that each directory contains 10 million tags. We next made peak calls with a very low threshold as required for IDR (findPeaks -style factor -o auto) on the individual samples, combined replicates, individual pseudo replicates and combined pseudo replicates. We then applied the HOMER-IDR program[4] to format the data for the IDR R package to determine the IDR threshold and identify the top peaks above that threshold. TOP1cc peak identification, quality control and motif analysis were performed following the same rules we used for ChIP–seq.

## Heatmap and tag density analyses

To generate histograms for the average distribution of tag densities, position-corrected, normalized tags in 100 bp windows were tabulated within the indicated distance from specific sites in the genome. Clustering plots for normalized tag densities at each genomic region were generated using HOMER[4] and then clustered using Gene Cluster 3.0 (ref. [62]) and visualized using Java TreeView[63].

## PRO-seq analysis

PRO-seq data analyses were performed as previously reported[57]. The sequencing reads were aligned to hg38 using Bowtie v.2 using very sensitive parameters. The common artifacts derived from clonal amplification were circumvented by considering maximal three tags from each unique genomic position as determined from the mapping data. To determine $E_2$-dependent changes in gene body, the sequencing reads for RefSeq genes were counted over the first 13 kb of the entire gene body, excluding the 500 bp promoter-proximal region on the sense strand with respect to the gene orientation by using HOMER[4]. EdgeR[64] was used to compute the significance of the differential gene expression (fold change (FC) ≥ 1.5, false discovery rate ≤ 0.01). Additionally, a read density threshold (that is, normalized total read counts per kilobase) was used to exclude low-expressed genes. PRO-seqs were normalized to 10 million tags, and HOMER[4] was used to quantify eRNA expression by tabulating normalized tag numbers surrounding ±1,000 bp from the center of the peaks. eRNAs with a FC > 1.5 in PRO-seq signals were differentially expressed.

## Bioinformatic characterization of enhancer groups

We followed our previously published method to define enhancer groups in MCF7 cells[13]. Briefly, putative enhancer sites were first defined based on ChIP–seq enrichment of H3K27Ac (GSM1115992) flanking ±1,000 bp from the center of the ERα peaks or assay for transposase accessible chromatin with high-throughput sequencing (ATAC–seq) peaks.

ERα-marked MegaTrans enhancers were defined in our previous reports with the following criteria: (1) regions are at least 3 kb away from annotated transcription start sites (TSSs); (2) regions have at least 16 tags from H3K27Ac ChIP–seq normalized to 10 million tags; (3) regions are at least 10 tags from PRO-seq normalized to 10 million tags when MCF7 cells were treated with $E_2$; and (4) FC of eRNA expression between $E_2$ and ethanol conditions was at least 1.5.

ERα-marked other active enhancers were defined by the following criteria: (1) regions were at least 3 kb away from annotated TSSs; (2) regions had at least 16 tags from H3K27Ac ChIP–seq normalized to 10 million tags; (3) regions had at least 10 tags from PRO-seq normalized to 10 million tags when MCF7 cells were treated with either ethanol or $E_2$; and (4) FC of eRNA expression between $E_2$ and ethanol condition were more than 0.67 and less than 1.5.

Other active enhancers were defined by the following criteria: (1) regions were at least 3 kb away from annotated TSSs and were not marked by ERα; (2) regions had at least 16 tags from H3K27Ac ChIP–seq normalized to 10 million tags; (3) regions had at least 10 tags from PRO-seq normalized to 10 million tags when MCF7 cells were treated with either ethanol or $E_2$; and (4) FC of eRNA expression between $E_2$ and ethanol conditions were more than 0.67 and less than 1.5.

p65-marked proinflammatory enhancers were defined by the following criteria: (1) regions are at least 3 kb away from annotated TSSs; (2) regions have at least 16 tags from H3K27Ac ChIP–seq normalized to 10 million tags; and (3) regions are at least 16 tags from Pol II ChIP–seq and p65 ChIP–seq signals normalized to 10 million tags when MCF7 cells were treated with TNFα.

DHT-induced active enhancers were defined by the following criteria: (1) regions are at least 3 kb away from annotated TSSs; (2) regions have at least 16 tags from H3K27Ac ChIP–seq normalized to 10 million tags; (3) regions are at least 10 tags from Pol II ChIP–seq and AR ChIP–seq signals normalized to 10 million tags when LNCAP cells were treated with DHT; and (4) FC of Pol II ChIP–seq tags between DHT and Veh condition is at least 1.5.

KCl-induced neuronal enhancers were defined by the following criteria: (1) regions are at least 3 kb away from annotated TSSs; (2) regions have at least 16 tags from H3K27Ac ChIP–seq normalized to 10 million tags; (3) regions are at least 10 tags from Pol II ChIP–seq signals normalized to 10 million tags when primary cortical neurons were treated with KCl (30 mins); and (4) FC of Pol II ChIP–seq tags between KCl (30 mins) and KCl (0 min) conditions is at least 1.5.

## Motif analysis and gene ontology analysis

For de novo motif analysis, transcription factor motif finding was performed on ±200 bp relative to the centers defined from ChIP–seq peaks or TOP1cc peaks using HOMER[4]. Peak sequences were compared with random genomic fragments of the same size and normalized G/C content to identify motifs enriched in the ChIP–seq targeted sequence. Sequence logos were generated using WebLOGO[65]. Gene ontology analysis was performed with Metascape[66].

## Overlaps

The overlaps between sites identified in ChIP–seq for DNA-binding proteins and TOP1cc signals were calculated using BEDTools[67] and their statistical significance (versus background distribution) was confirmed using HOMER[4].

## Statistics and reproducibility

All qPCR experiments were performed with at least three independent biological replicates, and results are shown as means ± s.d. Statistical analyses were conducted using Prism v.6 software (GraphPad Software). Statistical comparisons between groups were analyzed for significance by paired two-tailed $t$-test. Differences are considered significant at $P < 0.05$. NS, nonstatistically significant; **$P < 0.01$; ***$P < 0.001$. The exact values of $n$, statistical measures (mean ± s.d.) and statistical significance are reported in the figure legends. For western blots in Figs. 1d, 3b,e,f and 4c and Extended Data Figs. 3a, 7a and 9a, at least two independent biological replicates were performed. For ChIP–seq, ATAC–seq and all the CUT&RUN assays, we initially generated two biological replicates and calculate the Pearson correlation. If the correlation was <0.9, additional replicates were generated. For PRO-seq experiments, we generated a minimum of three biological replicates. For all the boxplots for the genome-wide experiments analysis, unpaired two-tailed $t$-tests were performed. For all the Pearson correlations in Fig. 3d and Extended Figure 6c–e, one-tailed $t$-tests were adopted.

## Data resources

We used some published ChIP–seq data from the Gene Expression Omnibus database for DNA-PKcs under accession number GSE60270 (ref. [24]), and GRO-seq data for MCF7 with 1 h $E_2$ treatment under accession number GSE45822 (ref. [25]).

## Reporting summary

Further information on research design is available in the Nature Portfolio Reporting Summary linked to this article.

## Data availability

Most data are available in the main text or the supplementary materials. Whole genome sequencing datasets have been deposited to NCBI GSE135808. Please direct any requests for further information or reagents to the lead contact M.G.R., School of Medicine, UCSD, La Jolla, CA 92093, USA. Source data are provided with this paper.

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

## Acknowledgements

The authors are grateful to J. Hightower for assistance with figure preparation; to M. J. Friedman at UCSD for the plasmids encoding Top1 wt and Y723F mutant and to other members of the Rosenfeld laboratory for generous help throughout this work. M.G.R. was an investigator with the Howard Hughes Medical Institute during this study. This work was supported by grants from National Institute of Diabetes and Digestive and Kidney Diseases (RO1DK018477, RO1DK039949), National Heart, Lung, and Blood Institute (R01HL150521) and National Institute of Neurological Disorders and Stroke (RO1NS034934) to M.G.R., and a grant from National Institute of General Medical Science (R35-GM131780) to A.K.A. S.J.N. is supported by RO3DK131250.

## Author contributions

Y.T. and M.G.R. conceived the project. Y.T. performed most of the experiments reported with the assistance from L.Y. A.G. isolated the primary cortical neurons from E15.5 C57BL/6 embryonic mouse cortices with assistance from H.T. S.J.N. expressed and purified the eRNAs for the TOP1 activity assay. Y.T. performed most of bioinformatics analysis, with input from A.K.A. K.O. prepared the libraries and performed deep sequencing. Y.T. and M.G.R. wrote the paper with input from A.K.A., A.G. and S.J.N.

## Competing interests

The authors declare no competing interests.

## Additional information

**Extended data** is available for this paper at https://doi.org/10.1038/s41594-022-00883-8.

**Correspondence and requests for materials** should be addressed to Yuliang Tan or Michael G. Rosenfeld.

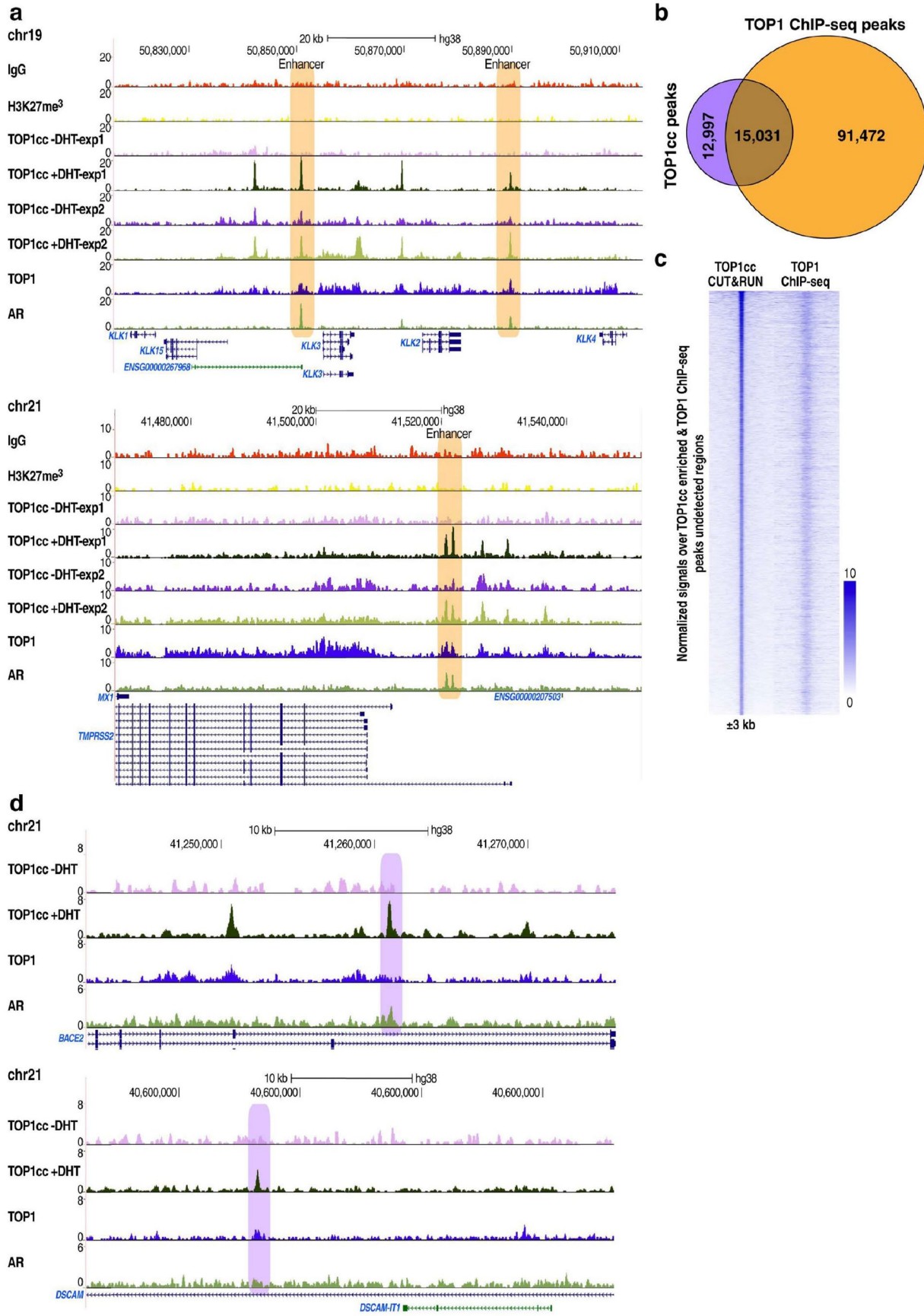

**Extended Data Fig. 1 | See next page for caption.**

**Extended Data Fig. 1 | CUT&RUN assays disclose TOP1cc signals in LNCAP cells.** (**a**) Genomic browsers show TOP1cc CUT&RUN signals, AR and Top1 ChIP-seq signals in DHT treated LNCAP cells. IgG and H3K27me3 CUT&RUN signals are serving as the experimental controls for the CUT&RUN assays. Enhancers are highlighted with light-brown boxes. (**b**) Co-localization analysis of Top1 ChIP-seq signals and TOP1cc CUT&RUN signals in DHT treated LNCAP cells. (**c**) Heatmaps show Top1 ChIP-seq signals and TOP1cc CUT&RUN signals at TOP1cc-enriched but Top1 ChIP-seq peak undetected regions. Additional 3 kb from the center of the peaks are shown, and the color scale shows the normalized tag numbers. (**d**) Genomic browsers show TOP1cc CUT&RUN signals, AR and Top1 ChIP-seq signals in DHT treated LNCAP cells. TOP1cc-enriched but Top1 peaks undetected regions are highlighted with light-purple boxes.

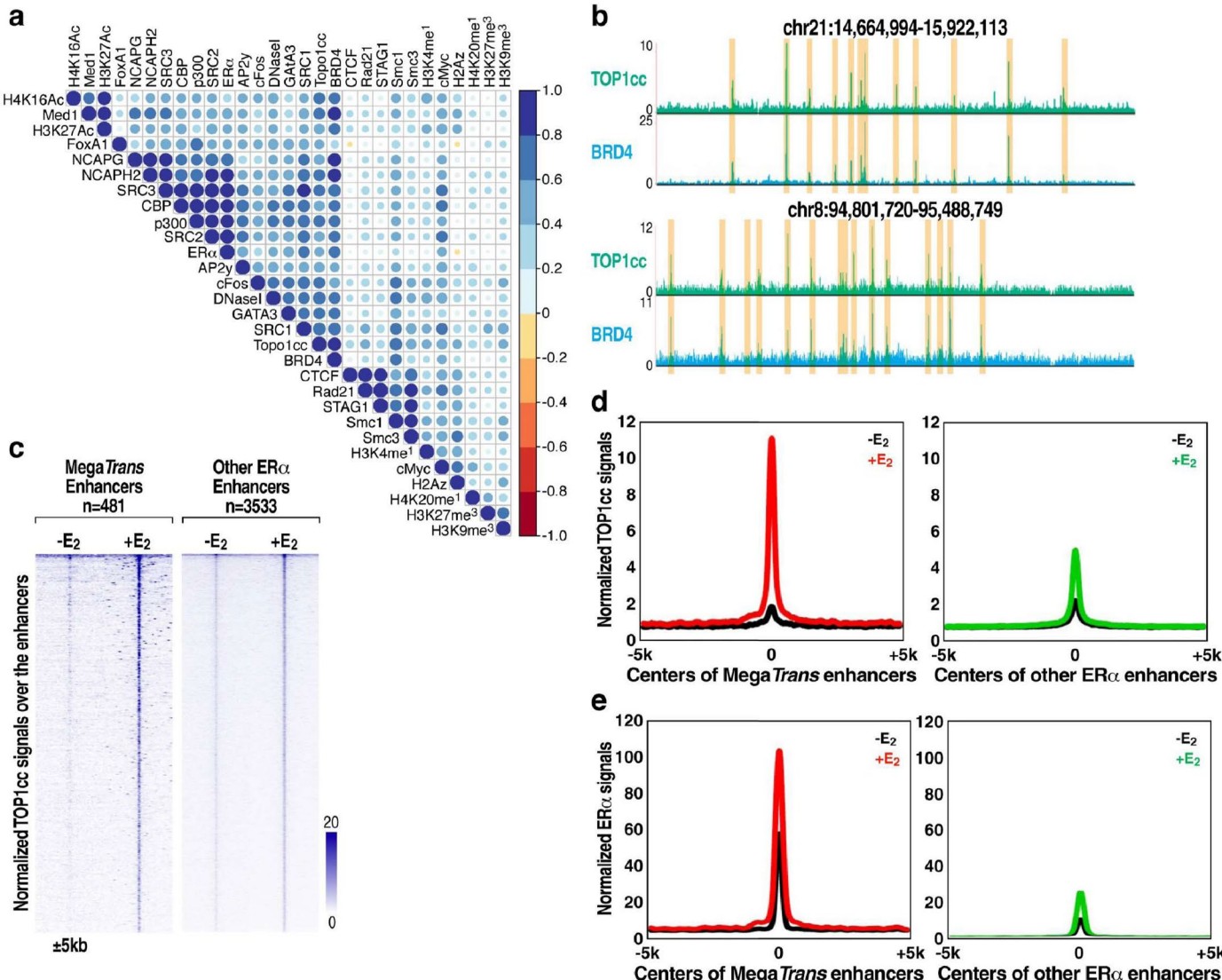

**Extended Data Fig. 2 | TOP1cc is associated with transcriptional activation at ERα marked active enhancers.** (**a**) Pearson correlation between TOP1cc CUT&RUN signals with histone modifications, coactivators and other signals at the active enhancers in $E_2$ treated MCF7 cells was plotted. (**b**) Genome browsers show TOP1cc CUT&RUN signals and BRD4 ChIP-seq signals at the selected loci in $E_2$ treated MCF7 cells. TOP1cc and BRD4 overlapped regions are highlighted with light-brown boxes. (**c**) Heatmaps show TOP1cc CUT&RUN signals and

ERα ChIP-seq signals at ERα marked robustly activated MegaTrans enhancers (n = 481) and ERα marked non MegaTrans enhancers (n = 3,533). Additional 3 kb from the center of the peaks are shown, and the color scale shows the normalized tag numbers. (**d**, **e**) Histogram plots show TOP1cc CUT&RUN signals (**d**) and ERα ChIP-seq signals (**e**) at ERα marked robustly activated MegaTrans and non MegaTrans enhancers. Additional 5 kb from the center of the peaks are shown, and the color scale shows the normalized tag numbers.

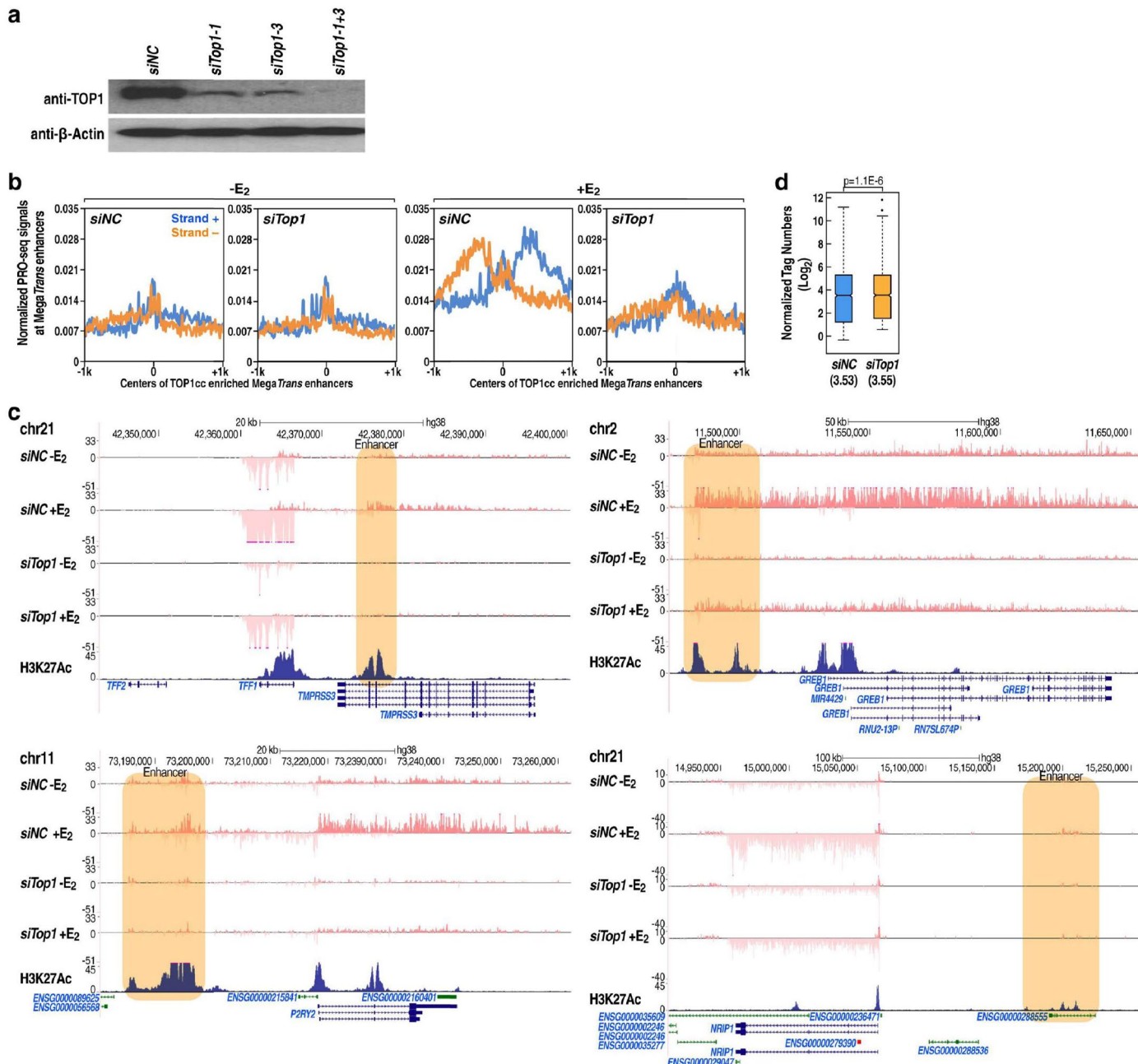

**Extended Data Fig. 3 | TOP1cc is required for E₂-dependent enhancer activation.** (**a**) Western blots show the expression of Top1 protein upon the knock-down of *Top1* mRNA with siRNAs. (**b**) Histogram plots show the normalized PRO-seq tag intensities at TOP1cc-enriched MegaTrans enhancers. Additional 1 kb from the center of the peaks is shown. (**c**) Box-and-whisker plots show PRO-seq tags (Log₂) for other active enhancers (n = 3,533) upon the knock-down of *Top1* gene in MCF7 cells. P value generated from unpaired two-tailed t

test denotes statistical differences between si*NC* and si*Top1* conditions. Center lines show the medians, box limits indicate the 25th and 75th percentiles, and whiskers extend 1.5× the interquartile range from the 25th and 75th percentiles. (**d**) Genome browsers show Top1 is required for E₂ induced robustly activation of the selected genes and enhancers by PRO-seq. Enhancers are highlighted with light-brown boxes. Uncropped images for **a** are available as source data.

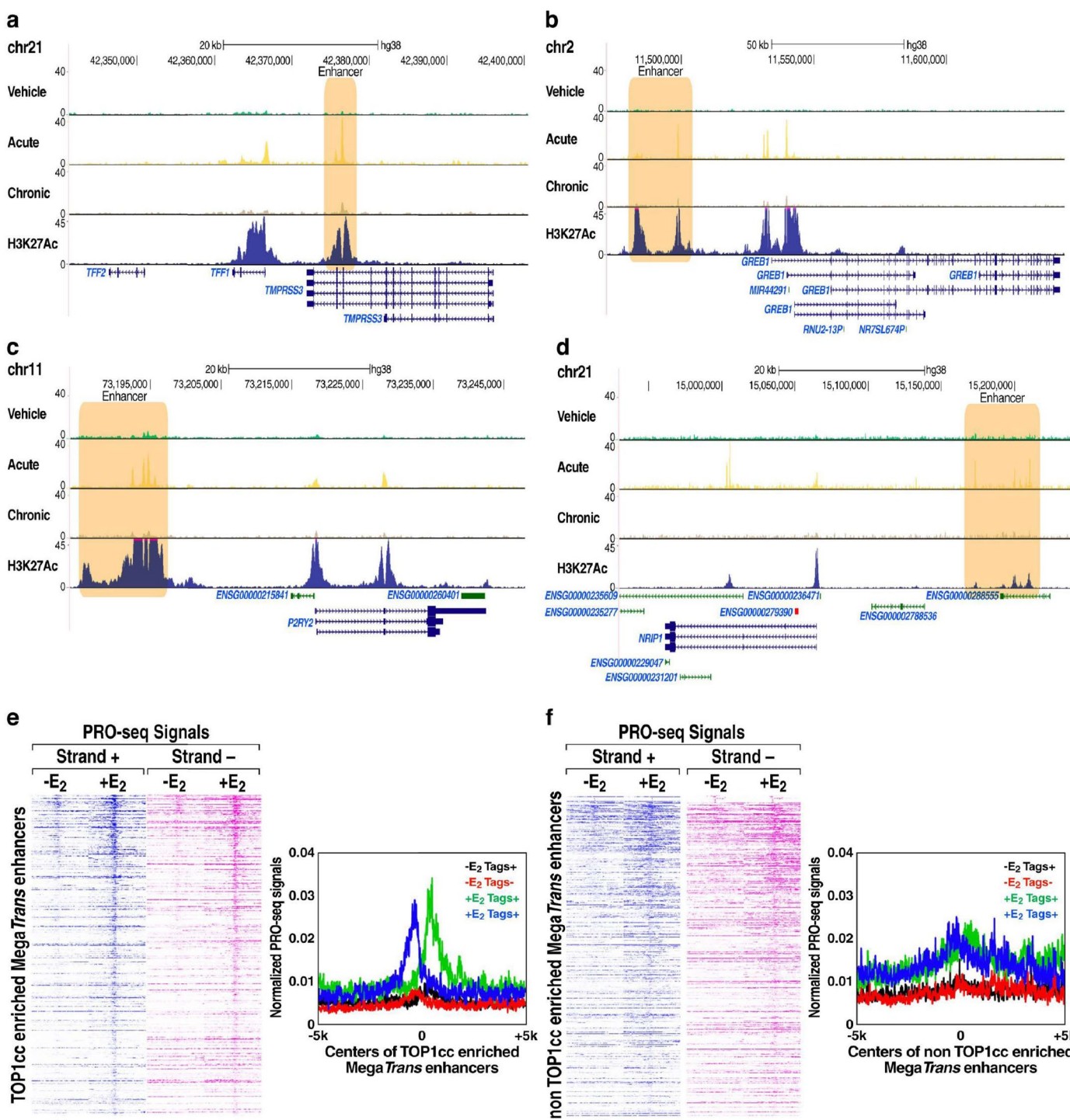

**Extended Data Fig. 4 | Enhancer RNAs are highly expressed at TOP1cc-enriched MegaTrans enhancers.** (**a–d**) Genomic browsers show TOP1cc CUT&RUN signals and H3K27Ac ChIP-seq signals at the selected enhancers upon acute $E_2$ (1 hr) treatment and chronical $E_2$ (~14 hrs) treatment. Enhancers are highlighted with light-brown boxes. (**e**) Heatmaps and histogram plots show

PRO-seq signals at TOP1cc-enriched MegaTrans enhancers. Additional 3 kb from the center of the peaks are shown, and the color scale shows the normalized tag numbers. (**f**) Heatmaps and histogram plots show PRO-seq signals at nonTOP1cc enriched MegaTrans enhancers. Additional 3 kb from the center of the peaks are shown.

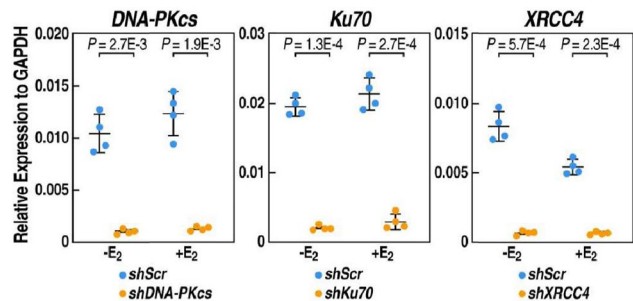

**Extended Data Fig. 5 | Knock-down efficiencies are shown.** RT–qPCR results show the relative expression levels of *Ku70, XRCC4* and *DNA-PKcs* genes in response to *shRNA* treatments in MCF7 cells. Data are shown as mean ± SD (n = 3, two-tailed student's t-test); n.s. = non-statistically significant. Raw data for these graphs are available as source data.

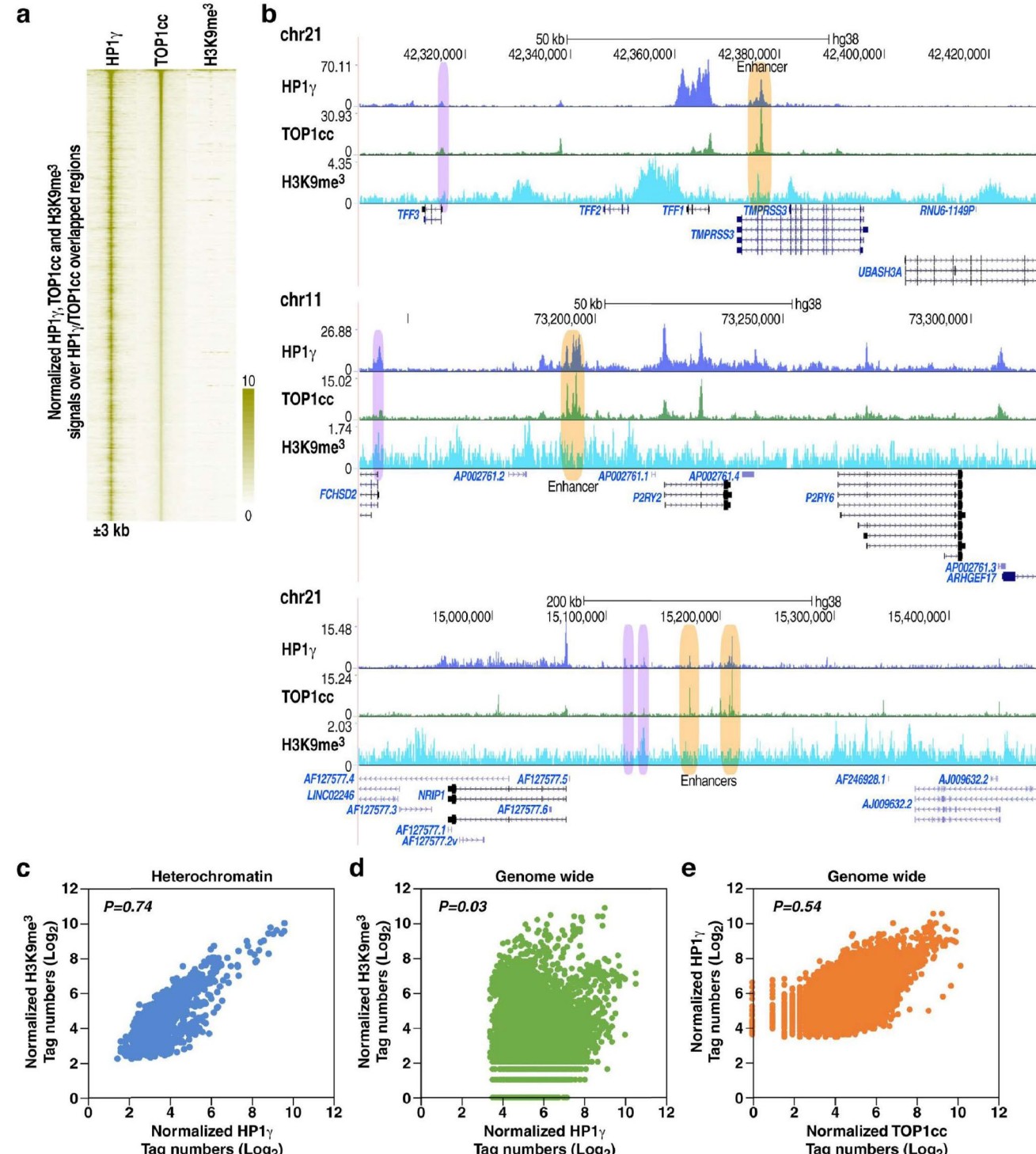

**Extended Data Fig. 6 | HP1γ signals are not correlated with H3K9me3 at euchromatin regions.** (**a**) Heatmaps show the TOP1cc-seq signals and H3K9me3 ChIP-seq signals over TOP1cc and HP1γ co-bound regions. Additional 3 kb from the center of the peaks are shown, and the color scale shows the normalized tag numbers. (**b**) Genomic browsers show TOP1cc CUT&RUN signals, HP1γ and H3K9me3 ChIP-seq signals at selected loci. Enhancer and TOP1cc associated HP1γ are highlighted with light-brown boxes. H3K9me3 associated HP1γ are highlighted with light-purple boxes. (**C**) Spearman's correlation analysis on the H3K9me3 ChIP-seq signals and HP1 ChIP-seq signals at H3K9me3 enriched heterochromatin regions is shown. (**D**) Spearman's correlation analysis on the H3K9me3 and HP1γ ChIP-seq signals at TOP1cc and HP1γ co-localized regions is shown. (**E**) Spearman's correlation analysis on the TOP1cc CUT&RUN signals and HP1γ ChIP-seq signals at TOP1cc and HP1γ co-locolaized regions is shown.

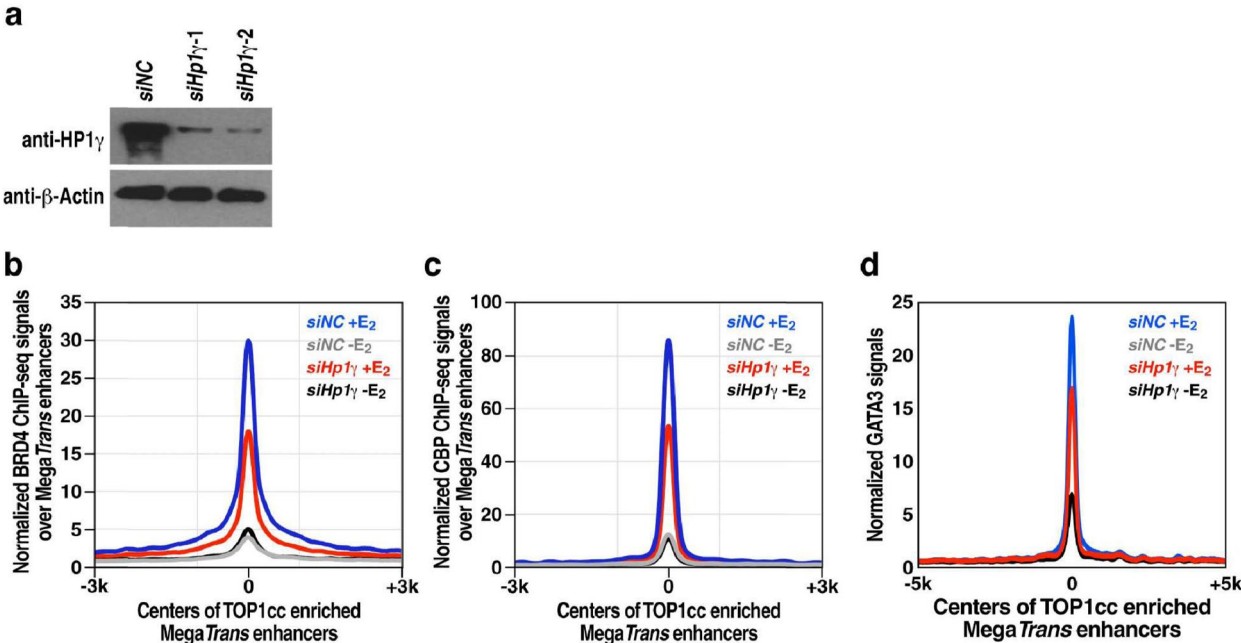

**Extended Data Fig. 7 | HP1γ is required for the acute enhancer activation.**
(**a**) Western blots show the expression of HP1γ protein in MCF7 cells. (**b**–**d**)
Histogram plots show the normalized BRD4 (**b**), CBP (**c**) and GATA3 (**d**) ChIP-seq
tags centered on peaks at TOP1cc-enriched MegaTrans enhancers. Additional
3 kb from the center of the peaks are shown. Uncropped images for **a** are available
as source data.

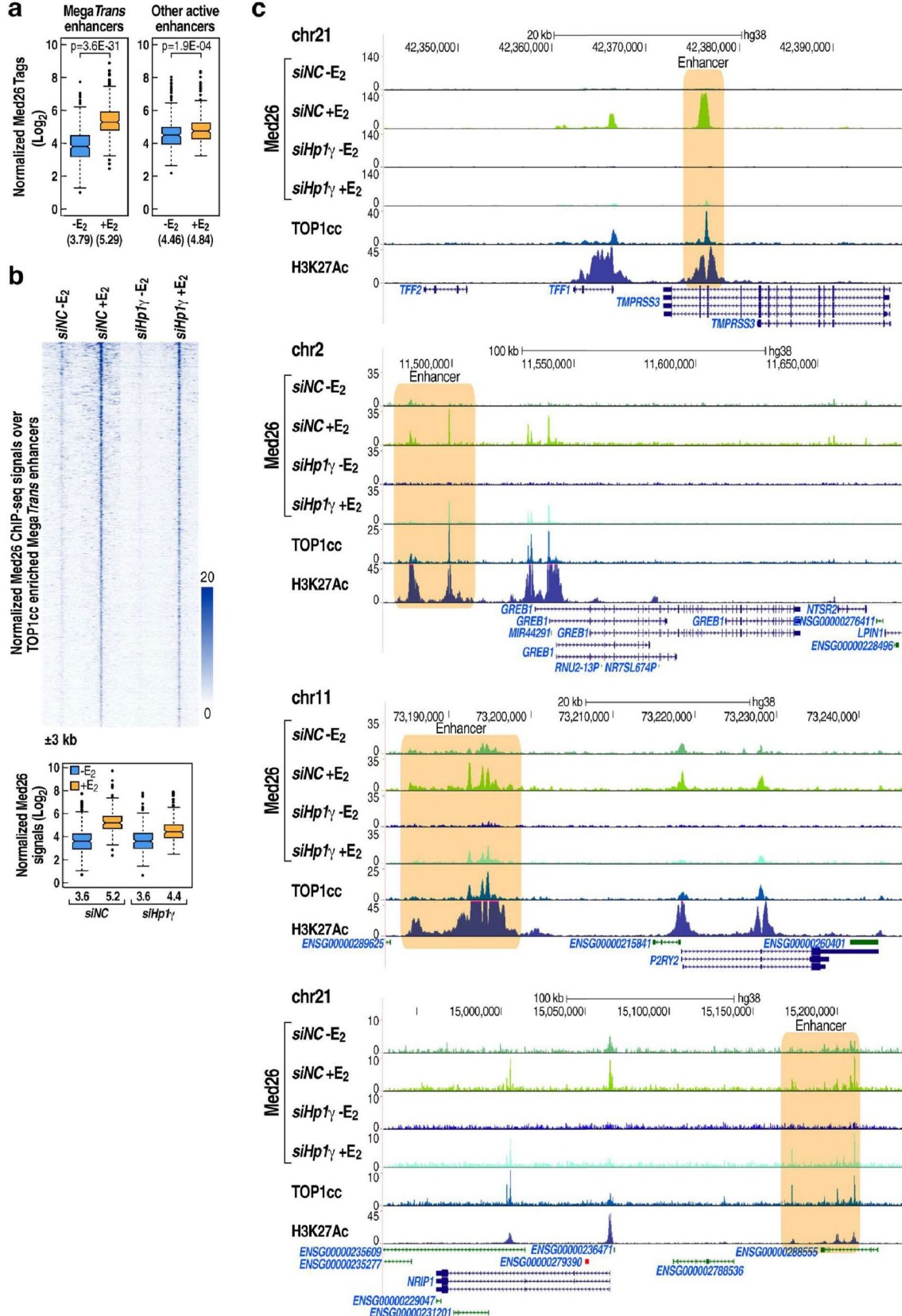

**Extended Data Fig. 8 | See next page for caption.**

**Extended Data Fig. 8 | HP1γ is required for the tethering of Med26 to the acutely activated enhancers.** (**a**) Box-and-whisker plots show Med26 ChIP-seq signals at and TOP1cc-enriched MegaTrans enhancers (n = 481) and other active enhancers (n = 3,533), which have similar eRNA expression levels with $E_2$ treated TOP1cc-enriched MegaTrans enhancers in MCF7 cells. P values generated from unpaired two-tailed t test denote statistical differences between −$E_2$ and +$E_2$ conditions in the box-and-whisker plots. The median value of normalized Med26 ChIP-seq tags ($Log_2$) are listed under the boxplots. (**b**) Heatmaps and box-and-whisker plots show Med26 ChIP-seq signals at TOP1cc-enriched MegaTrans enhancers (n = 481). Additional 3 kb from the center of the peaks are shown, and the color scale shows the normalized tag numbers, P values generated from unpaired two-tailed t test denote statistical differences between *shNC* + $E_2$ and *siHp1γ* + $E_2$ conditions in the box-and-whisker plots. The median values of normalized Med26 ChIP-seq tags ($Log_2$) are listed under the boxplots. (**c**) Genomic browsers show Med26 ChIP-seq, H3K27Ac ChIP-seq, and TOP1cc CUT&RUN signals at the selected gene locus. Enhancers are highlighted with light-brown box. For **a** and **b**, center lines show the medians, box limits indicate the 25th and 75th percentiles, and whiskers extend 1.5× the interquartile range from the 25th and 75th percentiles. The median values of normalized Med26 ChIP-seq tags ($Log_2$) are listed under the boxplots.

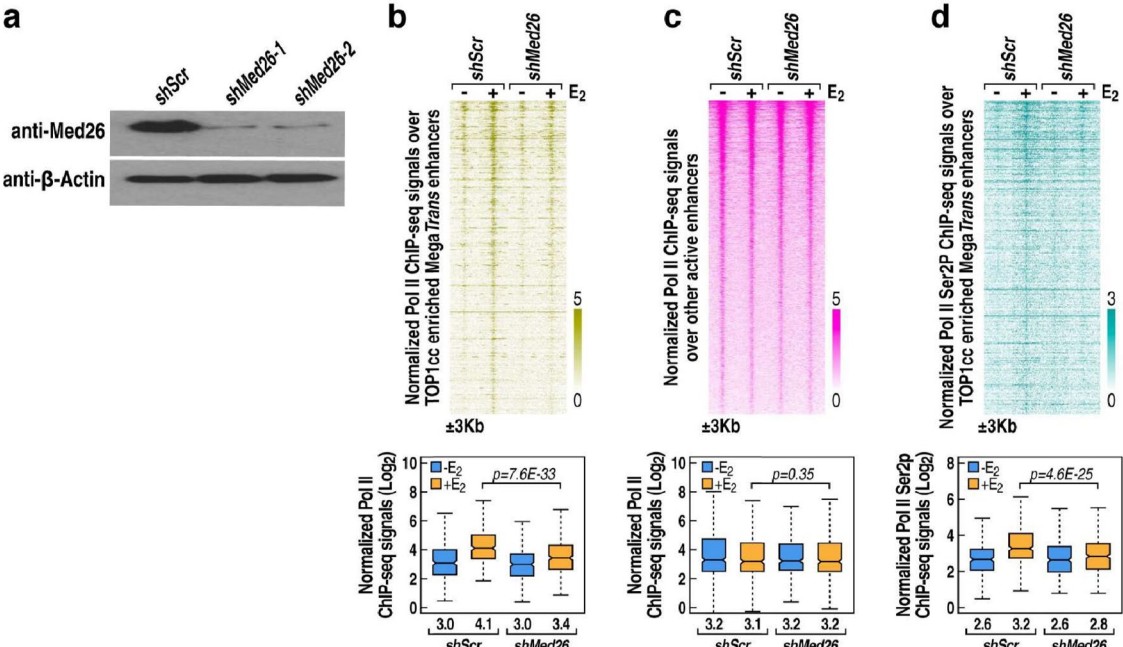

**Extended Data Fig. 9 | Med26 is involved in the transcriptional activation at enhancers.** (**a**) Western blots show the expression of Med26 protein upon the knock-down of *Med26* with shRNAs. (**b**) Heatmaps and box-and-whisker plots show Pol II ChIP-seq signals and PolIISer2p ChIP-seq signals at TOP1cc-enriched MegaTrans enhancers (n = 481). (**c**) Heatmaps show Pol II ChIP-seq signals at other active enhancers (n = 3,533). (**d**) Heatmaps and box-and-whisker plots show PolIISer2p ChIP-seq signals at TOP1cc-enriched MegaTrans enhancers (n = 481). For **b**, **c** and **d**, additional 3 kb from the center of the peaks are shown, and the color scale shows the normalized tag numbers. For **b**, **c** and **d**, P values generated from unpaired two-tailed t test denote statistical differences between *shSrc* and *shMed26* under $E_2$ conditions in the box-and-whisker plots. Center lines show the medians, box limits indicate the 25th and 75th percentiles, and whiskers extend 1.5× the interquartile range from the 25th and 75th percentiles. The median values of normalized Pol II ChIP-seq tags ($Log_2$) are listed under the boxplots. Uncropped images for **a** are available as source data.

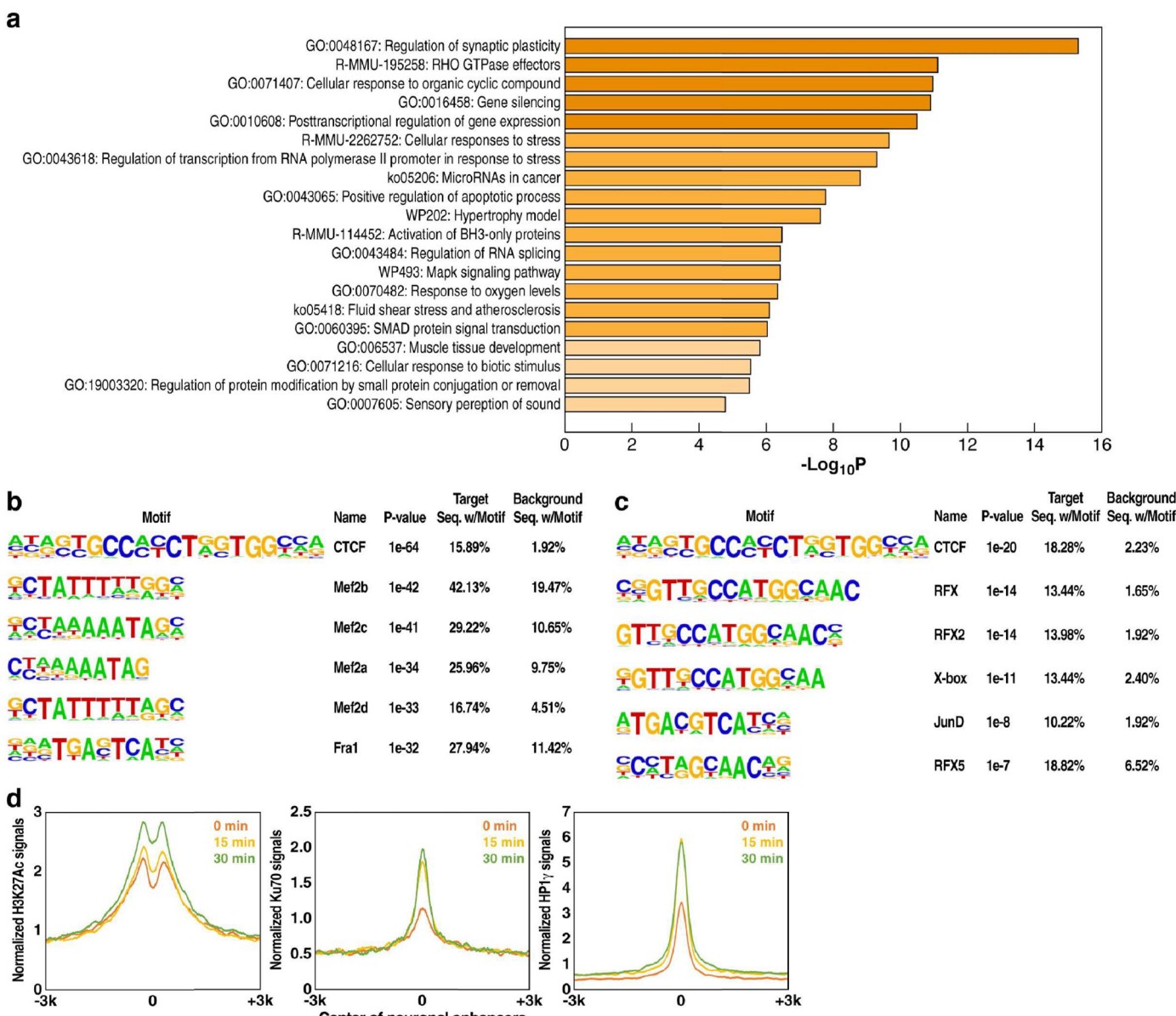

**Extended Data Fig. 10 | TOP1cc/HP1γ/Ku70 are employed in neuronal enhancers.** (**a**) Gene ontology analysis shows the functions of genes induced by 30 mins of KCl treated primary cortical neurons. (**b**) Motifs for sequences within 200 bp of the summit of the TOP1cc-enriched neuronal enhancers in KCl treated primary cortical neurons are presented. (**c**) Motifs for sequences within 200 bp of the summit of the non-TOP1cc-enriched neuronal enhancers are presented.

For **b** and **c**, P values generated from two-tailed binomial test denote statistical differences between the target and background sequences for enrichment. (**d**) Histogram plots show the normalized H3K27Ac, HP1γ and Ku70 ChIP-seq tag intensities centered on peaks at TOP1cc-enriched neuronal enhancers. Additional 3 kb from the center of the peaks are shown.

# Reporting Summary

## Statistics

For all statistical analyses, confirm that the following items are present in the figure legend, table legend, main text, or Methods section.

| n/a | Confirmed | |
|---|---|---|
| ☐ | ☒ | The exact sample size (*n*) for each experimental group/condition, given as a discrete number and unit of measurement |
| ☐ | ☒ | A statement on whether measurements were taken from distinct samples or whether the same sample was measured repeatedly |
| ☐ | ☒ | The statistical test(s) used AND whether they are one- or two-sided<br>*Only common tests should be described solely by name; describe more complex techniques in the Methods section.* |
| ☒ | ☐ | A description of all covariates tested |
| ☐ | ☒ | A description of any assumptions or corrections, such as tests of normality and adjustment for multiple comparisons |
| ☐ | ☒ | A full description of the statistical parameters including central tendency (e.g. means) or other basic estimates (e.g. regression coefficient) AND variation (e.g. standard deviation) or associated estimates of uncertainty (e.g. confidence intervals) |
| ☐ | ☒ | For null hypothesis testing, the test statistic (e.g. *F*, *t*, *r*) with confidence intervals, effect sizes, degrees of freedom and *P* value noted<br>*Give P values as exact values whenever suitable.* |
| ☒ | ☐ | For Bayesian analysis, information on the choice of priors and Markov chain Monte Carlo settings |
| ☒ | ☐ | For hierarchical and complex designs, identification of the appropriate level for tests and full reporting of outcomes |
| ☐ | ☒ | Estimates of effect sizes (e.g. Cohen's *d*, Pearson's *r*), indicating how they were calculated |

*Our web collection on statistics for biologists contains articles on many of the points above.*

## Software and code

Policy information about availability of computer code

Data collection | For all high throughput sequencing, the extracted DNA libraries were sequenced with Illumina HiSeq 4000 or NovaSeq 6000 system according to the manufacturer's instructions. And DNA sequences generated by the Illumina Pipeline were aligned to the human genome (hg38) or mouse genome (mm10) assembly using Bowtie2. The data were visualized by preparing custom tracks on the University of California, Santa Cruz (UCSC) genome browser using HOMER software package.
Clustering plots for normalized tag densities at each genomic region were generated using HOMER and then clustered using Gene Cluster 3.0 and visualized using Java TreeView.
EdgeR was used to compute the significance of the differential gene expression.
Sequence logos were generated using WebLOGO. Gene ontology analysis was performed with Metascape.
The overlaps between sites identified in ChIP-seq for DNA-binding proteins and TOP1cc signals were calculated using BEDTools.

Data analysis | Analytical programs used in this paper are provided in the Methods section

For manuscripts utilizing custom algorithms or software that are central to the research but not yet described in published literature, software must be made available to editors and reviewers. We strongly encourage code deposition in a community repository (e.g. GitHub). See the Nature Portfolio guidelines for submitting code & software for further information.

# Data

Policy information about availability of data

All manuscripts must include a data availability statement. This statement should provide the following information, where applicable:

- Accession codes, unique identifiers, or web links for publicly available datasets
- A description of any restrictions on data availability
- For clinical datasets or third party data, please ensure that the statement adheres to our policy

We used some published ChIP-seq data from the Gene Expression Omnibus database for DNA-PKcs under accession number GSE60270, and GRO-seq data for MCF7 with 1hr E2 treatment under accession number GSE45822. Most data are available in the main text or the supplementary materials. Whole genome sequencing datasets have been deposited to NCBI GSE135808.

# Field-specific reporting

Please select the one below that is the best fit for your research. If you are not sure, read the appropriate sections before making your selection.

☒ Life sciences          ☐ Behavioural & social sciences          ☐ Ecological, evolutionary & environmental sciences

For a reference copy of the document with all sections, see nature.com/documents/nr-reporting-summary-flat.pdf

# Life sciences study design

All studies must disclose on these points even when the disclosure is negative.

| | |
|---|---|
| Sample size | For each experiment, the desired effect representing a difference between the populations of samples under study will be computed by estimating the mean and the variance of the distributions from an initial set of 3 biological replicates. |
| Data exclusions | No data was excluded. |
| Replication | All genome-wide experiments were replicated at least twice; and non genome-wide experiments were replicated at least 3 times. |
| Randomization | Random genomic regions were selected to present the distribution of Top1cc in the genome. |
| Blinding | All the libraries preparation for the ChIP-seq, PRO-seq, CUT&RUN assays were performed with different people to make sure the blindness. For other qPCR and Western blots experiments, the knock-down experiments and final PCR/western blot were performed by different people to make sure the blindness. For cortical neurons, the tissue preparation, library preparation were also performed with different people. |

# Reporting for specific materials, systems and methods

We require information from authors about some types of materials, experimental systems and methods used in many studies. Here, indicate whether each material, system or method listed is relevant to your study. If you are not sure if a list item applies to your research, read the appropriate section before selecting a response.

## Materials & experimental systems

| n/a | Involved in the study |
|---|---|
| ☐ | ☒ Antibodies |
| ☐ | ☒ Eukaryotic cell lines |
| ☒ | ☐ Palaeontology and archaeology |
| ☒ | ☐ Animals and other organisms |
| ☒ | ☐ Human research participants |
| ☒ | ☐ Clinical data |
| ☒ | ☐ Dual use research of concern |

## Methods

| n/a | Involved in the study |
|---|---|
| ☐ | ☒ ChIP-seq |
| ☒ | ☐ Flow cytometry |
| ☒ | ☐ MRI-based neuroimaging |

# Antibodies

| | |
|---|---|
| Antibodies used | ERα (HC20) Santa Cruz Biotechnology sc-543 (Lot# I0514)<br>ANTI-FLAG M2 Affinity Gel Sigma-Aldrich A2220<br>RNA Pol II (N20) Santa Cruz Biotechnology sc-899 (Lot# D2315)<br>HA Abcam ab9110(Lot#GR3177614-4)<br>AP2γ(H-77) Santa Cruz Biotechnology sc-8977 (Lot#G1112)<br>GATA3(HG3-31) Santa Cruz Biotechnology sc-268 (Lot#J0515)<br>CBP diagenode C15410224 (Lot#39721) |

Rad21 Abcam ab992 (Lot# GR214359-8)
H3K27Ac Abcam ab4729 (Lot#GR288020-1)
Pol II diagenode C15200004 (Lot#001-11)
FoxA1 diagenode C15410231 (Lot#39435)
BRD4 diagenode C15410337 (Lot#A2710P)
SMC1 Bethyl Laboratories A300-055A (A302-055A-6)
CTCF diagenode C15410210 (Lot#A2359-0010)
H3K4me2 Abcam ab7766 (Lot#GR102810-4)
H3K9me3 Abcam ab8898(Lot#GR3217826-1)
PolIISer2p Abcam ab5095(Lot#GR3225147-1)
H3K9me2 Cell Signaling 9753S(Lot# 4)
MED26 Bethyl A302-371A (Lot#A302-371A-1)
Med26 (13641S, Cell Signaling)
Top1 Bethyl A302-589A (Lot#A302-589A-1)
Top1cc TopoGEN TG2017-2 (Lot# 17AG15)
Ku70 Santa Cruz Biotechnology sc-9033 (Lot#B0416)
Ku-80 MyBioSource MBS8533127 (Lot#T14S11)
Anti-HP1g, clone 42s2 Millipore 05-690 (Lot#3224566)
Ku70 Bethyl A302-624A (Lot#A302-624A-1)
H3K4me3 Abcam Ab8580(Lot#GR3201182-1)
CTCF Active Motif 61311(Lot#34614003)
Guinea Pig anti-Rabbit IgG Antibodies-online.com ABIN101961(43047)
Anti-Mouse IgG Millipore 06-371(Lot#3257057)

Validation

ERα (HC20) Santa Cruz Biotechnology sc-543 (Lot# I0514) IDENTIFIER: RRID:AB_631471
ANTI-FLAG M2 Affinity Gel Sigma-Aldrich A2220 IDENTIFIER: RRID:AB_10063035
RNA Pol II (N20) Santa Cruz Biotechnology sc-899 (Lot# D2315) IDENTIFIER: RRID:AB_632359
HA Abcam ab9110(Lot#GR3177614-4) IDENTIFIER: RRID:AB_307019
AP2γ(H-77) Santa Cruz Biotechnology sc-8977 (Lot#G1112) IDENTIFIER: RRID:AB_2286995
GATA3(HG3-31) Santa Cruz Biotechnology sc-268 (Lot#J0515) IDENTIFIER: RRID:AB_2108591
CBP diagenode C15410224 (Lot#39721) IDENTIFIER: RRID:AB_2722552
Rad21 Abcam ab992 (Lot# GR214359-8) IDENTIFIER: RRID:AB_2314019
H3K27Ac Abcam ab4729 (Lot#GR288020-1) IDENTIFIER: RRID:AB_2118291
Pol II diagenode C15200004 (Lot#001-11) IDENTIFIER: RRID:AB_2728744
FoxA1 diagenode C15410231 (Lot#39435) Applications: Western Blot (WB), Immunofluorescence (IF), Immunoprecipitation (IP), ChIP/ChIP-seq, Immunohistochemistry.
BRD4 diagenode C15410337 (Lot#A2710P) Applications: Western Blot (WB), ELISA, ChIP/ChIP-seq.
SMC1 Bethyl Laboratories A300-055A (A302-055A-6) IDENTIFIER: RRID:AB_2192467
CTCF diagenode C15410210 (Lot#A2359-0010) IDENTIFIER: RRID:AB_2753160
H3K4me2 Abcam ab7766 (Lot#GR102810-4) IDENTIFIER: RRID:AB_2560996
H3K9me3 Abcam ab8898 (Lot#GR3217826-1) IDENTIFIER: RRID:AB_306848
PolIISer2p Abcam ab5095 (Lot#GR3225147-1) IDENTIFIER: RRID:AB_304749
H3K9me2 Cell Signaling 9753S (Lot# 4) IDENTIFIER: RRID:AB_659848
MED26 Bethyl A302-371A (Lot#A302-371A-1) IDENTIFIER: RRID:AB_1907254
MED26 Abcam Ab50619 IDENTIFIER: RRID:AB_869274
MED26 (13641S, Cell Signaling)  DENTIFIER: RRID: AB_2798281
Top1 Bethyl A302-589A (Lot#A302-589A-1) IDENTIFIER: RRID:AB_2034865
Top1cc TopoGEN TG2017-2 (Lot# 17AG15) Applications: Western Blot (WB), ICE blot. CUT&RUN are validated in Fig.1.
Ku70 Santa Cruz Biotechnology sc-9033 (Lot#B0416) IDENTIFIER: RRID:AB_650476
Ku-80 MyBioSource MBS8533127 (Lot#T14S11) Applications: Western Blot (WB), Immunofluorescence (IF), Immunoprecipitation (IP), ChIP/ChIP-seq.
Anti-HP1g, clone 42s2 Millipore 05-690 (Lot#3224566) IDENTIFIER: RRID:AB_309910
Ku70 Bethyl A302-624A (Lot#A302-624A-1) IDENTIFIER: RRID:AB_10554672
H3K4me3 Abcam qb8580 (Lot#GR3201182-1) IDENTIFIER: RRID:AB_306649
CTCF Active Motif 61311 (Lot#34614003) IDENTIFIER: RRID:AB_61311
Guinea Pig anti-Rabbit IgG Antibodies-online.com ABIN101961(43047) IDENTIFIER: RRID:AB_10775589
Anti-Mouse IgG Millipore 06-371(Lot#3257057) IDENTIFIER: RRID:AB_390146

# Eukaryotic cell lines

Policy information about cell lines

Cell line source(s)

MCF7, 293T and LNCAP cells

Authentication

short tandem repeat (STR) was employed to determine the authentication.

Mycoplasma contamination

Mycoplasma contamination test  was performed every 3 months for the all the MCF7, 293T and LNCAP cells used in our lab to make sure no contamination.

Commonly misidentified lines
(See ICLAC register)

N/A

# ChIP-seq

## Data deposition

☒ Confirm that both raw and final processed data have been deposited in a public database such as GEO.

☒ Confirm that you have deposited or provided access to graph files (e.g. BED files) for the called peaks.

**Data access links**
*May remain private before publication.*

https://www.ncbi.nlm.nih.gov/geo/query/acc.cgi?acc=GSE135808

**Files in database submission**

AP2y_E2.fastq.gz
AP2y_Veh.fastq.gz
BRD4_E2_exp1.fastq.gz
BRD4_E2_exp2.fastq.gz
BRD4_Veh_exp1.fastq.gz
BRD4_Veh_exp2.fastq.gz
CBP_E2.fastq.gz
CBP_Veh.fastq.gz
CBX3_siNC_E2.fastq.gz
CBX3_siNC_Veh.fastq.gz
CBX3_siTop1_E2.fastq.gz
CTCF_E2.fastq.gz
ERa_E2.fastq.gz
ERa_Veh.fastq.gz
FoxA1_E2.fastq.gz
FoxA1_Veh.fastq.gz
Gata3_E2.fastq.gz
Gata3_Veh.fastq.gz
H3K27Ac_E2.fastq.gz
H3K27Ac_Veh.fastq.gz
H3K4me3_E2.fastq.gz
H3K4me3_Veh.fastq.gz
H3K9me3_E2.fastq.gz
H3K9me3_Veh.fastq.gz
MED1_siCbx3_E2.fastq.gz
MED1_siCbx3_Veh.fastq.gz
MED1_siNC_E2.fastq.gz
MED1_siNC_Veh.fastq.gz
MED26_siCbx3_E2.fastq.gz
MED26_siCbx3_Veh.fastq.gz
MED26_siNC_E2.fastq.gz
MED26_siNC_Veh.fastq.gz
NCAPG_E2.fastq.gz
NCAPG_Veh.fastq.gz
PolIISer2P_shMed26_1_E2.fastq.gz
PolIISer2P_shMed26_1_Veh.fastq.gz
PolIISer2P_shMed26_2_E2.fastq.gz
PolIISer2P_shMed26_2_Veh.fastq.gz
PolIISer2P_shscramble_E2.fastq.gz
PolIISer2P_shscramble_Veh.fastq.gz
PolII_shMed26_1_E2.fastq.gz
PolII_shMed26_1_Veh.fastq.gz
PolII_shMed26_2_E2.fastq.gz
PolII_shMed26_2_Veh.fastq.gz
PolII_shscramble_E2.fastq.gz
PolII_shscramble_Veh.fastq.gz
PolII_siCbx3_DHT_LNCAP.fastq.gz
PolII_siCbx3_E2_MCF7.fastq.gz
PolII_siCbx3_Veh_MCF7.fastq.gz
PolII_siNC_DHT_LNCAP.fastq.gz
PolII_siNC_E2_MCF7.fastq.gz
PolII_siNC_Veh_LNCAP.fastq.gz
PolII_siNC_Veh_MCF7.fastq.gz
Rad21_E2.fastq.gz
Rad21_Veh.fastq.gz
Smc1_E2.fastq.gz
Smc1_Veh.fastq.gz
AP2y_E2.bed
AP2y_Veh.bed
BRD4_E2_exp1.bed
BRD4_E2_exp2.bed
BRD4_Veh_exp1.bed
BRD4_Veh_exp2.bed

CBP_E2.bed
CBP_Veh.bed
CBX3_siNC_E2.bed
CBX3_siNC_Veh.bed
CBX3_siTop1_E2.bed
CBX3_siTop1_Veh.bed
CTCF_E2.bed
CTCF_Veh.bed
ERa_E2.bed
ERa_Veh.bed
FoxA1_E2.bed
FoxA1_Veh.bed
Gata3_E2.bed
Gata3_Veh.bed
H3K27Ac_E2.bed
H3K27Ac_Veh.bed
H3K4me3_E2.bed
H3K4me3_Veh.bed
H3K9me3_E2.bed
H3K9me3_Veh.bed
MED1_siCbx3_E2.bed
MED1_siCbx3_Veh.bed
MED1_siNC_E2.bed
MED1_siNC_Veh.bed
MED26_siCbx3_E2.bed
MED26_siCbx3_Veh.bed
MED26_siNC_E2.bed
MED26_siNC_Veh.bed
NCAPG_E2.bed
NCAPG_Veh.bed
PolIISer2P_shMed26_1_E2.bed
PolIISer2P_shMed26_1_Veh.bed
PolIISer2P_shMed26_2_E2.bed
PolIISer2P_shMed26_2_Veh.bed
PolIISer2P_shscramble_E2.bed
PolIISer2P_shscramble_Veh.bed
PolII_shMed26_1_E2.bed
PolII_shMed26_1_Veh.bed
PolII_shMed26_2_E2.bed
PolII_shMed26_2_Veh.bed
PolII_shscramble_E2.bed
PolII_shscramble_Veh.bed
PolII_siCbx3_DHT_LNCAP.bed
PolII_siCbx3_E2_MCF7.bed
PolII_siCbx3_Veh_MCF7.bed
PolII_siNC_DHT_LNCAP.bed
PolII_siNC_E2_MCF7.bed
PolII_siNC_Veh_LNCAP.bed
PolII_siNC_Veh_MCF7.bed
Rad21_E2.bed
Rad21_Veh.bed
Smc1_E2.bed
Smc1_Veh.bed
AP2y_E2.ucsc.bigWig
AP2y_Veh.ucsc.bigWig
BRD4_E2_exp1.ucsc.bigWig
BRD4_E2_exp2.ucsc.bigWig
BRD4_Veh_exp1.ucsc.bigWig
BRD4_Veh_exp2.ucsc.bigWig
CBP_E2.ucsc.bigWig
CBP_Veh.ucsc.bigWig
CBX3_siNC_E2.ucsc.bigWig
CBX3_siNC_Veh.ucsc.bigWig
CBX3_siTop1_E2.ucsc.bigWig
CBX3_siTop1_Veh.ucsc.bigWig
CTCF_E2.ucsc.bigWig
CTCF_Veh.ucsc.bigWig
ERa_E2.ucsc.bigWig
ERa_Veh.ucsc.bigWig
FoxA1_E2.ucsc.bigWig
FoxA1_Veh.ucsc.bigWig
Gata3_E2.ucsc.bigWig
Gata3_Veh.ucsc.bigWig
H3K27Ac_E2.ucsc.bigWig
H3K27Ac_Veh.ucsc.bigWig
H3K4me3_E2.ucsc.bigWig

H3K4me3_Veh.ucsc.bigWig
H3K9me3_E2.ucsc.bigWig
H3K9me3_Veh.ucsc.bigWig
MED1_siCbx3_E2.ucsc.bigWig
MED1_siCbx3_Veh.ucsc.bigWig
MED1_siNC_E2.ucsc.bigWig
MED1_siNC_Veh.ucsc.bigWig
MED26_siCbx3_E2.ucsc.bigWig
MED26_siCbx3_Veh.ucsc.bigWig
MED26_siNC_E2.ucsc.bigWig
MED26_siNC_Veh.ucsc.bigWig
NCAPG_E2.ucsc.bigWig
NCAPG_Veh.ucsc.bigWig
PolIISer2P_shMed26_1_E2.ucsc.bigWig
PolIISer2P_shMed26_1_Veh.ucsc.bigWig
PolIISer2P_shMed26_2_E2.ucsc.bigWig
PolIISer2P_shMed26_2_Veh.ucsc.bigWig
PolIISer2P_shscramble_E2.ucsc.bigWig
PolIISer2P_shscramble_Veh.ucsc.bigWig
PolII_shMed26_1_E2.ucsc.bigWig
PolII_shMed26_1_Veh.ucsc.bigWig
PolII_shMed26_2_E2.ucsc.bigWig
PolII_shMed26_2_Veh.ucsc.bigWig
PolII_shscramble_E2.ucsc.bigWig
PolII_shscramble_Veh.ucsc.bigWig
PolII_siCbx3_DHT_LNCAP.ucsc.bigWig
PolII_siCbx3_E2_MCF7.ucsc.bigWig
PolII_siCbx3_Veh_MCF7.ucsc.bigWig
PolII_siNC_DHT_LNCAP.ucsc.bigWig
PolII_siNC_E2_MCF7.ucsc.bigWig
PolII_siNC_Veh_LNCAP.ucsc.bigWig
PolII_siNC_Veh_MCF7.ucsc.bigWig
Rad21_E2.ucsc.bigWig
Rad21_Veh.ucsc.bigWig
Smc1_E2.ucsc.bigWig
Smc1_Veh.ucsc.bigWig
H3K27ac_KCl_0min_R1_001.fastq.gz
H3K27ac_KCl_0min_R2_001.fastq.gz
H3K27ac_KCl_15min_R1_001.fastq.gz
H3K27ac_KCl_15min_R2_001.fastq.gz
H3K27ac_KCl_30min_R1_001.fastq.gz
H3K27ac_KCl_30min_R2_001.fastq.gz
H3K27ac_KCl_60min_R1_001.fastq.gz
H3K27ac_KCl_60min_R2_001.fastq.gz
H3K27me3_KCl_0min_exp1.fastq.gz
H3K27me3_KCl_0min_exp2.fastq.gz
HP1g_KCl_0min_exp2_R1_001.fastq.gz
HP1g_KCl_0min_exp2_R2_001.fastq.gz
HP1g_KCl_0min_R1_001.fastq.gz
HP1g_KCl_0min_R2_001.fastq.gz
HP1g_KCl_15min_R1_001.fastq.gz
HP1g_KCl_15min_R2_001.fastq.gz
HP1g_KCl_30min_exp2_R1_001.fastq.gz
HP1g_KCl_30min_exp2_R2_001.fastq.gz
HP1g_KCl_30min_R1_001.fastq.gz
HP1g_KCl_30min_R2_001.fastq.gz
IgG_KCl_0min_exp1.fastq.gz
IgG_KCl_0min_exp2.fastq.gz
Ku70_KCl_0min_exp2_R1_001.fastq.gz
Ku70_KCl_0min_exp2_R2_001.fastq.gz
Ku70_KCl_0min_R1_001.fastq.gz
Ku70_KCl_0min_R2_001.fastq.gz
Ku70_KCl_15min_R1_001.fastq.gz
Ku70_KCl_15min_R2_001.fastq.gz
Ku70_KCl_30min_exp2_R1_001.fastq.gz
Ku70_KCl_30min_exp2_R2_001.fastq.gz
Ku70_KCl_30min_R1_001.fastq.gz
Ku70_KCl_30min_R2_001.fastq.gz
PolII_KCl_0min_exp2_R1_001.fastq.gz
PolII_KCl_0min_exp2_R2_001.fastq.gz
PolII_KCl_0min_R1_001.fastq.gz
PolII_KCl_0min_R2_001.fastq.gz
PolII_KCl_15min_R1_001.fastq.gz
PolII_KCl_15min_R2_001.fastq.gz
PolII_KCl_180min_exp2_R1_001.fastq.gz
PolII_KCl_180min_exp2_R2_001.fastq.gz

```
PolII_KCl_30min_R1_001.fastq.gz
PolII_KCl_30min_R2_001.fastq.gz
PolII_KCl_60min_R1_001.fastq.gz
PolII_KCl_60min_R2_001.fastq.gz
Topo1cc_KCl_0min_exp2_R1_001.fastq.gz
Topo1cc_KCl_0min_exp2_R2_001.fastq.gz
Topo1cc_KCl_0min_R1_001.fastq.gz
Topo1cc_KCl_0min_R2_001.fastq.gz
Topo1cc_KCl_15min_R1_001.fastq.gz
Topo1cc_KCl_15min_R2_001.fastq.gz
Topo1cc_KCl_30min_exp2_R1_001.fastq.gz
Topo1cc_KCl_30min_exp2_R2_001.fastq.gz
Topo1cc_KCl_30min_R1_001.fastq.gz
Topo1cc_KCl_30min_R2_001.fastq.gz
H3K27me3_Veh_exp3_R1_001.fastq.gz
H3K27me3_Veh_exp3_R2_001.fastq.gz
H3K27me3_Veh_LNCAP_R1_001.fastq.gz
H3K27me3_Veh_LNCAP_R2_001.fastq.gz
HP1g_DHT_60min_LNCAP_exp1_R1_001.fastq.gz
HP1g_DHT_60min_LNCAP_exp1_R2_001.fastq.gz
HP1g_DHT_60min_LNCAP_exp2_R1_001.fastq.gz
HP1g_DHT_60min_LNCAP_exp2_R2_001.fastq.gz
HP1g_E2_14h_exp4_R1_001.fastq.gz
HP1g_E2_14h_exp4_R2_001.fastq.gz
HP1g_E2_1h_exp4_R1_001.fastq.gz
HP1g_E2_1h_exp4_R2_001.fastq.gz
HP1g_E2_1h_siKu70_exp6_R1_001.fastq.gz
HP1g_E2_1h_siKu70_exp6_R2_001.fastq.gz
HP1g_E2_1h_siNC_exp6_R1_001.fastq.gz
HP1g_E2_1h_siNC_exp6_R2_001.fastq.gz
HP1g_TNFa_14h_exp4_R1_001.fastq.gz
HP1g_TNFa_14h_exp4_R2_001.fastq.gz
HP1g_TNFa_1h_exp4_R1_001.fastq.gz
HP1g_TNFa_1h_exp4_R2_001.fastq.gz
HP1g_Veh_exp4_R1_001.fastq.gz
HP1g_Veh_exp4_R2_001.fastq.gz
HP1g_Veh_LNCAP_exp1_R1_001.fastq.gz
HP1g_Veh_LNCAP_exp1_R2_001.fastq.gz
HP1g_Veh_LNCAP_exp2_R1_001.fastq.gz
HP1g_Veh_LNCAP_exp2_R2_001.fastq.gz
HP1g_Veh_siKu70_exp6_R1_001.fastq.gz
HP1g_Veh_siKu70_exp6_R2_001.fastq.gz
HP1g_Veh_siNC_exp6_R1_001.fastq.gz
HP1g_Veh_siNC_exp6_R2_001.fastq.gz
IgG_Veh_exp3_R1_001.fastq.gz
IgG_Veh_exp3_R2_001.fastq.gz
IgG_Veh_exp4_R1_001.fastq.gz
IgG_Veh_exp4_R2_001.fastq.gz
IgG_Veh_LNCAP_R1_001.fastq.gz
IgG_Veh_LNCAP_R2_001.fastq.gz
Ku70_DHT_60min_LNCAP_exp1_R1_001.fastq.gz
Ku70_DHT_60min_LNCAP_exp1_R2_001.fastq.gz
Ku70_DHT_60min_LNCAP_exp2_R1_001.fastq.gz
Ku70_DHT_60min_LNCAP_exp2_R2_001.fastq.gz
Ku70_E2_14h_exp2_R1_001.fastq.gz
Ku70_E2_14h_exp2_R2_001.fastq.gz
Ku70_TNFa_14h_exp2_R1_001.fastq.gz
Ku70_TNFa_14h_exp2_R2_001.fastq.gz
Ku70_TNFa_1h_exp2_R1_001.fastq.gz
Ku70_TNFa_1h_exp2_R2_001.fastq.gz
Ku70_Veh_exp2_R1_001.fastq.gz
Ku70_Veh_exp2_R2_001.fastq.gz
Ku70_Veh_LNCAP_exp1_R1_001.fastq.gz
Ku70_Veh_LNCAP_exp1_R2_001.fastq.gz
Ku70_Veh_LNCAP_exp2_R1_001.fastq.gz
Ku70_Veh_LNCAP_exp2_R2_001.fastq.gz
Topo1cc_DHT_60min_LNCAP_exp1_R1_001.fastq.gz
Topo1cc_DHT_60min_LNCAP_exp1_R2_001.fastq.gz
Topo1cc_DHT_60min_LNCAP_exp2_R1_001.fastq.gz
Topo1cc_DHT_60min_LNCAP_exp2_R2_001.fastq.gz
Topo1cc_E2_14h_exp4_R1_001.fastq.gz
Topo1cc_E2_14h_exp4_R2_001.fastq.gz
Topo1cc_Veh_exp4_R1_001.fastq.gz
Topo1cc_Veh_exp4_R2_001.fastq.gz
Topo1cc_Veh_LNCAP_exp1_R1_001.fastq.gz
Topo1cc_Veh_LNCAP_exp1_R2_001.fastq.gz
```

```
Topo1cc_Veh_LNCAP_exp2_R1_001.fastq.gz
Topo1cc_Veh_LNCAP_exp2_R2_001.fastq.gz
Topo1_E2_14h_exp3_R1_001.fastq.gz
Topo1_E2_14h_exp3_R2_001.fastq.gz
Topo1_E2_1h_exp3_R1_001.fastq.gz
Topo1_E2_1h_exp3_R2_001.fastq.gz
Topo1_Veh_exp3_R1_001.fastq.gz
Topo1_Veh_exp3_R2_001.fastq.gz
Gata3_E2_1h_siHP1g_exp2.fastq.gz
Gata3_E2_1h_siNC_exp2.fastq.gz
Gata3_Veh_siHP1g_exp2.fastq.gz
Gata3_Veh_siNC_exp2.fastq.gz
HP1g_E2_14h_exp2.fastq.gz
HP1g_E2_14h_exp3.fastq.gz
HP1g_E2_1h_exp2.fastq.gz
HP1g_E2_1h_exp3.fastq.gz
HP1g_E2_1h_siKu70_exp5.fastq.gz
HP1g_E2_1h_siNC_exp5.fastq.gz
HP1gS83A_E2_1h_exp1.fastq.gz
HP1gS83A_E2_1h_exp2.fastq.gz
HP1gS83D_E2_1h_exp1.fastq.gz
HP1gS83D_E2_1h_exp2.fastq.gz
HP1g_Veh_exp2.fastq.gz
HP1g_Veh_exp3.fastq.gz
HP1g_Veh_siKu70_exp5.fastq.gz
HP1g_Veh_siNC_exp5.fastq.gz
Ku70_E2_1h_exp1.fastq.gz
Ku70_E2_1h_exp3.fastq.gz
Ku70_Veh_exp1.fastq.gz
Ku70_Veh_exp3.fastq.gz
Ku80_E2_1h_exp3.fastq.gz
Ku80_Veh_exp3.fastq.gz
Topo1cc_E2_1h_exp1.fastq.gz
Topo1cc_E2_1h_exp2.fastq.gz
Topo1cc_Veh_exp1.fastq.gz
Topo1cc_Veh_exp2.fastq.gz
H3K27ac_KCl_0min.ucsc.bedGraph.gz
H3K27ac_KCl_15min.ucsc.bedGraph.gz
H3K27ac_KCl_30min.ucsc.bedGraph.gz
H3K27ac_KCl_60min.ucsc.bedGraph.gz
H3K27me3_KCl_0min_exp1.ucsc.bedGraph.gz
H3K27me3_KCl_0min_exp2.ucsc.bedGraph.gz
HP1g_KCl_0min_exp2.ucsc.bedGraph.gz
HP1g_KCl_0min.ucsc.bedGraph.gz
HP1g_KCl_15min.ucsc.bedGraph.gz
HP1g_KCl_30min_exp2.ucsc.bedGraph.gz
HP1g_KCl_30min.ucsc.bedGraph.gz
IgG_KCl_0min_exp1.ucsc.bedGraph.gz
IgG_KCl_0min_exp2.ucsc.bedGraph.gz
Ku70_KCl_0min_exp2.ucsc.bedGraph.gz
Ku70_KCl_0min.ucsc.bedGraph.gz
Ku70_KCl_15min.ucsc.bedGraph.gz
Ku70_KCl_30min_exp2.ucsc.bedGraph.gz
Ku70_KCl_30min.ucsc.bedGraph.gz
PolII_KCl_0min_exp2.ucsc.bedGraph.gz
PolII_KCl_0min.ucsc.bedGraph.gz
PolII_KCl_15min.ucsc.bedGraph.gz
PolII_KCl_180min_exp2.ucsc.bedGraph.gz
PolII_KCl_30min.ucsc.bedGraph.gz
PolII_KCl_60min.ucsc.bedGraph.gz
Topo1cc_KCl_0min_exp2.ucsc.bedGraph.gz
Topo1cc_KCl_0min.ucsc.bedGraph.gz
Topo1cc_KCl_15min.ucsc.bedGraph.gz
Topo1cc_KCl_30min_exp2.ucsc.bedGraph.gz
Topo1cc_KCl_30min.ucsc.bedGraph.gz
H3K27ac_KCl_0min.bed
H3K27ac_KCl_15min.bed
H3K27ac_KCl_30min.bed
H3K27ac_KCl_60min.bed
H3K27me3_KCl_0min_exp1.bed
H3K27me3_KCl_0min_exp2.bed
HP1g_KCl_0min.bed
HP1g_KCl_0min_exp2.bed
HP1g_KCl_15min.bed
HP1g_KCl_30min.bed
HP1g_KCl_30min_exp2.bed
```

```
IgG_KCl_0min_exp1.bed
IgG_KCl_0min_exp2.bed
Ku70_KCl_0min.bed
Ku70_KCl_0min_exp2.bed
Ku70_KCl_15min.bed
Ku70_KCl_30min.bed
Ku70_KCl_30min_exp2.bed
PolII_KCl_0min.bed
PolII_KCl_0min_exp2.bed
PolII_KCl_15min.bed
PolII_KCl_180min_exp2.bed
PolII_KCl_30min.bed
PolII_KCl_60min.bed
Topo1cc_KCl_0min.bed
Topo1cc_KCl_0min_exp2.bed
Topo1cc_KCl_15min.bed
Topo1cc_KCl_30min.bed
Topo1cc_KCl_30min_exp2.bed
Gata3_E2_1h_siHP1g_exp2.bed
Gata3_E2_1h_siHp1g_exp2.fastq.gz
Gata3_E2_1h_siHp1g_exp2.ucsc.bedGraph.gz
Gata3_E2_1h_siNC_exp2.bed
Gata3_E2_1h_siNC_exp2.fastq.gz
Gata3_E2_1h_siNC_exp2.ucsc.bedGraph.gz
Gata3_Veh_siHP1g_exp2.bed
Gata3_Veh_siHP1g_exp2.fastq.gz
Gata3_Veh_siHP1g_exp2.ucsc.bedGraph.gz
Gata3_Veh_siNC_exp2.bed
Gata3_Veh_siNC_exp2.fastq.gz
Gata3_Veh_siNC_exp2.ucsc.bedGraph.gz
HP1g_E2_14h_exp2.bed
HP1g_E2_14h_exp2.fastq.gz
HP1g_E2_14h_exp2.ucsc.bedGraph.gz
HP1g_E2_14h_exp3.bed
HP1g_E2_14h_exp3.fastq.gz
HP1g_E2_14h_exp3.ucsc.bedGraph.gz
HP1g_E2_1h_exp2.bed
HP1g_E2_1h_exp2.fastq.gz
HP1g_E2_1h_exp2.ucsc.bedGraph.gz
HP1g_E2_1h_exp3.bed
HP1g_E2_1h_exp3.fastq.gz
HP1g_E2_1h_exp3.ucsc.bedGraph.gz
HP1g_E2_1h_siKu70_exp5.bed
HP1g_E2_1h_siKu70_exp5.fastq.gz
HP1g_E2_1h_siKu70_exp5.ucsc.bedGraph.gz
HP1g_E2_1h_siNC_exp5.bed
HP1g_E2_1h_siNC_exp5.fastq.gz
HP1g_E2_1h_siNC_exp5.ucsc.bedGraph.gz
HP1gS83A_E2_1h_exp1.bed
HP1gS83A_E2_1h_exp1.fastq.gz
HP1gS83A_E2_1h_exp1.ucsc.bedGraph.gz
HP1gS83A_E2_1h_exp2.bed
HP1gS83A_E2_1h_exp2.fastq.gz
HP1gS83A_E2_1h_exp2.ucsc.bedGraph.gz
HP1gS83D_E2_1h_exp1.bed
HP1gS83D_E2_1h_exp1.fastq.gz
HP1gS83D_E2_1h_exp1.ucsc.bedGraph.gz
HP1gS83D_E2_1h_exp2.bed
HP1gS83D_E2_1h_exp2.fastq.gz
HP1gS83D_E2_1h_exp2.ucsc.bedGraph.gz
HP1g_Veh_exp2.bed
HP1g_Veh_exp2.fastq.gz
HP1g_Veh_exp2.ucsc.bedGraph.gz
HP1g_Veh_exp3.bed
HP1g_Veh_exp3.fastq.gz
HP1g_Veh_exp3.ucsc.bedGraph.gz
HP1g_Veh_siKu70_exp5.bed
HP1g_Veh_siKu70_exp5.fastq.gz
HP1g_Veh_siKu70_exp5.ucsc.bedGraph.gz
HP1g_Veh_siNC_exp5.bed
HP1g_Veh_siNC_exp5.fastq.gz
HP1g_Veh_siNC_exp5.ucsc.bedGraph.gz
Ku70_E2_1h_exp1.bed
Ku70_E2_1h_exp1.fastq.gz
Ku70_E2_1h_exp1.ucsc.bedGraph.gz
Ku70_E2_1h_exp3.bed
```

```
Ku70_E2_1h_exp3.fastq.gz
Ku70_E2_1h_exp3.ucsc.bedGraph.gz
Ku70_Veh_exp1.bed
Ku70_Veh_exp1.fastq.gz
Ku70_Veh_exp1.ucsc.bedGraph.gz
Ku70_Veh_exp3.bed
Ku70_Veh_exp3.fastq.gz
Ku70_Veh_exp3.ucsc.bedGraph.gz
Ku80_E2_1h_exp3.bed
Ku80_E2_1h_exp3.fastq.gz
Ku80_E2_1h_exp3.ucsc.bedGraph.gz
Ku80_Veh_exp3.bed
Ku80_Veh_exp3.fastq.gz
Ku80_Veh_exp3.ucsc.bedGraph.gz
Topo1cc_E2_1h_exp1.bed
Topo1cc_E2_1h_exp1.fastq.gz
Topo1cc_E2_1h_exp1.ucsc.bedGraph.gz
Topo1cc_E2_1h_exp2.bed
Topo1cc_E2_1h_exp2.fastq.gz
Topo1cc_E2_1h_exp2.ucsc.bedGraph.gz
Topo1cc_Veh_exp1.bed
Topo1cc_Veh_exp1.fastq.gz
Topo1cc_Veh_exp1.ucsc.bedGraph.gz
Topo1cc_Veh_exp2.bed
Topo1cc_Veh_exp2.fastq.gz
Topo1cc_Veh_exp2.ucsc.bedGraph.gz
H3K27me3_Veh_exp3.bed
H3K27me3_Veh_exp3_R1_001.fastq.gz
H3K27me3_Veh_exp3_R2_001.fastq.gz
H3K27me3_Veh_exp3.ucsc.bedGraph.gz
H3K27me3_Veh_LNCAP.bed
H3K27me3_Veh_LNCAP_R1_001.fastq.gz
H3K27me3_Veh_LNCAP_R2_001.fastq.gz
H3K27me3_Veh_LNCAP.ucsc.bedGraph.gz
HP1g_DHT_60min_LNCAP_exp1.bed
HP1g_DHT_60min_LNCAP_exp1_R1_001.fastq.gz
HP1g_DHT_60min_LNCAP_exp1_R2_001.fastq.gz
HP1g_DHT_60min_LNCAP_exp1.ucsc.bedGraph.gz
HP1g_DHT_60min_LNCAP_exp2.bed
HP1g_DHT_60min_LNCAP_exp2_R1_001.fastq.gz
HP1g_DHT_60min_LNCAP_exp2_R2_001.fastq.gz
HP1g_DHT_60min_LNCAP_exp2.ucsc.bedGraph.gz
HP1g_E2_14h_exp4.bed
HP1g_E2_14h_exp4_R1_001.fastq.gz
HP1g_E2_14h_exp4_R2_001.fastq.gz
HP1g_E2_14h_exp4.ucsc.bedGraph.gz
HP1g_E2_1h_exp4.bed
HP1g_E2_1h_exp4_R1_001.fastq.gz
HP1g_E2_1h_exp4_R2_001.fastq.gz
HP1g_E2_1h_exp4.ucsc.bedGraph.gz
HP1g_E2_1h_siKu70_exp6.bed
HP1g_E2_1h_siKu70_exp6_R1_001.fastq.gz
HP1g_E2_1h_siKu70_exp6_R2_001.fastq.gz
HP1g_E2_1h_siKu70_exp6.ucsc.bedGraph.gz
HP1g_E2_1h_siNC_exp6.bed
HP1g_E2_1h_siNC_exp6_R1_001.fastq.gz
HP1g_E2_1h_siNC_exp6_R2_001.fastq.gz
HP1g_E2_1h_siNC_exp6.ucsc.bedGraph.gz
HP1g_TNFa_14h_exp4.bed
HP1g_TNFa_14h_exp4_R1_001.fastq.gz
HP1g_TNFa_14h_exp4_R2_001.fastq.gz
HP1g_TNFa_1h_exp4.bed
HP1g_TNFa_1h_exp4_R1_001.fastq.gz
HP1g_TNFa_1h_exp4_R2_001.fastq.gz
HP1g_TNFa_1h_exp4.ucsc.bedGraph.gz
HP1g_Veh_exp4.bed
HP1g_Veh_exp4_R1_001.fastq.gz
HP1g_Veh_exp4_R2_001.fastq.gz
HP1g_Veh_exp4.ucsc.bedGraph.gz
HP1g_Veh_LNCAP_exp1.bed
HP1g_Veh_LNCAP_exp1_R1_001.fastq.gz
HP1g_Veh_LNCAP_exp1_R2_001.fastq.gz
HP1g_Veh_LNCAP_exp1.ucsc.bedGraph.gz
HP1g_Veh_LNCAP_exp2.bed
HP1g_Veh_LNCAP_exp2_R1_001.fastq.gz
HP1g_Veh_LNCAP_exp2_R2_001.fastq.gz
```

```
HP1g_Veh_LNCAP_exp2.ucsc.bedGraph.gz
HP1g_Veh_siKu70_exp6.bed
HP1g_Veh_siKu70_exp6_R1_001.fastq.gz
HP1g_Veh_siKu70_exp6_R2_001.fastq.gz
HP1g_Veh_siKu70_exp6.ucsc.bedGraph.gz
HP1g_Veh_siNC_exp6.bed
HP1g_Veh_siNC_exp6_R1_001.fastq.gz
HP1g_Veh_siNC_exp6_R2_001.fastq.gz
HP1g_Veh_siNC_exp6.ucsc.bedGraph.gz
IgG_Veh_exp4.bed
IgG_Veh_exp4_R1_001.fastq.gz
IgG_Veh_exp4_R2_001.fastq.gz
IgG_Veh_exp4.ucsc.bedGraph.gz
IgG_Veh_LNCAP.bed
IgG_Veh_LNCAP_R1_001.fastq.gz
IgG_Veh_LNCAP_R2_001.fastq.gz
IgG_Veh_LNCAP.ucsc.bedGraph.gz
Ku70_DHT_60min_LNCAP_exp1.bed
Ku70_DHT_60min_LNCAP_exp1_R1_001.fastq.gz
Ku70_DHT_60min_LNCAP_exp1_R2_001.fastq.gz
Ku70_DHT_60min_LNCAP_exp1.ucsc.bedGraph.gz
Ku70_DHT_60min_LNCAP_exp2.bed
Ku70_DHT_60min_LNCAP_exp2_R1_001.fastq.gz
Ku70_DHT_60min_LNCAP_exp2_R2_001.fastq.gz
Ku70_DHT_60min_LNCAP_exp2.ucsc.bedGraph.gz
Ku70_E2_14h_exp2.bed
Ku70_E2_14h_exp2_R1_001.fastq.gz
Ku70_E2_14h_exp2_R2_001.fastq.gz
Ku70_E2_14h_exp2.ucsc.bedGraph.gz
Ku70_TNFa_14h_exp2.bed
Ku70_TNFa_14h_exp2_R1_001.fastq.gz
Ku70_TNFa_14h_exp2_R2_001.fastq.gz
Ku70_TNFa_14h_exp2.ucsc.bedGraph.gz
Ku70_TNFa_1h_exp2.bed
Ku70_TNFa_1h_exp2_R1_001.fastq.gz
Ku70_TNFa_1h_exp2_R2_001.fastq.gz
Ku70_TNFa_1h_exp2.ucsc.bedGraph.gz
Ku70_Veh_exp2.bed
Ku70_Veh_exp2_R1_001.fastq.gz
Ku70_Veh_exp2_R2_001.fastq.gz
Ku70_Veh_exp2.ucsc.bedGraph.gz
Ku70_Veh_LNCAP_exp1.bed
Ku70_Veh_LNCAP_exp1_R1_001.fastq.gz
Ku70_Veh_LNCAP_exp1_R2_001.fastq.gz
Ku70_Veh_LNCAP_exp1.ucsc.bedGraph.gz
Ku70_Veh_LNCAP_exp2.bed
Ku70_Veh_LNCAP_exp2_R1_001.fastq.gz
Ku70_Veh_LNCAP_exp2_R2_001.fastq.gz
Ku70_Veh_LNCAP_exp2.ucsc.bedGraph.gz
Topo1cc_DHT_60min_LNCAP_exp1.bed
Topo1cc_DHT_60min_LNCAP_exp1_R1_001.fastq.gz
Topo1cc_DHT_60min_LNCAP_exp1_R2_001.fastq.gz
Topo1cc_DHT_60min_LNCAP_exp1.ucsc.bedGraph.gz
Topo1cc_DHT_60min_LNCAP_exp2.bed
Topo1cc_DHT_60min_LNCAP_exp2_R1_001.fastq.gz
Topo1cc_DHT_60min_LNCAP_exp2_R2_001.fastq.gz
Topo1cc_DHT_60min_LNCAP_exp2.ucsc.bedGraph.gz
Topo1cc_E2_14h_exp4.bed
Topo1cc_E2_14h_exp4_R1_001.fastq.gz
Topo1cc_E2_14h_exp4_R2_001.fastq.gz
Topo1cc_E2_14h_exp4.ucsc.bedGraph.gz
Topo1cc_Veh_exp4.bed
Topo1cc_Veh_exp4_R1_001.fastq.gz
Topo1cc_Veh_exp4_R2_001.fastq.gz
Topo1cc_Veh_exp4.ucsc.bedGraph.gz
Topo1cc_Veh_LNCAP_exp1.bed
Topo1cc_Veh_LNCAP_exp1_R1_001.fastq.gz
Topo1cc_Veh_LNCAP_exp1_R2_001.fastq.gz
Topo1cc_Veh_LNCAP_exp1.ucsc.bedGraph.gz
Topo1cc_Veh_LNCAP_exp2.bed
Topo1cc_Veh_LNCAP_exp2_R1_001.fastq.gz
Topo1cc_Veh_LNCAP_exp2_R2_001.fastq.gz
Topo1cc_Veh_LNCAP_exp2.ucsc.bedGraph.gz
Topo1_E2_14h_exp3.bed
Topo1_E2_14h_exp3_R1_001.fastq.gz
Topo1_E2_14h_exp3_R2_001.fastq.gz
```

Topo1_E2_14h_exp3.ucsc.bedGraph.gz
Topo1_E2_1h_exp3.bed
Topo1_E2_1h_exp3_R1_001.fastq.gz
Topo1_E2_1h_exp3_R2_001.fastq.gz
Topo1_E2_1h_exp3.ucsc.bedGraph.gz
Topo1_Veh_exp3.bed
Topo1_Veh_exp3_R1_001.fastq.gz
Topo1_Veh_exp3_R2_001.fastq.gz
Topo1_Veh_exp3.ucsc.bedGraph.gz

**Genome browser session**
(e.g. UCSC)

For human cancer cell data:
http://genome.ucsc.edu/s/yuliangtan/Topo1cc_hg38

For cortical neuron data:
http://genome.ucsc.edu/s/yuliangtan/Topo1cc_mm10

## Methodology

**Replicates**

All experiments were replicated at least twice.

**Sequencing depth**

At least 30, 000,000 reads were detected at most of the sequencing data.

**Antibodies**

ERα (HC20) Santa Cruz Biotechnology sc-543 (Lot# I0514)
RNA Pol II (N20) Santa Cruz Biotechnology sc-899 (Lot# D2315)
AP2γ(H-77) Santa Cruz Biotechnology sc-8977 (Lot#G1112)
GATA3(HG3-31) Santa Cruz Biotechnology sc-268 (Lot#J0515)
CBP diagenode C15410224 (Lot#39721)
Rad21 Abcam ab992 (Lot# GR214359-8)
H3K27Ac Abcam ab4729 (Lot#GR288020-1)
Pol II diagenode C15200004 (Lot#001-11)
FoxA1 diagenode C15410231 (Lot#39435)
BRD4 diagenode C15410337 (Lot#A2710P)
SMC1 Bethyl Laboratories A300-055A (A302-055A-6)
H3K4me2 Abcam ab7766 (Lot#GR102810-4)
H3K9me3 Abcam ab8898(Lot#GR3217826-1)
PolIISer2p Abcam ab5095(Lot#GR3225147-1)
H3K9me2 Cell Signaling 9753S(Lot# 4)
MED26 (13641S, Cell Signaling)
TOP1 Bethyl A302-589A (Lot#A302-589A-1)
TOP1cc TopoGEN TG2017-2 (Lot# 17AG15)
Ku-80 MyBioSource MBS8533127 (Lot#T14S11)
Anti-HP1g, clone 42s2 Millipore 05-690 (Lot#3224566)
Ku70 Bethyl A302-624A (Lot#A302-624A-1)
H3K4me3 Abcam Ab8580(Lot#GR3201182-1)
CTCF Active Motif 61311(Lot#34614003)

**Peak calling parameters**

For TFs: findPeaks -style factor -o auto
For histones: findPeaks -style histone -o auto

**Data quality**

# maximum tags considered per bp = 1.0
# effective number of tags used for normalization = 10000000.0
# Peaks have been centered at maximum tag pile-up
# FDR rate threshold = 0.001000000
# FDR tag threshold = 20.0
# size of region used for local filtering = 10000
# Fold over local region required = 4.00
# Poisson p-value over local region required = 1.00e-04
# Maximum fold under expected unique positions for tags = 2.00
# Putative peaks filtered for being too clonal = 0

**Software**

DNA sequences generated by the Illumina Pipeline were aligned to the human genome (hg38) or mouse genome (mm10) assembly using Bowtie2. The data were visualized by preparing custom tracks on the University of California, Santa Cruz (UCSC) genome browser using HOMER software package.
Clustering plots for normalized tag densities at each genomic region were generated using HOMER and then clustered using Gene Cluster 3.0 and visualized using Java TreeView.
Sequence logos were generated using WebLOGO. Gene ontology analysis was performed with Metascape.
The overlaps between sites identified in ChIP-seq for DNA-binding proteins and TOP1cc signals were calculated using BEDTools.

