## [Peer Review File · Nature Structural & Molecular Biology]

Peer Review Information

Manuscript Title: Signal-induced Enhancer Activation Requires Ku70 to Read Topoisomerase1-DNA Covalent Complexes

Corresponding author name(s): Michael Rosenfeld, Yuliang Tan

Reviewer Comments & Decisions:

Decision Letter, initial version:
--

Mess 13th Jan 2020

age:

Dear Dr. Rosenfeld,

Thank you again for submission of your manuscript "TOP1cc/HP1γ/Med26 Axis Is A Predictive Signature for Enhancers Exhibiting Phase Separation-like Properties". My apologies for the delay in reaching a decision; we are still working on a slight backlog due to the recent holidays. Nevertheless, we have now carefully evaluated the work and discussed it among the full editorial team. Unfortunately, we have decided not to consider the manuscript further for publication in Nature Structural & Molecular Biology.

We can only consider a small proportion of the manuscripts submitted to our journal and are often forced to make difficult decisions. Manuscripts are evaluated editorially for their potential interest to a broad audience, the level of novel insight obtained and whether the findings represent a significant advance relative to the published literature, among other considerations.

In this case, we appreciate the demonstration that TOP1 nicking at acutely activated enhancers promotes HP1γ/Med26-mediated transcription and is required for enhancer activation. We also recognize that these observations and the potential use of the TOP1cc/HP1γ/Med26 signalling axis as a predictive signature for robust active enhancers with dynamic phase separation-like properties will be compelling to researchers working in this field. However, after discussion among the editorial staff, we are concerned about the level of mechanistic insights into how TOP1-nicking promotes HP1γ/Med26-mediated transcriptional activation at these enhancers, which, in our view, limits the interest to a wider audience. In addition, we felt that the predictive properties of the TOP1cc/HP1γ/Med26 signalling axis would require confirmation in additional cellular systems, also because the experimental system used seems to be somewhat unusual given the poor correlation between H3K9me3 and HP1γ ChIP-

seq signals. These concerns are sufficient to prevent us from considering the manuscript further for publication in Nature Structural & Molecular Biology.

You might, however, want to consider our sister journal [Nature Communications](https://www.nature.com/ncomms/about) as a potential venue for the publication of these results. *Nature Communications* publishes high quality and influential research and across the full spectrum of the natural sciences. More information on the journal, the potential benefits of transfer and a link to transfer your paper, can be found at the bottom of this email. Please note that the editorial team at *Nature Communications* will consider your manuscript independently of our suggestion to transfer.

I am sorry we could not be more positive on this occasion. We thank you for the opportunity to consider this work and wish you success in seeking publication elsewhere.

With kind regards,
Anke

Anke Sparmann, PhD
Senior Editor
Nature Structural and Molecular Biology
ORCID 0000-0001-7695-2049

** *Nature Communications* is the Nature Research flagship Open Access journal. If you would like this work to be considered for publication there, you can easily transfer the manuscript by following the instructions below. It is not necessary to reformat your paper. Once all files are received, the editors at *Nature Communications* will assess your manuscript's suitability for potential publication; they aim to provide feedback quickly, with a median decision time of 8 days for first editorial decisions on suitability. The journal is also proud to offer double blind and transparent peer review options. For 2017, the 2-year impact factor for *Nature Communications* is 12.353 and the 2-year median is 8 (for further information on journal metrics, please visit our [Nature journals metrics page](http://www.nature.com/npg_/company_info/journal_metrics.html)). Our [open access pages](http://www.nature.com/ncomms/open_access/index.html) contain information about article processing charges, open access funding, and advice and support from Springer Nature.

** If you wish to transfer your manuscript to Nature Communications, please use our manuscript transfer portal to initiate the transfer to this journal (or to another journal of your choice in the Nature Research portfolio). If you transfer to Nature-branded journals or to the Communications journals, you will not have to re-supply manuscript metadata and files. This link can only be used once and remains active until used.

All Nature Research journals are editorially independent, and the decision to consider your manuscript will be taken by their own editorial staff. For more information, please see our [manuscript transfer FAQ](http://www.nature.com/authors/author_resources/transfer_manuscripts.html?WT.mc_id=EMI_NPG_1511_AUTHORTRANSF&WT.ec_id=AUTHOR) page.

** For Springer Nature Limited general information and news for authors, see <http://npg.nature.com/authors>

Author Appeal Letter

Dear Dr. Sparmann,

Thank you for your prompt and thoughtful response, which was both informative and helpful. We have discussed your points and concur that modifications based on your suggestions would greatly improve the suitability of the manuscript for *Nature Structural & Molecular Biology* (NSMB). We appreciate your concern with respect to the manuscript as initially presented, but we firmly believe that the novelty of our methodology and data will be of particular interest to a wider audience.

In our manuscript, we present a new, broadly applicable method, topoisomerase I (TOP1)-DNA covalent complexes-sequencing (TOP1cc-seq) that enables us for the first time to disclose TOP1-DNA transient interactions/intermediates genome-wide. Importantly, TOP1-DNA transient intermediates are limited to regulatory elements, predominantly enhancers, representing ~90% of all genomic sites. This in itself will be of major interest to the readers of NSMB, as our understanding of DNA-protein transient intermediates in the mammalian genome (from proteins as diverse as topoisomerases, Poly [ADP-ribose] polymerase, Ku, DNA glycosylases, DNA polymerases and DNA methyltransferases, for example) is rather rudimentary due to lack of available strategies to uncover their natural and frequent occurrences in cellular metabolic processes (PMID: 28655905). Moreover, deciphering the molecular principles underlying the DNA-protein covalent interactions is essential because trapping the DNA-protein transient intermediates are still serving as the first-line treatment regimens against human diseases including cancers, neurodegeneration diseases and autoimmune syndromes (PMID: 20534341).

Indeed, our new methodology led us to the rather unexpected result that TOP1-DNA transient intermediates were essential for the robust activation of enhancers acutely activated in response to ligand. Most surprisingly, this occurred through the recruitment of HP1 γ -a protein that one may have least unexpected on the strongest, acutely activated ligand-dependent enhancers. We show that this recruitment is not dependent on the known property of HP1 γ as a reader of H3K9me3, a mark not present on these enhancers, but rather (as found in the open-ended re-CLIP experiments) based on interactions with TOP1cc *per se*. Indeed, the question you asked as to why we did not observe the H3K9me3 mark on enhancers to which HP1 γ is bound is exactly the reason why this discovery is so timely and novel and will come as a surprise to the readers of NSMB. That is, although HP1 γ is recruited to heterochromatin as we show now in our revised manuscript for its silencing function through its well known H3K9me3 reader function, for the activation of ligand-activated enhancers it is instead recruited via interactions with TOP1cc

(present as a peak, without the spreading characteristic of heterochromatin). No one has suspected the ability of TOPcc to recruit HP1 γ to euchromatic regions in this transcriptional-dependent way –a result that will be of broad interest to the readers of NSMB.

Here, to understand how the recruitment of HP1 γ in this TOP1-dependent fashion activates transcription, we show that it licenses the recruitment of Med26, a component of the Mediator complex, for Pol II elongation. Moreover, these events occur on acutely activated enhancers that possess dynamic liquid-like condensates, but not on chronically activated ligand/signal stimulation. Again, this result will be of major interest to the readers of NSMB because it provides a mechanistic explanation for the strategy underlying acute signal-dependent activation of enhancers. Indeed, as a further test of this idea we used a second transcription factor, NF κ B-dependent enhancer activation, where a putative Megatrans complex has not yet been fully identified, to show that this new TOP1-dependent strategy applies to enhancer activation in different signaling pathways. We also further showed that the androgen receptor induced activation of enhancers in human prostate cancer LNCAP cells is marked by TOP1cc. Thus, we believe that these observations are important not only for a deeper understanding of transcription regulation but also potential utilization in the identifying the acutely activated enhancers which are the key regulators in multiple biological processes such as development, cell lineage determination, homeostasis and cellular responses to stimuli.

Finally, in light of your cogent comments, we have revised the manuscript (attached) accordingly to both emphasize the important, novel contributions, and to avoid any overstatement. We firmly believe that our study provides a framework to better dissect the underlying molecular mechanisms of activation of ligand-dependent acutely activated enhancers, and it also provides a further functional linkage between DNA-protein transient interactions/intermediates and transcriptional activation in mammalian cells. We hope that, with these modifications you will share our opinion regarding the broad significance, impact and novelty of this paper, and that you will find it acceptable for presentation in *Nature Structural & Molecular Biology*.

With best regards,

Geoff Rosenfeld

Decision Letter, Appeal:

Message: 7th Feb 2020

Dear Geoff,

Thank you again for your letter concerning your manuscript "TOP1cc/HP1 γ /Med26 Axis Is A Predictive Signature for Enhancers Exhibiting Phase Separation-like Properties" and the

revised manuscript. We have now had a chance to discuss the points you raised in detail, and we have decided to send your paper out to review.

In order to update the order of the Extended Data Figures, please use the link below to change the manuscript files:

[Redacted]

In addition to the Reporting Summary, which you already supplied, we also ask authors to fill out an editorial Policy Checklist, which confirms compliance with our editorial policies, including the declaration of Competing Interests.

This document can be found here:

Editorial Policy Checklist:

<https://www.nature.com/documents/nr-editorial-policy-checklist.pdf>

Please complete this form and upload it as well. Please note that these forms are dynamic 'smart pdfs' and must, therefore, be downloaded and completed in Adobe Reader. We will then flatten them for ease of use by the reviewers. If you would like to reference the guidance text as you complete the template, please access these flattened versions at <http://www.nature.com/authors/policies/availability.html>.

Note that you are not required to revise your paper to include the information provided in the reporting summary. However, all points on the policy checklist must be addressed; please update the manuscript if needed.

Once we receive the final manuscript file and the Policy checklist, we will proceed to send your paper for review. If you have questions or anticipate delays, please let me know as soon as possible.

With kind regards,
Anke

Anke Sparmann, PhD
Senior Editor
Nature Structural and Molecular Biology
ORCID 0000-0001-7695-2049

Decision letter, first revision:

Message: 16th Mar 2020

Dear Geoff,

Thank you again for submitting your manuscript "TOP1cc-seq Reveals HP1 γ Is Required for Acute Ligand-dependent Activation of Enhancers". I apologize for the delay in reaching a decision, which resulted from the difficulty in obtaining timely referee reports. Nevertheless, we now have comments from the three reviewers who evaluated your paper (appended below). In light of those reports, we remain interested in your study and would like to invite you to respond to the comments of the referees, in the form of a revised manuscript.

You will see that while the reviewers find the results interesting, they also raise substantial concerns that will need to be addressed in a major revision. We agree with reviewer #1 and #3 that the emphasis on TOP1cc-seq as a novel technique should be decreased, although the comparisons with TOP1-ChIP-Seq data requested by reviewer #1 and #2 need to be included. Importantly, potential confounding effects from DNA damage caused by CPT treatment need to be ruled out. Finally, I would like to emphasize that additional mechanistic insights into how TOP1, Med26 and HP1 γ cooperate to promote activation of MegaTrans enhancers, as requested by all three reviewers, will be required for further consideration of the study here.

We appreciate that the requested revisions are extensive. Given the time and effort such a revision would entail, we would understand if you prefer to seek publication elsewhere. If you wish to submit a revision, please be sure to address/respond to all concerns of the referees in full in a point-by-point response and highlight all changes in the revised manuscript text file.

We expect to see the revised manuscript within 6 months. If you cannot send it within this time, please let us know. We will be happy to consider your revision as long as nothing similar has been accepted for publication at NSMB or published elsewhere. Should your manuscript be substantially delayed without notifying us in advance and your article is eventually published, the received date would be that of the revised, not the original, version. If you decide to submit the work elsewhere instead, please let us know, so that our process can be closed (otherwise, it would be considered dual submission).

Reporting Summary:

Please note that all key data shown in the main figures as cropped gels or blots should be presented in uncropped form, with molecular weight markers. These data can be aggregated into a single supplementary figure. While these data can be displayed in a

relatively informal style, they must refer back to the relevant figures. These data should be submitted with the last revision, prior to acceptance, but you may want to start putting it together at this point.

We require deposition of coordinates (and, in the case of crystal structures, structure factors) into the Protein Data Bank with the designation of immediate release upon publication (HPUB). Electron microscopy-derived density maps and coordinate data must be deposited in EMDB and released upon publication. Deposition and immediate release of NMR chemical shift assignments are highly encouraged. Deposition of deep sequencing and microarray data is mandatory, and the datasets must be released prior to or upon publication. Proteomics data should be deposited in PRIDE. To avoid delays in publication, dataset accession numbers must be supplied with the final accepted manuscript and appropriate release dates must be indicated at the galley proof stage. Please find the complete NRG policies on data availability at <http://www.nature.com/authors/policies/availability.html>.

[Redacted]

With kind regards,
Anke

Anke Sparmann, PhD
Senior Editor
Nature Structural and Molecular Biology
ORCID 0000-0001-7695-2049

Referee expertise:

Referee #1: Topoisomerases

Referee #2: enhancer activation, eRNAs

Referee #3: transcription

Reviewers' Comments:

Reviewer #1:

Remarks to the Author:

The connection between Topoisomerase 1 (Top1) function and ligand-responsive enhancer activation has been well established by Puc and Rosenfeld in Puc J. et al. Cell 2015. In this paper, Tan and colleagues extend the story by proposing that Top1 activity is important to regulate enhancers induced by acute but not by chronic stimuli. In response to estrogen (E2), enzymatically active Top1 (Top1cc) is detected at E2-responsive enhancers and required for the transcription of eRNA. The Top1cc mediates the tethering of heterochromatin protein 1 gamma (HP1g) to the enhancers and this, in turn, favours recruitment of Med26 and Pol2. Because the Top1cc/HP1g/Med26 complexes are preferentially enriched at Mega Trans enhancers, which are sensitive to 1,6 HD (shown previously by the same group, Nair S. et al. NSMB 2019), the authors suggest a mechanistic link between Top1 activity and formation of liquid-like condensates to achieve enhancers activation.

The result presented by Tan and colleagues are interesting and could have impact on our understanding of Top1 function during enhancer activation but not conceptually new. Unfortunately, in the current form the work is not appropriate for publication in NSMB. My reservations and specific points of critique are outlined below.

Major comments:

- The first concern I have in this paper is with the Top1cc-seq protocol. The authors describe it as a novel approach "enabling for the first time to disclose the TOP1-DNA transient intermediates genome-wide". Except for the antibody used – anti-Top1cc from TopoGen vs. anti-Top1 from Abcam – and the mapping resolution, Top1cc-seq is quite similar to Top1-seq published by Baranello et al. Cell 2016. The SDS-based lysis buffer

used in the Top1-seq ensures that only SDS-resistant Top1-DNA complexes would be trapped and immunoprecipitated by Top1 antibody. In addition, Top1-seq maps active Top1 at a single nucleotide resolution, while the Top1cc-seq does not.

Nevertheless the authors claim that the Top1-seq “cannot provide a direct measurement of Top1-DNA transient intermediate” without explaining why. But looking at the data and considering that only 22% of peaks from Top1cc localize at promoters and 70% at gene’s body and intergenic regions (and similar results are obtained in the LNCAP cells), the Top1cc-seq resembles quite well the profile from Top1-seq, where the signal is strongly enriched at gene’s body. It would be important to generate a metaplot (histogram plot) showing the normalized Top1cc-seq coverage across all expressed genes (from transcription start to termination regions) to see how Top1-DNA complexes detected by Top1cc-seq distribute in relation to Top1 ChIP-seq and Top1-seq data. Because of the above considerations, I do not see novelty in the Top1cc-seq technique and I suggest the authors to rephrase the description of Top1cc-seq as a modified version of the Top1-seq approach.

- The second concern I have is with the timing of CPT treatment to trap Top1 on the DNA. The high eRNA levels produced at E2-activated enhancers will require high Top1 activity to release transcription-generated supercoiling, therefore 10 minutes of Top1 inhibition by CPT will most likely activate DNA repair pathways. Accordingly, recent publications highlight the strong connection between Top1cc and the proteasome pathway, which triggers DNA repair (Canela A. et al. Mol. Cell 2019, and Sciascia N. et al. eLife 2020). This could explain many of the observations made in the paper. For example, the increased interaction between Top1cc and HP1g. It is known that HP1g is recruited in response to DNA damage (Oka Y, et al. Biochem Biophys Res Commun. 2011 and Soria G. & Almouzni G. Cell Cycle 2012) together with components of cohesin complex, which show increased Top1cc binding in the ReClip experiment. The authors need to rule out this possibility by looking at markers of DNA damage (for example γ -H2AX, or phospho-ATM and phospho-ATR) at selected ligand-responsive enhancers positive for Top1cc, with a time-course of CPT (including 10 minutes treatment).

Further, the binding of RNA Pol2 at genes is affected after 10 minutes of CPT (Baranello et al. Cell 2016), thus to correlate Top1 activity and eRNA levels at ligand-responsive enhancers, the PRO-seq assay should be performed in the conditions of the Top1cc-seq (CPT 10uM for 10 minutes).

- The third concern I have is related to the correlation between Mega Trans enhancers activity, Top1cc and phase-separation, which, in my view, could represent an exciting discovery but needs to be carefully addressed. Are Top1, HP1g, Med 26 involved in the formation of liquid-like condensates at Mega Trans enhancers? Is this the mechanism leading to activation of Mega Trans enhancers? These aspects need to be addressed. Performing the experiments in presence of 1,6 and 2,5 HD could provide mechanistic details about the nature of the phase.

Minor comments:

- In the Methods section (line 317) “(+)-JQ-1 (JQ1, MCE, Cat# HY-13030) treatment was added to a final concentration of 10 μ M and 1hr before we treated cells with EtOH or E2”. I do not see any experiments with JQ1 in the paper. Probably a mistake, please clarify.

- Line 141. “These data suggest that TOP1cc serves as a critical signature for the optimally

robustly ligand-dependent activated Mega Trans enhancers." A simpler interpretation of the data is that Mega Trans enhancers have higher transcriptional activity (eRNA levels), which will generate higher level of torsional stress and require higher Top1 activity compared to weak Mega Trans enhancers.

On the same line, the authors compare Med26 binding at Mega Trans enhancers vs "other active enhancers" claiming a selective recruitment of Med26 at Mega Trans enhancers. However is not clear from the Methods section whether the two classes of enhancers are comparable in term of amount of eRNAs transcribed. To make a stronger case, it would be important to study Med26 binding comparing Mega Trans and "other active" enhancers with similar level of eRNA expression.

- For consistency in the presentation of data, the single locus enhancers selected to quantify Top1cc and eRNAs should be always the same. It is quite confusing that in Fig 1F, 3B, 4A, Ext 1C, Ext 5B etc. the enhancers are different. I suggest to improve this aspect and the figures, and look at P2ry2e, Tff1e, Greb1 and Nrip1 enhancers in all the single locus experiments.

- Ext Fig 1 A, typo in Y axis. It should be "Top1cc at 30 min (Log2)".

- Line 223. Typo: it should be "acutely" instead of "actuely".

Reviewer #2:

Remarks to the Author:

The authors detail a novel method for mapping sites of DNA-protein covalent adducts by mapping the location of Topoisomerase 1 covalent complexes (TOP1cc-seq). Novelty comes from the use of TOP1 poison Camptothecin (CPT) to stabilise covalent complexes formed between TOP1 and DNA. This allows direct measurement of the position of TOP1-cc, reducing signal from non-covalently bound TOP1 evident in signal in ChIP-sequencing and formaldehyde induced PARP. The authors use TOP1cc-seq to highlight the binding of TOP1 to covalent adducts at 17 β -estradiol (E2) induced enhancers in MCF7 cells. They define a novel interaction with a transcription inhibitor/activator HP1 γ , and show that at a subset of enhancers acutely activated in response to E2 treatment, TOP1cc/HP1 γ interact with mediator component Med26 to regulate phase separated condensate formation.

The TOP1-cc approach appears to present a relatively simple approach to solve a number of problems associated with identification of DNA-protein covalent adducts, therefore could be a useful approach to the field. Importantly, the authors appear to have considered and controlled for confounding effects due to CPT treatment. The observation of greatly increased formation of covalent adducts at enhancers is also important, in particular in light of the potential for elevated DNA damage at highly transcribed regions.

The defined role of HP1 γ in contributing to condensate formation and associated enhancer activation at MegaTrans enhancers is notable. However, the specific link between TOP1-cc formation at enhancers and HP1 γ recruitment to enhancers is not clearly defined. In particular, the authors hypothesized that TOP1cc recruited HP1 γ via a direct interaction, without considering that the proteins colocalize due to localisation in the same enhancer condensates.

The observation that acute and chronic activation of MegaTrans and TNFa dependent enhancers differ in their enrichment for TOP1cc and requirement for HP1γ for activation is also intriguing, and of broad interest.

Overall, I am supportive of the manuscript, providing comments below are sufficiently accounted for.

Specific Points

Line 83: TOP1cc-seq reveals an interesting result showing increased recruitment of TOP1 to covalent adducts formed at enhancers activated by E2. This highlights an increased potential for covalent adduct formation at active enhancers. 19,003 TOP1cc sites were identified, corresponding to cumulative E2 dependent and E2 independent sites. However it is unclear whether sites present in CPT treated but E2 untreated cells are maintained following E2 treatment, or whether these represent distinct sites in each condition. The authors should clarify whether E2 treatment causes a switch in TOP1cc sites, or whether additional sites are generated upon E2 treatment. Lines 83-91 should be more logically ordered to clearly define these different peak populations. It would also be useful to understand more about how the increase in TOP1cc sites corresponds to the total number of enhancers activated by E2 treatment e.g gain of K27ac or novel sites of MegaTrans binding following E2.

Line 100: The authors demonstrate that TOP1cc is decreased in the TOP1 Y723F catalytic mutant expressing cells, highlighting the requirement for enzymatically active TOP1 for covalent adduct binding. The problem with this validation strategy is that the Y723F mutant is also used later in the paper to demonstrate that TOP1cc is important for enhancer activity (Extended figure 5). It's therefore unclear whether the decrease in TOP1cc signal with the Y723F mutant is a result of reduced binding of Y723F TOP1 to covalent adducts at enhancers, or due to a reduced occurrence of covalent adducts at a less active enhancer.

Line 106: The authors demonstrate that although general enrichment patterns appear similar, TOP1cc-seq captures more defined TOP1 signals than conventional TOP1 ChIPseq in LNCAP cells. This is reflected in fewer called peaks in TOP1cc-seq (10,479) compared to TOP1 ChIPseq (106,306). While 6,533 peaks overlap between each dataset, it is concerning that 3,946 out of 10,479 called peaks in the TOP1cc-seq data did not overlap TOP1 ChIPseq peaks, given that TOP1cc-seq uses TOP1 occupancy to identify sites of covalent adducts. The authors acknowledge that "that enrichment of TOP1 at the regulatory elements by conventional ChIP-seq is not always correlated with TOP1cc signals". However, it would be useful to show example UCSC browser views alongside the heatmap in extended figure 2C, and to compare enrichment of TOP1 ChIPseq signal at control loci that do not show any TOP1cc peaks. It would also be interesting to test whether the overlap between TOP1cc and TOP1 ChIPseq peaks improves at induced enhancers following E2 treatment, although we acknowledge that this could represent significant extra work, so do not consider the experiment essential.

Line 126: MegaTrans enhancers show the most enrichment for TOP1cc following E2 treatment. It would be interesting to understand how this relates to TOP1 ChIPseq? Are loci that are most enriched for TOP1cc also show the most enrichment in TOP1 ChIPseq data?

Line 136. The authors statement that TOP1cc serves as a mark for the condensation of MegaTrans components at E2 induced enhancers seems like an overstatement. Based on the data, this could also result from increased transcriptional activity (i.e PolII occupancy) at the most active enhancers, therefore is more likely a measure of enhancer activity (cofactor recruitment, PolII transcription) than condensate formation. Authors should take care to at least acknowledge alternative possibilities alongside broad statements such as this.

Line 148: TOP1 knockdown causes an accumulation of covalent adducts and reduced eRNA transcription. The observation that TOP1 activity is needed to resolve covalent adducts to facilitate enhancer transcription and enhancer activity is interesting and important.

Line 165: HP1 γ ChIPseq signal is highly correlated to TOP1cc signal genome wide and knock-down of Top1 decreased the enrichment of HP1 γ at the TOP1cc/HP1 γ co-localized regions, with no effect at activation-independent regions. Recruitment of HP1 γ was impaired by the Y723F mutant TOP1. It was suggested that these results demonstrate that TOP1 is required for direct recruitment of HP1 γ .

The link between TOP1-cc and HP1 γ is well made. However, the proposed mechanism for this interaction is currently poorly defined and fails to discriminate between the direct recruitment of HP1 γ by TOP1-cc and sequestration of HP1 γ into a condensate formed at MegaTrans enhancers upon acute activation. More evidence in support of the direct recruitment hypothesis should be considered, if this is indeed the model, and acknowledgement of alternative models for recruitment would be welcome.

For example:

- knockdown of other MegaTrans components to study recruitment of HP1 γ and TOP1cc to enhancers upon reduced enhancer activity resulting from disruption of MegaTrans complex. Is recruitment of HP1 γ impaired without disrupting TOP1cc?

- Co-Immunoprecipitation of TOP1 and HP1 γ following E2 treatment. Comparison with the Y723F TOP1 mutant with impaired recruitment to covalent adducts would discriminate direct interactions of TOP1 with HP1 γ from co-localisation in enhancer condensates.

- If available, identification and disruption of directly interacting domains of either HP1 γ or TOP1 would be convincing, although it is understood that this could represent significant extra work for the current manuscript.

Line 171: Knock-down of Top1 decreased the E2-dependent enrichment of HP1 γ at the TOP1cc-enriched MegaTrans enhancers. It was suggested that this demonstrated a direct role for TOP1cc in recruiting HP1 γ to acutely activated enhancers. However, as knockdown of Top1 also decreases enhancer activity and the production of eRNAs (extended data 4), another possibility is that this decreases condensate formation at enhancers, and therefore disrupts sequestration of HP1 γ into the enhancer condensate. The authors could consider alternative methods of disrupting enhancer function without altering underlying levels of DNA adducts to confirm that this process is specifically due to TOP1 rather than condensate formation. For example depletion of other MegaTrans components or targeting the enhancer local with CRISPRi.

Minor points

Line 93: Figure legend for Ext 1 C should state that this is a ChIP experiment.

Line 100: Reference to demonstrate importance of enzymatic activity for TOP1cc formation, and Y723F mutant should be included here.

Line 130: need reference for MegaTrans enhancers.
The figure legend for Figure 3b should clearly state that the Dox induction replacement strategy uses the enzymatically dead Y723F TOP1 mutant.

Line 164: Extended figure 6a shows overlap of TOP1cc and HP1 γ peaks, not a correlation of TOP1cc and HP1 γ . Either the text or the figure should be corrected to reflect this.

Line 175: An explanation should be provided for enrichment of HP1 γ at Nrip1e in absence of E2 and DOX?

Line 264 "finding that TOP1-DNA transient intermediates, as assessed by TOP1cc-seq, is required for acute signal-dependent activation of functional regulatory enhancers." The results don't currently demonstrate a requirement for TOP1 covalent adducts for enhancer function, rather that increased adduct formation occurs at acutely activated enhancers, and a requirement for TOP1 to resolve adducts for efficient enhancer induction. These are still important results, but the discussion should reflect this.

Reviewer #3:

Remarks to the Author:

In this paper, the authors modify a ChIP-seq protocol to identify regions occupied by TOP1 in response to estrogen induction. They go on to show that these regions are often associated with enhancers and follow gene activation. They subsequently identify HP1 γ as an interacting protein. This is unexpected as HP1 γ is thought to be associated with heterochromatin. They go on to show that HP1 γ is required for the enhancer activity and the activation of associated genes. Interestingly, it only appears to be required for acutely activated genes and not chronically activated genes. This in turn is correlated with different forms of phase separation (identified in a previous paper) that is also associated with an acute response. Overall, this is therefore an interesting advance, although it is not clear what role HP1 γ has in this context and how it can have both repressive and activating roles.

Technically, the paper is excellent and in general, the conclusions drawn from the paper are appropriate. The manuscript is well written and clear. However, there are a few areas that should be revised to help with the clarity of the manuscript.

Issues to address:

(1) I am not convinced that "TOP1cc-seq" needs to be in the title as this technique is not of broad utility, only to those doing ChIP-seq with TOP1. To focus on the novel findings more, better to just say it is TOP1-dependent HP1 recruitment?

(2) In Fig 2, the authors introduce the concept of Mega Trans enhancers (elucidated in their previous work). In addition to the extended description in the materials and methods, a brief introduction should be provided in the main text, which lists the key features that define these enhancers.

(3) For completeness, a WT rescue should be added to Fig. 3b.

(4) Is HP1 γ required for liquid-liquid phase separation at enhancers? Can this be tested?

(5) In extended data 3, the ERE is one of the least abundant motifs. Can the authors comment on the others and their relevance? Also, the axes are presumably mislabelled

and it should be 20% and not 0.2% of targets.

(6) For clarity, in Ext Fig. 4b, add the tracks of histone modifications that define enhancers if available as this alternative mark will help visualise the designation of the enhancer region.

(7) In extended Ext Fig. 6c, the HP1 γ signal is the same in the non-shared regions, so what is recruiting it there?

(8) In Ext fig 7e, the regions that are heterochromatic and overlap with HP1 γ as indicated in the text should be marked. Currently the overlaps with TOP1 are clear but not with the heterochromatin.

(9) Can the authors comment on the loss of HP1 γ signal on the TFF1 locus? Also they should show the tracks for TOP1 in Ext Fig. 8b.

(10) More quantitative analysis of the data in Ext Fig. 10b&c should be shown to make the differences clearer to visualise (eg average tag density plots).

(11) In general, the choice of adding things to the Supplementary is rather random. Some is clearly contributory and more of a control or deeper analysis. However other things are more central to the story. The authors should consider bringing more of the data into the main manuscript by creating addition figures as this would make the paper easier to follow.

Author Rebuttal, first revision:

Dear Dr. Moorefield,

One of the benefits of having a manuscript reviewed at *Nature Structural and Molecular Biology (NSMB)* is the opportunity to receive suggestions that can provoke experiments that elevate the level of the submitted manuscript. We highly appreciate the rigorous review of our manuscript, and the cogent concerns and excellent suggestions by the Referees. Indeed, based on the reviews, we have taken the opportunity to bring the manuscript to a higher level, and taken the full time required to explore in depth the issues raised by the Reviewers, particularly their concerns about interpretation of genome-wide localization data using a TOP1 inhibitor, and a requirement to determine the factor interacting with TOP1cc that underlies to recruitment of HP1 γ . Pursuing these questions has resulted in a profound and previously unappreciated insight into enhancer activation. We hope that you and the Referees will permit a rather protracted overview of the new findings and of the reasons that we are proud to submit our extensively revised manuscript titled with "Acute Signal-dependent Enhancer Activation Requires Ku70 Binding of Topoisomerase1-DNA Covalent Complexes".

The basic question explored in the manuscript centers around the well-documented observations that activation of enhancers, characterized by augmented eRNA transcription, serves as the major mechanism regulating acute signal-dependent modulation of transcriptional programs in virtually all metazoan cell types, reflecting the preferential binding of the vast majority of signal-dependent transcription factors to enhancers, rather than promoters. Thus, signal-dependent enhancer activation generally temporally precedes activation of its cognate promoter, and most strong and acutely activated enhancers exhibit transcribed eRNAs, required to induce target gene transcription. However, because of the great diversity of enhancer sequences and transcription factors bound, a crucial remaining question has been whether there was an as yet an undiscovered epigenomic strategy that is universally required for all acute signal-dependent enhancer activation, despite their diversity.

The first issue was to develop an assay that accurately identified the location of TOP1cc in the genome. In light of the cogent comment by Referee 1 on potential concerns with CPT treatment, we modified the approach and took advantage of a monoclonal antibody specific against TOP1cc and combined it with a CUT&RUN assay in the absence of CPT. In the process, we have established a powerful assay that, for the first time, has provided with unprecedented precision the activation of topoisomerase 1 with the appearance of TOP1cc on enhancers that are acutely activated in response to signals/ligands, such as the hormone estradiol-17 β (E2). We have included comparisons with TOP1 ChIP-Seq data in this revised version as requested by the Referees. These data are presented in **Fig. 1 and Extended Data Fig 1.**

Empowered by this new assay to detect TOP1cc, we were in a position to investigate whether actions of TOP1cc represented a general mechanism for most, or possibly all, signal-dependent enhancer activation events. To our total surprise, we found that TOP1cc and subsequent downstream events are required for the activation of enhancers regulating a variety of cellular process in multiple cell types, including nuclear hormone receptor ligand (E.g. estrogen and androgen), inflammatory signal regulators (e.g. TNF α), and even

for depolarization-induced neuronal enhancer activation in cortical neurons (**Fig. 5**). Thus, from our studies, TOP1cc emerges as a broadly used transcriptional code for the activation of acute signal-dependent enhancers.

To provide a deep mechanism underlying the connection of between TOP1cc and HP1 γ , we performed novel proteomics studies which, unexpectedly, revealed that TOP1cc promoted the tethering of DNA damage repair protein Ku70, but apparently not its classic interaction partner, Ku80. Moreover, Ku70 appears to “read” TOP1cc as an “epigenomic” mark, with no apparent participation of other DNA damage-repair machinery. Indeed, Ku70 interactions with TOP1cc proved to be the beacon for the subsequent recruitment of additional transcriptional machinery for acute signal-dependent enhancer activation, including phosphorylated HP1 γ and Med26. Thus, in contrast to the conventional view that Ku70 functions exclusively in DNA damage repair, it appears that Ku70 has co-evolved a second, independent role as a transcriptional coactivator, functioning as a “reader” of TOP1cc. Together, these studies reveal that TOP1cc, and subsequent downstream events are required for the activation of enhancers regulating a variety of cellular processes in multiple cell types. TOP1cc emerges as a new kind of transcriptional signal “read” by Ku70 to license nucleation of the HP1 γ /Med26 complex for acute signal-induced enhancer activation (**Fig.3 and Fig. 4**). This new molecular strategy appears to be a general feature of acute (but not chronic) signal-dependent enhancer activation events in many cell types. Our data also suggests a new non-canonical role for topoisomerase 1 in regulating signal-dependent enhancer activation, whereby it helps to mobilize the assembly of transcription factors. We really believe that these new results fundamentally change our concepts of the relationship between DNA damage repair factors and transcription.

In answering the concerns by the Referees, we believe that we have made a discovery linking DNA damage repair machinery and enhancer-dependent transcriptional programs in response to signal/ligand activation, which elevates the manuscript to a fundamentally higher level. Our specific responses to the Referees:

Reviewers' Comments:

Reviewer #1:

Remarks to the Author:

The connection between Topoisomerase 1 (Top1) function and ligand-responsive enhancer activation has been well established by Puc and Rosenfeld in Puc J. et al. Cell 2015. In this paper, Tan and colleagues extend the story by proposing that Top1 activity is important to regulate enhancers induced by acute but not by chronic stimuli. In response to estrogen (E2), enzymatically active Top1 (Top1cc) is detected at E2-responsive enhancers and required for the transcription of eRNA. The Top1cc mediates the tethering of heterochromatin protein 1 gamma (HP1g) to the enhancers and this, in turn, favours recruitment of Med26 and Pol2. Because the Top1cc/HP1g/Med26 complexes are preferentially enriched at Mega Trans enhancers, which are sensitive to 1,6 HD (shown previously by the same group, Nair S. et al. NSMB 2019), the authors suggest a mechanistic

link between Top1 activity and formation of liquid-like condensates to achieve enhancers activation.

The result presented by Tan and colleagues are interesting and could have impact on our understanding of Top1 function during enhancer activation but not conceptually new. Unfortunately, in the current form the work is not appropriate for publication in NSMB. My reservations and specific points of critique are outlined below.

Our response: We appreciate that reviewer noting the potential impact of our work in understanding of TOP1 function during enhancer activation. In this revised manuscript, we provide remarkable new evidence showing how the classic DNA damage sensor protein-Ku70 is recruited to the enhancers to facilitate the transcriptional activation programs by nucleating HP1 γ /Med26 complexes at enhancers. Furthermore, we find that this TOP1cc/Ku70/HP1 γ strategy is not merely limited to Estrogen receptor-activated enhancers but also extends to NF κ B and Androgen receptor-activated enhancers, and even to depolarization-dependent activation of enhancers in post-mitotic cortical neurons. Overall, the revised manuscript now includes documentation of a novel TOP1cc/Ku70/HP1 γ transcriptional code required for most, if not all, signal-dependent acutely activated enhancers. These results greatly enhance the conceptual newness of our manuscript.

Major comments:

- The first concern I have in this paper is with the Top1cc-seq protocol. The authors describe it as a novel approach “enabling for the first time to disclose the TOP1-DNA transient intermediates genome-wide”. Except for the antibody used – anti-Top1cc from TopoGen vs. anti-Top1 from Abcam – and the mapping resolution, Top1cc-seq is quite similar to Top1-seq published by Baranello et al. Cell 2016. The SDS-based lysis buffer used in the Top1-seq ensures that only SDS-resistant Top1-DNA complexes would be trapped and immunoprecipitated by Top1 antibody. In addition, Top1-seq maps active Top1 at a single nucleotide resolution, while the Top1cc-seq does not. Nevertheless the authors claim that the Top1-seq “cannot provide a direct measurement of Top1-DNA transient intermediate” without explaining why. But looking at the data and considering that only 22% of peaks from Top1cc localize at promoters and 70% at gene’s body and intergenic regions (and similar results are obtained in the LNCAP cells), the Top1cc-seq resembles quite well the profile from Top1-seq, where the signal is strongly enriched at gene’s body. It would be important to generate a metaplot (histogram plot) showing the normalized Top1cc-seq coverage across all expressed genes (from transcription start to termination regions) to see how Top1-DNA complexes detected by Top1cc-seq distribute in relation to Top1 ChIP-seq and Top1-seq data. Because of the above considerations, I do not see novelty in the Top1cc-seq technique and I suggest the authors to rephrase the description of Top1cc-seq as a modified version of the Top1-seq approach.

Our response: We fully appreciate the conclusion of the paper by Baranello et al. Cell (2016), which clearly showed that SDS-based lysis buffer could ensure that only SDS-resistant TOP1-DNA complexes would be trapped and immunoprecipitated. However, while the anti-TOP1

antibody from Abcam could pull-down the complexes with TOP1, DNA nicks are not necessarily due to the actions by TOP1, because there are about 20 different proteins could potentially form covalent interactions with DNA. Further, the recent papers in *Nature* and *Science* also clearly showed that the enhancers are the hotspots of the single-strand DNA breaks, and those single-strand DNA breaks are TOP1-independent (Reid et al., *Science*, 2021; Wu et al., *Nature*, 2021). Here, in our manuscript, anti-TOP1cc antibody from TopoGen detects only the covalent interactions with TOP1 and DNA. In addition, although we fully agree that TOP1-seq maps active TOP1 at a single nucleotide resolution, Baranello *et al.* tried to add the adaptors to both ends of the double strand DNA generated by sonication, and then added another adaptor to detect the regions with DNA nicks. However, due to the relative low ligation efficiency in the first step, and the fact that sonication might generate some possible single-strand DNA breaks, the data generated by that method would generate high experimental noise, which prevents the detection of the signal-dependent TOP1 DNA nicking at enhancers. The detection of TOP1 dependent DNA nicking at regulatory enhancer is the major focus of our manuscript.

- The second concern I have is with the timing of CPT treatment to trap Top1 on the DNA. The high eRNA levels produced at E2-activated enhancers will require high Top1 activity to release transcription-generated supercoiling, therefore 10 minutes of Top1 inhibition by CPT will most likely activate DNA repair pathways. Accordingly, recent publications highlight the strong connection between Top1cc and the proteasome pathway, which triggers DNA repair (Canela A. et al. *Mol. Cell* 2019, and Sciascia N. et al. *eLife* 2020). This could explain many of the observations made in the paper. For example, the increased interaction between Top1cc and HP1g. It is known that HP1g is recruited in response to DNA damage (Oka Y, et al. *Biochem Biophys Res Commun.* 2011 and Soria G. & Almouzni G. *Cell Cycle* 2012) together with components of cohesin complex, which show increased Top1cc binding in the ReClip experiment. The authors need to rule out this possibility by looking at markers of DNA damage (for example γ -H2AX, or phospho-ATM and phospho-ATR) at selected ligand-responsive enhancers positive for Top1cc, with a time-course of CPT (including 10 minutes treatment). Further, the binding of RNA Pol2 at genes is affected after 10 minutes of CPT (Baranello et al. *Cell* 2016), thus to correlate Top1 activity and eRNA levels at ligand-responsive enhancers, the PRO-seq assay should be performed in the conditions of the Top1cc-seq (CPT 10uM for 10 minutes).

Our response: We highly appreciate this cogent concern, and indeed we do find that CPT treatment has transcriptional effects, and it will increase the transcriptional activation in the genome. Therefore, we highly agree that it is very complicated and challenging to explain the correlation between the formation of TOP1cc and transcriptional activation in the presence of the CPT. As such, we have invested almost 2 years to optimize our methods for mapping TOP1cc in the absence of CPT. In the revised manuscript, we have now successfully employed CUT&RUN assays to detect TOP1cc and discovered that TOP1cc and its subsequently assembled Ku70/HP1 γ complexes are not only detected in human proliferating cells upon acute signals, but also induced by depolarized post-mitotic neurons, which profoundly extends the generality of our initial observations.

- The third concern I have is related to the correlation between Mega Trans enhancers activity, Top1cc and phase-separation, which, in my view, could represent an exciting discovery but needs to be carefully addressed. Are Top1, HP1g, Med 26 involved in the formation of liquid-like condensates at Mega Trans enhancers? Is this the mechanism leading to activation of Mega Trans enhancers? These aspects need to be addressed. Performing the experiments in presence of 1,6 and 2,5 HD could provide mechanistic details about the nature of the phase.

Our response: We agree that the mechanism underlying TOP1, HP1 γ , Med 26 involvement in the signal-dependent formation of estrogen-receptor-mediated liquid-like condensates at the MegaTrans enhancers is quite interesting. However, one concern we had in performing experiments in the presence of 1,6HD is that most coactivators might be removed from the activated *cis*-regulatory elements. Thus, we now show that knock-down of HP1 γ impairs the recruitment of a crucial component for the MegaTrans complex at these enhancers, namely GATA3. GATA3 has previously been shown to be essential to maintain the MegaTrans complexes at enhancers (Liu et al., *Cell*, 2015; Nair et al., *NSMB*, 2019); as such, we are tempted to suggest that HP1 γ is important in maintaining the liquid-like condensate properties of the MegaTrans enhancers.

Minor comments:

- In the Methods section (line 317) "(+)-JQ-1 (JQ1, MCE, Cat# HY-13030) treatment was added to a final concentration of 10 μ M and 1hr before we treated cells with EtOH or E2". I do not see any experiments with JQ1 in the paper. Probably a mistake, please clarify.

Our response: We thank the Referee for this point; we have now deleted this information.

- Line 141. "These data suggest that TOP1cc serves as a critical signature for the optimally robustly ligand-dependent activated Mega Trans enhancers." A simpler interpretation of the data is that Mega Trans enhancers have higher transcriptional activity (eRNA levels), which will generate higher level of torsional stress and require higher Top1 activity compared to weak Mega Trans enhancers.

Our response: We thank the Referee for this suggestion, but note that, unlike promoters, no supercoiling has been observed at enhancer elements (e.g. Kouzine et al. *Nature Struct. Mol. Biol.*, 2013 – see supplementary Figure 1b). Much of our data points to the intriguing possibility of a "non-canonical" role for TOP1, wherein rather than relief of supercoiling or torsional stress per se, it instead helps to mobilize the assembly of transcription factors. For example, if relief of torsional stress was the primary initiator of enhancer transcription, we would see TOP1cc at all active enhancers, but this is not the case. Instead, there is no increased TOP1cc at either basally active, or chronically ligand-regulated enhancers. We only observe elevated TOP1cc on acutely activated signal-dependent enhancers, leading, as show now, to the recruitment of Ku70 and to the licensing of eRNA elongation. In the absence of Ku70, enhancer activation and eRNA transcription is not observed. Further, on enhancers enriched with only TOP1cc (without Ku70), RNA polymerase II could not be activated. Also, we note that stabilized TOP1cc generated by TOP1 inhibitors, which is well-

known to inhibit the relief of torsional stress, could induce the transcription of nascent RNA in previously published studies. Overall, we favor the idea that rather than relief of torsional stress at enhancers per se, TOP1cc serves as intermediate read by Ku70, and have inserted these statements in line 380~397.

On the same line, the authors compare Med26 binding at Mega Trans enhancers vs “other active enhancers” claiming a selective recruitment of Med26 at Mega Trans enhancers. However is not clear from the Methods section whether the two classes of enhancers are comparable in term of amount of eRNAs transcribed. To make a stronger case, it would be important to study Med26 binding comparing Mega Trans and “other active” enhancers with similar level of eRNA expression.

Our response: We thank the Referee for these points; we now have selected another group of active enhancers with similar level of eRNAs and show these data in Extended Fig.9a.

- For consistency in the presentation of data, the single locus enhancers selected to quantify Top1cc and eRNAs should be always the same. It is quite confusing that in Fig 1F, 3B, 4A, Ext 1C, Ext 5B etc. the enhancers are different. I suggest to improve this aspect and the figures, and look at P2ry2e, Tff1e, Greb1 and Nrip1 enhancers in all the single locus experiments.

Our response: We thank the Referee for these points; we now have provided genome browser images for P2ry2, Tff1, Greb1 and Nrip1 enhancers in the revised manuscript.

- Ext Fig 1 A, typo in Y axis. It should be “Top1cc at 30 min (Log2)”.

Our response: We thank the Referee for this point; we have now employed a new method to detect TOP1cc and have deleted these data cited above in this new version.

- Line 223. Typo: it should be “acutely” instead of “actuely”.

Our response: We now have corrected it.

Summary of Response to Reviewer1: We particularly thank the Reviewer for the critical contribution to this revised manuscript based on the cogent comments regarding both the assay and the precise role of TOP1cc in the initial submission.

The experiments provoked by these comments have served as the basis of the additional revealing discovery that the critical role of TOP1cc is as an acute signal-dependent activation mark for most, if not all, regulated enhancers, providing an epigenomic “mark” read by Ku70, required for assembling the machinery permitting eRNA elongation and robust enhancer activation. These comments were spot on, and we appreciated the Referee’s key contributions to the final, revised manuscript.

Reviewer #2:

Remarks to the Author:

The authors detail a novel method for mapping sites of DNA-protein covalent adducts by mapping the location of Topoisomerase 1 covalent complexes (TOP1cc-seq). Novelty comes from the use of TOP1 poison Camptothecin (CPT) to stabilise covalent complexes formed between TOP1 and DNA. This allows direct measurement of the position of TOP1-cc, reducing signal from non-covalently bound TOP1 evident in signal in ChIP-sequencing and formaldehyde induced PARP. The authors use TOP1cc-seq to highlight the binding of TOP1 to covalent adducts at 17 β -estradiol (E2) induced enhancers in MCF7 cells. They define a novel interaction with a transcription inhibitor/activator HP1 γ , and show that at a subset of enhancers acutely activated in response to E2 treatment, TOP1cc/HP1 γ interact with mediator component Med26 to regulate phase separated condensate formation.

The TOP1-cc approach appears to present a relatively simple approach to solve a number of problems associated with identification of DNA-protein covalent adducts, therefore could be a useful approach to the field. Importantly, the authors appear to have considered and controlled for confounding effects due to CPT treatment. The observation of greatly increased formation of covalent adducts at enhancers is also important, in particular in light of the potential for elevated DNA damage at highly transcribed regions.

The defined role of HP1 γ in contributing to condensate formation and associated enhancer activation at MegaTrans enhancers is notable. However, the specific link between TOP1-cc formation at enhancers and HP1 γ recruitment to enhancers is not clearly defined. In particular, the authors hypothesized that TOP1cc recruited HP1 γ via a direct interaction, without considering that the proteins colocalize due to localisation in the same enhancer condensates.

The observation that acute and chronic activation of MegaTrans and TNF α dependent enhancers differ in their enrichment for TOP1cc and requirement for HP1 γ for activation is also intriguing, and of broad interest.

Overall, I am supportive of the manuscript, providing comments below are sufficiently accounted for.

Our response: We highly appreciate the Referee’s comments concerning our method to detect the genomic sites of DNA-protein covalent adducts by mapping the location of

Topoisomerase 1 covalent complexes (TOP1cc-seq), and we also appreciate the comments of the novel interaction between TOP1cc and HP1 γ , which facilitate the transcriptional activation at enhancers through recruitment of Med26. We, however, took particular note of the Referee's comment "However, the specific link between TOP1cc formation at enhancers and HP1 γ recruitment to enhancers is not clearly defined". To appropriately investigate this important point required almost 2 years, ultimately revealing that a classic DNA damage sensor protein-Ku70- could read the TOP1cc at enhancers and establishing that it was Ku70 that brought HP1 γ to acutely activated enhancers, which is consistent with the ability of Ku70 and HP1 γ to interact (Lomberk et al., *Nat Cell Biol*, 2003). We thank the Referee for suggesting the additional clarification, which has catalyzed an important clarification of the role of TOP1cc in acute signal/ligand-dependent activation of enhancers.

Specific Points

Line 83: TOP1cc-seq reveals an interesting result showing increased recruitment of TOP1 to covalent adducts formed at enhancers activated by E2. This highlights an increased potential for covalent adduct formation at active enhancers. 19,003 TOP1cc sites were identified, corresponding to cumulative E2 dependent and E2 independent sites. However it is unclear whether sites present in CPT treated but E2 untreated cells are maintained following E2 treatment, or whether these represent distinct sites in each condition. The authors should clarify whether E2 treatment causes a switch in TOP1cc sites, or whether additional sites are generated upon E2 treatment. Lines 83-91 should be more logically ordered to clearly define these different peak populations. It would also be useful to understand more about how the increase in TOP1cc sites corresponds to the total number of enhancers activated by E2 treatment e.g gain of K27ac or novel sites of MegaTrans binding following E2.

Our response: We highly appreciate the Referee raising this important point. Indeed, we subsequently found that CPT treatment can increase the transcriptional activation in the genome, and that it would be unwise to interpret the location of TOP1cc in the presence of CPT. Therefore, although it required almost 2 years to optimize our methods, in this revised manuscript, we have developed and employed CUT&RUN assays in the absence of CPT to detect TOP1cc globally, now establishing its appearance on acute signal/ligand-dependent activation of enhancers. This and the subsequent formation of Ku70/HP1 γ complexes proved to be the case for estrogen, androgen or NF κ B-activated enhancers, as well as for depolarization-activated enhancers in neurons, indicating the widespread use of this regulatory strategy.

Line 100: The authors demonstrate that TOP1cc is decreased in the TOP1 Y723F catalytic mutant expressing cells, highlighting the requirement for enzymatically active TOP1 for covalent adduct binding. The problem with this validation strategy is that the Y723F mutant is also used later in the paper to demonstrate that TOP1cc is important for enhancer activity (Extended figure 5). It's therefore unclear whether the decrease in

TOP1cc signal with the Y723F mutant is a result of reduced binding of Y723F TOP1 to covalent adducts at enhancers, or due to a reduced occurrence of covalent adducts at a less active enhancer.

Our response: We highly appreciate this point. In our manuscript, we could not exclude the possibility that Y723F mutant is a result of reduced binding of Y723F TOP1 at enhancers. Instead, we show that the formation of the TOP1cc is important for the transcriptional activation at enhancers, and the mutant Y723F of TOP1 failed to promote the formation of TOP1cc at signal-dependent acutely activated enhancers (**Fig. 1d,e**).

Line 106: The authors demonstrate that although general enrichment patterns appear similar, TOP1cc-seq captures more defined TOP1 signals than conventional TOP1 ChIPseq in LNCAP cells. This is reflected in fewer called peaks in TOP1cc-seq (10,479) compared to TOP1 ChIPseq (106,306). While 6,533 peaks overlap between each dataset, it is concerning that 3,946 out of 10,479 called peaks in the TOP1cc-seq data did not overlap TOP1 ChIPseq peaks, given that TOP1cc-seq uses TOP1 occupancy to identify sites of covalent adducts. The authors acknowledge that “that enrichment of TOP1 at the regulatory elements by conventional ChIP-seq is not always correlated with TOP1cc signals”. However, it would be useful to show example UCSC browser views alongside the heatmap in extended figure 2C, and to compare enrichment of TOP1 ChIPseq signal at control loci that do not show any TOP1cc peaks.

Re: We now provide the UCSC browser views alongside the heatmap in **Extended Data Fig. 1d**.

It would also be interesting to test whether the overlap between TOP1cc and TOP1 ChIPseq peaks improves at induced enhancers following E2 treatment, although we acknowledge that this could represent significant extra work, so do not consider the experiment essential.

Line 126: MegaTrans enhancers show the most enrichment for TOP1cc following E2 treatment. It would be interesting to understand how this relates to TOP1 ChIPseq? Are loci that are most enriched for TOP1cc also show the most enrichment in TOP1 ChIPseq data?

Our response: We thank the Referee for the comments on the TOP1 ChIP-seq data following E2 treatment. We tried several times to perform TOP1 ChIP-seq in MCF7 cells, however, it seems that TOP1 antibody only worked effectively in LNCAP cells. We believe that it is because traditional chromatin-immunoprecipitation (ChIP) assays induce very high artificial formation of poly ADP-ribose, which is involved in the formation and release of TOP1cc. However, we are now providing the TOP1 ChIP-seq data in MCF7 cells in the revised **Fig. 1c**, as the CUT&RUN assay dramatically increased the detection of TOP1cc.

Line 136. The authors statement that TOP1cc serves as a mark for the condensation of MegaTrans components at E2 induced enhancers seems like an overstatement. Based on

the data, this could also result from increased transcriptional activity (i.e PolII occupancy) at the most active enhancers, therefore is more likely a measure of enhancer activity (cofactor recruitment, PolII transcription) than condensate formation. Authors should take care to at least acknowledge alternative possibilities alongside broad statements such as this.

Our response: We thank the Referee for these comments and agree that we had overstated when suggesting that TOP1cc serves as a mark for the condensation at enhancers, and now have removed these statements in the revised manuscript.

Line 148: TOP1 knockdown causes an accumulation of covalent adducts and reduced eRNA transcription. The observation that TOP1 activity is needed to resolve covalent adducts to facilitate enhancer transcription and enhancer activity is interesting and important.

Our response: We thank for the Referee for these comments on the TOP1 and CPT-induced TOP1cc. We agree that the CPT makes it is very complicated to prove the function of TOP1-dependent transitory DNA nicking in enhancer activation, and now have updated our methods, establishing that TOP1cc is required for the recruitment of the classic DNA damage protein-Ku70, which is functioning surprisingly as the transcriptional activator for acute enhancer activation via recruitment of HP1 γ and Med26.

Line 165: HP1 γ ChIPseq signal is highly correlated to TOP1cc signal genome wide and knock-down of Top1 decreased the enrichment of HP1 γ at the TOP1cc/HP1 γ co-localized regions, with no effect at activation-independent regions. Recruitment of HP1 γ was impaired by the Y723F mutant TOP1. It was suggested that these results demonstrate that TOP1 is required for direct recruitment of HP1 γ . The link between TOP1-cc and HP1 γ is well made. However, the proposed mechanism for this interaction is currently poorly defined and fails to discriminate between the direct recruitment of HP1 γ by TOP1-cc and sequestration of HP1 γ into a condensate formed at MegaTrans enhancers upon acute activation. More evidence in support of the direct recruitment hypothesis should be considered, if this is indeed the model, and acknowledgement of alternative models for recruitment would be welcome.

For example:

-knockdown of other MegaTrans components to study recruitment of HP1 γ and TOP1cc to enhancers upon reduced enhancer activity resulting from disruption of MegaTrans complex. Is recruitment of HP1 γ impaired without disrupting TOP1cc?

-Co-Immunoprecipitation of TOP1 and HP1 γ following E2 treatment. Comparison with the Y723F TOP1 mutant with impaired recruitment to covalent adducts would discriminate direct interactions of TOP1 with HP1 γ from co-localisation in enhancer condensates.

-If available, identification and disruption of directly interacting domains of either HP1 γ or TOP1 would be convincing, although it is understood that this could represent significant extra work for the current manuscript.

Our response: We highly appreciate the comments indicating that the proposed mechanism for this interaction between the direct recruitment of HP1 γ by TOP1-cc was unconvincing, prompting new proteomic experiments. We modified our Mass spec experiments by avoiding DNase, as DNase would impair the detection of DNA-based protein complexes such as the NHEJ complex (Suwa et al., *PNAS*, 1994; Liu et al., *Nature Communications*, 2015; Xing et al., *Nature Communications*, 2015; Pellarin et al., *PLoS One*, 2016). These additional experiments revealed that the classic DNA damage sensor protein Ku70 could be tethered to the active enhancers due to the TOP1 DNA nicking and formation of TOP1cc (**Fig.3 and Extended Data Fig 6a, b**), and we showed that Ku70 is the key factor mediating the tethering of HP1 γ to the enhancers (**Fig.4**). Interestingly, this phenomenon is dependent on the phosphorylation of HP1 γ (**Extended Data Fig 7, 8**), which is consistent with previous findings about HP1 γ phosphorylation (Lomber et al., *Nat Cell Biol*, 2003).

Line 171: Knock-down of Top1 decreased the E2-dependent enrichment of HP1 γ at the TOP1cc-enriched MegaTrans enhancers. It was suggested that this demonstrated a direct role for TOP1cc in recruiting HP1 γ to acutely activated enhancers. However, as knockdown of Top1 also decreases enhancer activity and the production of eRNAs (extended data 4), another possibility is that this decreases condensate formation at enhancers, and therefore disrupts sequestration of HP1 γ into the enhancer condensate. The authors could consider alternative methods of disrupting enhancer function without altering underlying levels of DNA adducts to confirm that this process is specifically due to TOP1 rather than condensate formation. For example depletion of other MegaTrans components or targeting the enhancer local with CRISPRi.

Our response: We thank the Referee for the comments indicating the complexity of the events that occur with acute signal-dependent enhancer activation. It appears that activation of TOP1 on enhancers is one of the earliest events (Zobeck et al., *Mol Cell*, 2010; Puc et al., *Cell*, 2015), and that formation of TOP1cc is required for effective activation of the enhancers, but somewhat daunting to precisely separate the feed-forward events, However, we now provide evidence that the TOP1cc- dependent recruitment of Ku70 is required for robust enhancer activation (**Fig.3**); providing some independent evidence for the importance of TOP1cc itself and the subsequent recruitment of Ku70/ HP1 γ for enhancer activation. We favor the explanation that TOP1cc requirement for the liquid-like properties of enhancers is actually due to its association with the production of eRNAs, dependent on HP1 γ /Med26. As such, without HP1 γ , the liquid-like properties of the MegaTrans enhancers would be impaired. We now show that knock-down of HP1 γ can impair the recruitment of the crucial component for the Mega Trans complexes, namely GATA3 at the enhancers (**Extended Data Fig. 8d**). GATA3 is the key protein mediating the formation of MegaTrans on activated enhancers imparting putative liquid-like properties to the Mega Trans enhancers (Liu et al., *Cell*, 2015; Nair et al., *NSMB*, 2019), and we suggest that HP1 γ /Med26 and the induction of eRNA elongation is quite important in maintaining the liquid-like properties of the MegaTrans enhancers.

Minor points

Line 93: Figure legend for Ext 1 C should state that this is a ChIP experiment.

Our response: In response to this point, in the revised manuscript, we have employed CUT&RUN assays to detect TOP1cc, and also verified the TOP1cc signals.

Line 100: Reference to demonstrate importance of enzymatic activity for TOP1cc formation, and Y723F mutant should be included here.

Our response: We have added the reference for Y723F mutant in Line 133.

Line 130: need reference for MegaTrans enhancers.

Our response: We have added the reference for MegaTrans enhancers in Line 121.

The figure legend for Figure 3b should clearly state that the Dox induction replacement strategy uses the enzymatically dead Y723F TOP1 mutant.

Our response: We now explained this clearly in the figure legend for **Fig.1e**.

Line 164: Extended figure 6a shows overlap of TOP1cc and HP1 γ peaks, not a correlation of TOP1cc and HP1 γ . Either the text or the figure should be corrected to reflect this.

Our response: We provide these data in **Extended Data Fig. 7e**.

Line 175: An explanation should be provided for enrichment of HP1 γ at Nrip1e in absence of E2 and DOX?

Our response: The enrichment of HP1 γ is dependent on the TOP1cc/Ku70, and although it appears that the TOP1cc signal is lower in the absence of E₂, there might some TOP1 signals that could not be captured by our current assays due to the experimental limitations.

Line 264 “finding that TOP1-DNA transient intermediates, as assessed by TOP1cc-seq, is required for acute signal-dependent activation of functional regulatory enhancers.” The results don’t currently demonstrate a requirement for TOP1 covalent adducts for enhancer function, rather that increased adduct formation occurs at acutely activated enhancers, and a requirement for TOP1 to resolve adducts for efficient enhancer induction. These are still important results, but the discussion should reflect this.

Our response: We have removed this sentence in the revised manuscript.

Summary of Response to Reviewer 2: We particularly thank the Reviewer for the cogent comments and suggestions regarding the assays and proposed mechanisms that prompted us to unravel an unexpected and important new insight into the function of TOP1cc as an “epigenomic platform “read” by Ku70 that underlies the requirement for TOP1cc in acute signal-dependent activation of most, if not all, regulated enhancers.

Reviewer #3:

Remarks to the Author:

In this paper, the authors modify a ChIP-seq protocol to identify regions occupied by TOP1 in response to estrogen induction. They go on to show that these regions are often associated with enhancers and follow gene activation. They subsequently identify HP1 γ as an interacting protein. This is unexpected as HP1 γ is thought to be associated with heterochromatin. They go on to show that HP1 γ is required for the enhancer activity and the activation of associated genes. Interestingly, it only appears to be required for acutely activated genes and not chronically activated genes. This in turn is correlated with different forms of phase separation (identified in a previous paper) that is also associated with an acute response. Overall, this is therefore an interesting advance, although it is not clear what role HP1 γ has in this context and how it can have both repressive and activating roles.

Technically, the paper is excellent and in general, the conclusions drawn from the paper are appropriate. The manuscript is well written and clear. However, there are a few areas that should be revised to help with the clarity of the manuscript.

Our response: We highly appreciate the Referee’s comments and concur that the requirement of TOP1cc/HP1 γ for the acutely activated genes and not chronically activated genes is quite interesting. In appreciation of the comment that “it is not clear what role HP1 γ has in this context and how it can have both repressive and activating roles”, we modified our Mass spec experiments by not adding the DNase. We showed that the classic DNA damage sensor Ku70 protein could be tethered to the active enhancers due to the TOP1cc and showed that Ku70 is the key factor mediating the tethering of HP1 γ to the enhancers (**Fig. 4**). Interestingly, this phenomenon is dependent on the phosphorylation of HP1 γ , which is consistent with previous findings about HP1 γ , phosphorylation (Lomberk et al., *Nat Cell Biol*, 2003). For HP1 γ , we believe that unphosphorylated form is still resident at heterochromatin regions and is associated with transcriptional repression, while the phosphorylated HP1 γ is tethered to euchromatin and is required for the signal-dependent enhancer activation.

Issues to address:

(1) I am not convinced that “TOP1cc-seq” needs to be in the title as this technique is not of

broad utility, only to those doing CHIP-seq with TOP1. To focus on the novel findings more, better to just say it is TOP1-dependent HP1 recruitment?

Our response: We totally agree with this, and we have modified the title in the revised manuscript to reflect the cogent findings.

(2) In Fig 2, the authors introduce the concept of Mega Trans enhancers (elucidated in their previous work). In addition to the extended description in the materials and methods, a brief introduction should be provided in the main text, which lists the key features that define these enhancers.

Our response: We now has included this in line 119~121.

(3) For completeness, a WT rescue should be added to Fig. 3b.

Our response: We now provide these data in **Fig. 2C**.

(4) Is HP1 γ required for liquid-liquid phase separation at enhancers? Can this be tested?

Our response: We now show that knock-down of HP1 γ can impair the recruitment of the crucial component for the *MegaTrans* complex, namely GATA3- at the enhancers; GATA3 is the key protein mediating the liquid-like properties of the *MegaTrans* enhancers (Liu et al., *Cell*, 2015; Nair et al., *NSMB*, 2019); and HP1 γ is also important for the recruitment of Med26, important for eRNA elongation, and we therefore we can that HP1 γ is quite important in maintaining the liquid-like properties of the *MegaTrans* enhancers.

(5) In extended data 3, the ERE is one of the least abundant motifs. Can the authors comment on the others and their relevance? Also, the axes are presumably mislabelled and it should be 20% and not 0.2% of targets.

Our response: We now provide motif analysis on the new enhancer list, and find that not only ERE, but the motifs for FoxA1 and GRHL2 are enriched at TOP1cc enriched enhancers, and all of these three motifs are actually associated with E2 induced transcriptional activation as explained in line 168~170.

(6) For clarity, in Ext Fig. 4b, add the tracks of histone modifications that define enhancers if available as this alternative mark will help visualise the designation of the enhancer region.

Our response: We now have provided H3K27Ac to show the enhancer regions in our genome browser figures.

(7) In extended Ext Fig. 6c, the HP1 γ signal is the same in the non-shared regions, so what is recruiting it there?

Our response: HP1 γ is usually believed to be enriched at the heterochromatin regions, the phosphorylated state of HP1 γ could be tethered to the euchromatin by Ku70. We now show that TOP1cc could bring Ku70 to the active enhancers, and therefore contributes to the enrichment of HP1 γ at those HP1 γ /TOP1cc shared regions. For those non shared regions, we believe that Ku70 is one the major contributors for HP1 γ recruitment, but we can not exclude the role of other factors in this manuscript.

(8) In Ext fig 7e, the regions that are heterochromatic and overlap with HP1 γ as indicated in the text should be marked. Currently the overlaps with TOP1 are clear but not with the heterochromatin.

Our response: We now have marked the overlap regions between H3K9me3 enriched regions (heterochromatic regions) and HP1 γ enriched regions in the **Extended Data Fig. 7b**.

(9) Can the authors comment on the loss of HP1 γ signal on the TFF1 locus? Also they should show the tracks for TOP1 in Ext Fig. 8b.

Our response: We are pleased that the Referee mentioned that HP1 γ is reduced at the Tff1 gene locus, although HP1 γ is recruited the enhancers and (probably some promoters) for transcriptional activation, it is also associated with Pol II and Med26. Thus it is very likely associated with elongation complexes given that Med26 is one the key components for Pol II elongation, explaining why HP1 γ is not only reduced at enhancers and promoters, but also at gene bodies.

(10) More quantitative analysis of the data in Ext Fig. 10b&c should be shown to make the differences clearer to visualise (eg average tag density plots).

Our response: We thanks for the suggestions, and we now provide the quantitative analysis for these figures, and new data are presented in **Extended Data Fig. 10**.

(11) In general, the choice of adding things to the Supplementary is rather random. Some is clearly contributory and more of a control or deeper analysis. However other things are more central to the story. The authors should consider bringing more of the data into the main manuscript by creating addition figures as this would make the paper easier to follow.

Our response: We highly appreciate the Referee's comments, and we have significantly reorganized the figures in accord with the extensive additional data in response to the

review. We hope that we now present a more logically organized manuscript that is easier to follow.

Summary of Response to Reviewer 3: We thank the Reviewer for the helpful suggestions and have incorporated these points into the revised manuscript. We particularly thank the Referee for the critical suggestions that licensed an important new layer of insight, both generalizing and conceptually extending the discoveries in the manuscript.

Decision Letter, second revision:**Message:** 6th Jul 2022

Dear Dr. Rosenfeld,

Thank you again for submitting your manuscript entitled "Signal-induced Enhancer Activation Requires Ku70 to Read Topoisomerase1-DNA Covalent Complexes". I sincerely apologize for the delay in responding, which resulted from the difficulty in obtaining suitable referee reports and the fact that we are currently understaffed.

We now have comments from the 4 reviewers who evaluated your paper. In light of those reports (please see below), we remain interested in your study and would like to see your response to the comments of the referees, in the form of a revised manuscript.

Reviewer #1 is positive about the revision overall. Their points/questions are mainly about data presentation.

Reviewer #2 acknowledges the huge amount of work done and has only some minor comments; there are no experimental requests.

Reviewer #4 provides a short review, using words like "novel", "interesting" and "convincingly". The main criticism is that the role of Ku70 is not fully elucidated. The reviewer asks for new experiments to address this or simply acknowledging this limitation in the Discussion. The latter would be satisfactory.

Reviewer #5 has some technical concerns about the proteomic experiments. The reviewer highlights the lack of methodological information and notes that the ReCLIP experiments were not replicated. While we usually don't like to ask for new experiments at this stage, these serious technical concerns need to be fully addressed. Please do not hesitate to contact me directly if you would like to discuss these issues further.

Please be sure to address/respond to all concerns of the referees in full in a point-by-point response and highlight all changes in the revised manuscript text file. If you have comments that are intended for editors only, please include those in a separate cover letter.

We are committed to providing a fair and constructive peer-review process. Do not hesitate to contact me if there are specific requests from the reviewers that you believe are technically impossible or unlikely to yield a meaningful outcome.

We expect to see your revised manuscript within 3-6 months. If you cannot send it within this time, please contact us to discuss an extension; we would still consider your revision, provided that no similar work has been accepted for publication at NSMB or published elsewhere.

Reporting Summary:

Please note that all key data shown in the main figures as cropped gels or blots should be presented in uncropped form, with molecular weight markers. These data can be aggregated into a single supplementary figure item. While these data can be displayed in a relatively informal style, they must refer back to the relevant figures. These data should be submitted with the final revision, as source data, prior to acceptance, but you may want to start putting it together at this point.

Data availability: this journal strongly supports public availability of data. All data used in accepted papers should be available via a public data repository, or alternatively, as Supplementary Information. If data can only be shared on request, please explain why in your Data Availability Statement, and also in the correspondence with your editor. Please note that for some data types, deposition in a public repository is mandatory - more

information on our data deposition policies and available repositories can be found below:
<https://www.nature.com/nature-research/editorial-policies/reporting-standards#availability-of-data>

[Redacted]

Sincerely,

Tiago

Tiago Faial, PhD
Consulting Editor
Nature Structural & Molecular Biology

Reviewers' Comments:

Reviewer #1:

Remarks to the Author:

Overall, the revised manuscript has substantially improved, and I congratulate the authors for doing a thorough job at addressing the concerns raised by my original review. I have a few minor comments that should be addressed before accepting the manuscript for publication in NSMB, to improve clarity in the presentation.

- Fig 1A/B. How does the Top1cc signal compare to a random distribution across the genome of the same number of similar sized regions? For example, in Fig 1B the author could insert a different coloured bar of random distributed peaks within the bars.
- I cannot find info about how the ChIP-seq signal is represented. Are those Reads Per Million? Or maybe the ChIP-seq is spike-in normalized? In the methods and in the panel it says, "normalized tag densities". Please clarify.
- Sometimes the activation is indicated as $-/+ E2$, other times as $-/+ DHT$. For consistency, please always use the same term.
- Extended data 2b. It would be good to indicate what area to focus on in the genome browser shots. For example, mark the enhancers as it is done in Fig 1C and 3C with yellow boxes and in the legend indicate "Enhancers are highlighted with light-brown boxes".
- Extended data S2d-e. In panel d the Y axis is the same for TOP1cc but in panel e the Y axis are different for ERa. This is bit confusing. Please modify for consistency.
- Fig 3c is missing the labelling of Ku70.
- Line 207-208: the description of Fig 4g "In contrast, knock-down of Top1 had no or little effect on the enrichment of HP1 γ at the non-overlapped regions" does not correspond to the panel. There is no siTop1, but siHp1. The text indicates that the heatmap shows normalized Pol2 ChIP-seq signal, which is shown in Fig 4i. Please swap the panels.
- Extended data 7 c-e. These panels are hard to interpret without reading the legend/text. Maybe the author can add the information about heterochromatin or genome wide correlation on the actual plot.
- Extended data 8d: Why the colours and the region analysed (\pm 5Kb) are different as compared to Extended data 8b and c? For consistency better to use the same colours and genomic region.
- Extended data 9c. There is no indication that we are looking at Med26 upon siHp1/siNC. Would be helpful for the reader to state somewhere that siNC and siHP1 are related to MED26 ChIP-seq.
- Extended data 10c. Is there a reason not to include box plot for the heatmap with "other active enhancers"? Authors state that there is not difference of RNAPII enrichment after

knockdown in this case and it of course looks obvious from the heatmap, but why not show the quantification to be consistent? The other two panels show it.

- Line 254-56: "Specifically, factors such as MEF2a, MEF2b, MEF2c, MEF2d, which are known to be crucial for neuronal activity-regulated gene transcription⁴², were highly present at TOP1cc-enriched enhancers". Since this was based purely on the presence of binding motifs analysis, the authors might want to rephrase as follow "based on the presence of corresponding binding motifs, factors such as...might be present".

- Line 497. Protocol of the ReCLIP and Mass-Spectrometry. It is not clear when the streptavidin beads are added.

- Anti-TOP1cc (TG2017-2, TopoGEN). This antibody is not present in the Topogen website. Is there a typo with reporting the catalog number or has the antibody been discontinued?

Reviewer #2:

Remarks to the Author:

Reviewers comments: NSMB-A42739C

Title: Signal-induced Enhancer Activation Requires Ku70 to Read Topoisomerase1-DNA Covalent Complexes

-In their paper 'Signal-induced Enhancer Activation Requires Ku70 to Read Topoisomerase1-DNA Covalent Complexes', the authors demonstrate a link between DNA damage occurring at enhancers and signal dependent enhancer activation. Enhancers are hotspots for DNA damage, presumably because of high levels of eRNA transcription. The paper outlines a new mechanistic link between this DNA damage and enhancer activation, mediated by trapping of TOP1cc which is then read by DNA damage sensor Ku70, which nucleates recruitment of HP1 γ and Med26 to promote gene transcription..

The manuscript is a substantial revision of the original paper titled "TOP1cc-seq Reveals HP1 γ Is Required for Acute Ligand-dependent Activation of Enhancers" from January 2020. While I was broadly supportive of the original manuscript, there were a number of issues identified, especially around defining the biological mechanism that was crucial for the importance of the original paper. In particular, the mechanism linking recruitment of HP1 γ to DNA damage was poorly defined.

While I regret being the source of such a long review process, I appreciate the huge amount of extra work the authors have done to improve their story, especially during years disrupted by COVID-19. In my mind the new mechanism provided reinforces the story sufficiently to warrant publication in NSMB, providing the below points are addressed. I note that I don't expect further experimental work to address these points.

Main points

- While I understand the need for a powerful message, I find defining TOP1cc as an "epigenomic mark" to be confusing, and actually muddies the main point of the manuscript. The story is clear enough without such an oversimplification.

-In lines 174-180, the authors show that TOP1cc/Ku70 interaction require the nicking activity of TOP1 (demonstrated using the Y723F mutation) and DNA. The authors show

IP's to demonstrate this point (supplemental figures 6a,b). While the IP's are not the prettiest, I agree that they support this point. However, as this is important for the paper, these results should be shown in a main figure panel. The authors should also clarify in line 176 the Y273F mutant is used to demonstrate the requirement for nicking activity.

-The legend for Figure 4 has a number of errors:

- Figure 4G does not show box and whisker plots - the legend describes figure 4i.
- The legend for figure 4 i describes box and whisker plots for an unlabelled panel.
- Line 790 Figure 4 legend - Typo (CUT&RUN)

-The authors provide a link between HP1 γ recruitment and recruitment of Med26, demonstrating in extended figure 9 that siHP1 γ decreased Med26 ChIP signals over megatrans enhancers in response to E2 treatment. It is currently hard to judge this claim based on the heatmap alone. Could the authors include this information as a box and whiskers plot to highlight this change more clearly as done in extended figure 10 b,d?
Minor points

- Figure 1E demonstrates loss of binding TOP1cc binding with the catalytic Y723F mutant. Can the authors also provide a box plot comparing signal enrichment over these peaks in each condition (perhaps as a supplementary figure). Legend needs to state what the colour scale represents for all heatmaps, e.g 2B, 2D

-Line 128 Typo in section heading 'acutely'

-Line 192 Typo 'acutely'

-Extended Data 8, GATA3 panel d. The authors should change the legend and color scheme to match panels b and c. The altered labelling is confusing and the red/green colouring is poor for accessibility.

Line 242 - Typo 'underlying to acute'

Reviewer #4:

Remarks to the Author:

In this manuscript the authors report an interesting mechanism of enhancer activation through Top1 ccs through the recruitment of Ku70 which attracts HP1 γ and Med 26 complex to mediate transcriptional activation.

The authors provide a series of experiments which convincingly show that Topccs are generated at enhancers and they are required for acute enhancer activation. Then they use a proteomic approach to identify the factors that are interacting with Top1 after E2. This approach reveals among other proteins, factors from the NHEJ pathway and HP1 γ . Promoted by this, the authors investigate the genome wide recruitment of Ku70 and Ku80 as well DNAPKcs +/- E2. They find that that Ku70 enrichments is enhanced in E2 dependent promoters in the presence of E2 and Ku70 depletion affects the acute activation of certain downstream genes. HP1 γ shows a similar pattern of recruitment and behaviour.

The role of Ku70 in acute activation of transcriptional programs through Topccs at

enhancers is novel and interesting. In cells Ku70 exists in heterodimer with Ku80 and their protein stability is interdependent. Depletion of Ku80 reduces Ku70 and vice versa. In this study, although Ku80 is interacting to Top1 in the presence of E2, Ku80's recruitment at Topccs at enhancers is not increased. Moreover, although DNAPKcs plays a role in the acute regulation of some E2 dependent genes its recruitment in most of the tracks shown at the manuscript is not increased as the proteins binds to chromatin even in the absence of E2. Therefore, the unique role of Ku70 in the process has not been nailed. Further experiments including mutants which affect the dimerisation of Ku80/Ku70 are needed or the authors need to tone down this statement and include further discussion.

Reviewer #5:

Remarks to the Author:

The authors used ReCLIP and mass spectrometry to identify proteins that interact with DNA-bound TOP1, yet some details are missing how the experiment was performed, and how data were analyzed. Specifically, beads were used for IP, however the antibody it was coated with (likely TOP1cc) as well as incubation conditions were not mentioned. Next, it is not clear why streptavidin was used to pull down protein complexes. Minimal information should be provided on the instruments and experimental conditions for LCMS (type of instruments, gradient length). Similarly, it should be stated how data were analyzed (search algorithm plus key settings, FDR, etc). According to information provided in suppl tables 1-3, multiple (>10 as far as I could see) post-translational modifications were allowed. This is risky since it inflates the search space and significantly increases the chance of false identifications. More seriously, modified peptides should be disregarded for protein quantification, especially when this is performed by a crude method such as peptide counting, as was done here. Hence, data should be revised to only include non-modified peptides. Finally, ReCLIP experiments were performed without replicates, limiting the robustness of the data.

Author Rebuttal, second revision:

Overview:

We first submitted this manuscript on Jan 3rd 2020, and greatly appreciated the cogent comments of the Referees, which, however, necessitated more than 2 years of additional experimentation, leading to a conceptual advance in establishing the unexpected role of Ku70 in the Topoisomerase1/HP1g/Med26-mediated activation of estrogen-activated *MegaTrans* enhancers in mammalian cells. It was particularly gratifying that the expanded findings in the resubmission were greatly appreciated by the two most critical Referees from the initial submission, stating that we had “convincingly” shown the role of Top1cc at acutely activated enhancers.

Two additional Referees appear to have been consulted for the resubmission. Referee #4 appreciated the discovery, and we agree with the Referee's and your suggestion to add a caveat regarding Ku80 in the Discussion. Our data revealed that Ku70 was induced at the enhancers upon the acute signals, while Ku80 was not similarly induced. For the DNA-PKcs, we presented 3 examples showing a clear induction upon E₂ treatment. Based on the previous report of the interaction between HP1g and Ku70 (Lomberk, et al., Nature Cell Biology, 2006),

once Ku70 formed a complex with HP1g, Ku80 was not present, as we again point out in our final revised manuscript. However, we agree that a fully unique role of Ku70 in these events with no absolute role for Ku80 could be not unambiguously proven, especially without evidence from Ku70/Ku80 dimerization mutants. Therefore, as recommended by the Referee, we will now state that: “While it is now clear that Ku70 interaction with Topo1 and its function are required for the acutely activated enhancers, a more subtle quantitative role for Ku80 cannot be excluded”.

The new Referee #5, clearly an expert in mass spectrometry, requested a clarification that posttranslational modifications be disallowed in the table reporting the pull-down data. The Referee is entirely correct, and we appreciate the opportunity to accordingly modify the Table; however, the conclusions remain unaltered. The Referee cautioned that we might want to include a replicate of the ReCLIP experiment leading to the identification of Ku70 as further suggestive evidence; however, we must have made it insufficiently clear that we had instead performed an orthogonal and independent experiment to confirm this putative interaction. As we showed in Figs 3b, 3e, 3f and 4c, the pull-down assays confirmed the interactions between Top1, Ku70, and HP1g, which along with the knockdown data results in Figure 4f, provide the critical evidence of the importance of the interaction for acute ligand-dependent activation of the *MegaTrans* enhancers. Thus, we went much further than mere mass spectrometry to provide evidence for interactions between Top1, Ku70 and HP1g.

We appreciate the rigorous and insightful reviews and suggestions provided by the Referees. We incorporate all the suggestions by the Referees in this final revision. Thus, there are no remaining scientific concerns regarding this manuscript. We are quite pleased to provide a fundamental new insight into regulated enhancer activation, but there is an additional issue. We have just provided a positive review for a manuscript at *Nature*, which apparently will be accepted, that provides a partial statement on just one aspect of our manuscript. In fact, our manuscript presents a much broader, mechanistic and comprehensive story, and given that we want this to be fully “new news”, as we have been under review now for almost 3 years at *NSMB*, and as there are no significant scientific concerns, we strongly request that the manuscript be accepted without further re-review, as it is important to have this discovery be announced at *NSMB* in a timely fashion. Thank you for your consideration.

RESPONSE TO REFEREES

Reviewers' Comments:

Reviewer #1:

Remarks to the Author:

Overall, the revised manuscript has substantially improved, and I congratulate the authors for doing a thorough job at addressing the concerns raised by my original review. I have a few minor comments that should be addressed before accepting the manuscript for publication in *NSMB*, to improve clarity in the presentation.

Re: We thank the Referee's both for the cogent initial review and for the comment on we had "substantially" improved our manuscript and for the recommendation for publication in *NSMB*. We appreciate the suggestions for the final manuscript and have incorporated all into the final figures and text.

- Fig 1A/B. How does the Top1cc signal compare to a random distribution across the genome of the same number of similar sized regions? For example, in Fig 1B the author could insert a different coloured bar of random distributed peaks within the bars.

Re: We thank the Referee for this suggestion. Here we had selected 179,220 random regions from hg38, finding that only 348 TOP1cc enriched regions could be overlapped with these random regions, further confirming that our TOP1cc signals are selected enriched at the *cis*regulatory regions. These data are shown in the revised Fig.1b.

- I cannot find info about how the ChIP-seq signal is represented. Are those Reads Per Million? Or maybe the ChIP-seq is spike-in normalized? In the methods and in the panel it says, "normalized tag densities". Please clarify.

Re: We created tag directories for each individual sample, allowing no more than two tags per base pair and the combined replicates each treatment, and then normalized each directory by the total number of mapped tags such that each directory contains 10 million tags. We now clarify this in Page 21 (Line 702-703).

- Sometimes the activation is indicated as -/+ E2, other times as -/+ DHT. For consistency, please always use the same term.

Re: We regret any confusion, for human breast cancer MCF7 cells, we treated them with 17- β Estradiol (E₂; Steraloids, Inc.), for human prostate cancer LNCAP cells, we treated with 5 α dihydrotestosterone (DHT, Sigma). We state this in Page 13 (Line 415-419).

- Extended data 2b. It would be good to indicate what area to focus on in the genome browser shots. For example, mark the enhancers as it is done in Fig 1C and 3C with yellow boxes and in the legend indicate "Enhancers are highlighted with light-brown boxes".

Re: We thank the Referee for this suggestion; the TOP1cc and BRD4 co-enriched regions have now been highlighted with light-brown boxes, as suggested.

- Extended data S2d-e. In panel d the Y axis is the same for TOP1cc but in panel e the Y axis are different for ERa. This is bit confusing. Please modify for consistency.

Re: We thank the referee for the suggestion; the two Y-axis in Panel e are now the same for ERa.

- Fig 3c is missing the labelling of Ku70.

Re: Sorry for the oversight; we have now changed “Ku” to “Ku70”.

- Line 207-208: the description of Fig 4g “In contrast, knock-down of Top1 had no or little effect on the enrichment of HP1 γ at the non-overlapped regions” does not correspond to the panel. There is no siTop1, but siHp1. The text indicates that the heatmap shows normalized Pol2 ChIPseq signal, which is shown in Fig 4i. Please swap the panels.

Re: We appreciate this suggestion, noting the mislabeling of Fig.4g and Fig.4i; we have corrected this by swapping these two panels.

- Extended data 7 c-e. These panels are hard to interpret without reading the legend/text. Maybe the author can add the information about heterochromatin or genome wide correlation on the actual plot.

Re: As suggested, we have now marked Extended data 7c as heterochromatin and marked Extended data 7d & e as genome wide.

- Extended data 8d: Why the colours and the region analysed (+/- 5Kb) are different as compared to Extended data 8b and c? For consistency better to use the same colours and genomic region.

Re: We now switched the color and made these 3 panels consistent.

- Extended data 9c. There is no indication that we are looking at Med26 upon siHp1/siNC. Would be helpful for the reader to state somewhere that siNC and siHP1 are related to MED26 CHIP-seq.

Re: Med26 has been marked in these Figs.

- Extended data 10c. Is there a reason not to include box plot for the heatmap with “other active enhancers”? Authors state that there is not difference of RNAPII enrichment after knockdown in this case and it of course looks obvious from the heatmap, but why not show the quantification to be consistent? The other two panels show it.

Re: We did not show this box plot because the Reviewers in the first round did not ask for it. We now have inserted the boxplot in the Extended data 10c as requested.

- Line 254-56: “Specifically, factors such as MEF2a, MEF2b, MEF2c, MEF2d, which are known to be crucial for neuronal activity-regulated gene transcription⁴², were highly present at TOP1ccenriched enhancers”. Since this was based purely on the presence of binding motifs analysis, the authors might want to rephrase as follow “based on the presence of corresponding binding motifs, factors such as...might be present”.

Re: We thank the Referee for the suggestions; we now have accordingly modified our statement in the revised manuscript in Page 9 (Line 279-282).

- Line 497. Protocol of the ReCLIP and Mass-Spectrometry. Tt is not clear when the streptavidin beads are added.

Re: The ReCLIP and Mass-Spectrometry protocol has been updated in Page 17~18.

- Anti-TOP1cc (TG2017-2, TopoGEN). This antibody is not present in the Topogen website. Is there a typo with reporting the catalog number or has the antibody been discontinued?

Re: Unfortunately, this antibody (TOP1cc (TG2017-2, TopoGEN, Lot# 17AG15)) has been discontinued. Luckily, the following antibody has the similar efficienc- and we provide this information.

Top1cc clone 1.1A	Millipore	MABE1084 (Lot#3176723)
-------------------	-----------	---------------------------

Overview: We thank the Referee both for the important questions raised in the initial review, which greatly improved the insights provided in the final manuscript, and for the careful reading and suggestions for the final manuscript .

Reviewer #2:

Remarks to the Author:

Reviewers comments: NSMB-A42739C

Title: Signal-induced Enhancer Activation Requires Ku70 to Read Topoisomerase1-DNA Covalent Complexes

-In their paper 'Signal-induced Enhancer Activation Requires Ku70 to Read Topoisomerase1DNA Covalent Complexes', the authors demonstrate a link between DNA damage occurring at enhancers and signal dependent enhancer activation. Enhancers are hotspots for DNA damage, presumably because of high levels of eRNA transcription. The paper outlines a new mechanistic link between this DNA damage and enhancer activation, mediated by trapping of TOP1cc which is then read by DNA damage sensor Ku70, which nucleates recruitment of HP1 γ and Med26 to promote gene transcription..

The manuscript is a substantial revision of the original paper titled "TOP1cc-seq Reveals HP1 γ Is Required for Acute Ligand-dependent Activation of Enhancers" from January 2020. While I was broadly supportive of the original manuscript, there were a number of issues identified, especially around defining the biological mechanism that was crucial for the importance of the original paper. In particular, the mechanism linking recruitment of HP1 γ to DNA damage was poorly defined.

While I regret being the source of such a long review process, I appreciate the huge amount of extra work the authors have done to improve their story, especially during years disrupted by COVID-19. In my mind the new mechanism provided reinforces the story sufficiently to warrant publication in NSMB, providing the below points are addressed. I note that I don't expect further experimental work to address these points.

Re: We highly appreciate the rigorous and insightful reviews and suggestions provided by this Referee. It required 2 and a half years for the experimentation licensing the revision, of course impaired by COVID-19. However, the rigorous and insightful reviews and suggestions by this Referee really greatly improved the quality and ultimate impact of our manuscript.

Main points

- While I understand the need for a powerful message, I find defining TOP1cc as an “epigenomic mark” to be confusing, and actually muddies the main point of the manuscript. The story is clear enough without such an oversimplification.

Re: We agree that the “epigenomic mark” might be a little bit confusing. We now have changed it to epigenomic signature.

-In lines 174-180, the authors show that TOP1cc/Ku70 interaction require the nicking activity of TOP1 (demonstrated using the Y723F mutation) and DNA. The authors show IP's to demonstrate this point (supplemental figures 6a,b). While the IP's are not the prettiest, I agree that they support this point. However, as this is important for the paper, these results should be shown in a main figure panel. The authors should also clarify in line 176 the Y273F mutant is used to demonstrate the requirement for nicking activity.

Re: We appreciate the positive comments on the data showing TOP1cc/Ku70 interaction require the nicking activity of TOP1 (demonstrated using the Y723F mutation) and DNA. We agree that these data are very important, and had put them in the extended data due to the limitation of the figures. Now we have already moved these data to the main figures, as the Referee suggested. And we also have clarified that the Y723F could abolish the Top1-dependent DNA nicking activities in the revised manuscript in Page 6 (Line185-186).

-The legend for Figure 4 has a number of errors:

-Figure 4G does not show box and whisker plots - the legend describes figure 4i.

-The legend for figure 4 i describes box and whisker plots for an unlabelled panel. -Line 790 Figure 4 legend - Typo (CUT&RUN)

Re: We thank the reviewer for pointing out these errors. We have swapped Fig.4g and Fig.4i; added the figure legend to describe the unlabeled panel; and also corrected the typos in the legend.

-The authors provide a link between HP1 γ recruitment and recruitment of Med26, demonstrating in extended figure 9 that siHP1 γ decreased Med26 ChIP signals over megatrans enhancers in response to E2 treatment. It is currently hard to judge this claim based on the heatmap alone. Could the authors include this information as a box and whiskers plot to highlight this change more clearly as done in extended figure 10 b,d?

Re: A boxplot showing the normalized tags of Med26 has been inserted in Extended Fig.10b, as requested.

Minor points

- Figure 1E demonstrates loss of binding TOP1cc binding with the catalytic Y723F mutant. Can the authors also provide a box plot comparing signal enrichment over these peaks in each condition (perhaps as a supplementary figure). Legend needs to state what the colour scale represents for all heatmaps, e.g 2B, 2D

Re: We thank the Referee for the suggestions. A boxplot has been added in the Fig.1e as suggested by the reviewer, and the color scale shows the normalized tag numbers and we now have so stated in the figure legends.

-Line 128 Typo in section heading 'acutely'

-Line 192 Typo 'acutely'

Re: "Acutely" has been replaced with "acute" in these two cases.

-Extended Data 8, GATA3 panel d. The authors should change the legend and color scheme to match panels b and c. The altered labelling is confusing and the red/green colouring is poor for accessibility.

Re: In response to the suggestions, we now have switched the color to make all three Figs consistent.

Line 242 - Typo 'underlying to acute'

Re: We changed "underlying to acute" to "underlying acute".

Overview: We appreciate the Referee's insightful suggestions in the first review, and the suggestions for figure modifications for the final manuscript, while requesting no further experiments.

Reviewer #4:

Remarks to the Author:

In this manuscript the authors report an interesting mechanism of enhancer activation through Top1 ccs through the recruitment of Ku70 which attracts HP1gamma and Med 26 complex to mediate transcriptional activation.

The authors provide a series of experiments which convincingly show that Topccs are generated at enhancers and they are required for acute enhancer activation.

Then they use a proteomic approach to identify the factors that are interacting with Top1 after E2. This approach reveals among other proteins, factors from the NHEJ pathway and HP1g. Promoted by this, the authors investigate the genome wide recruitment of Ku70 and Ku80 as well DNAPKcs +/- E2. They find that that Ku70 enrichments is enhanced in E2 dependent promoters in the presence of E2 and Ku70 depletion affects the acute activation of certain downstream genes. HP1g shows a similar pattern of recruitment and behaviour.

The role of Ku70 in acute activation of transcriptional programs through Topccs at enhancers is novel and interesting. In cells Ku70 exists in heterodimer with Ku80 and their protein stability is interdependent. Depletion of Ku80 reduces Ku70 and vice versa. In this study, although Ku80 is interacting to Top1 in the presence of E2, Ku80's recruitment at Topccs at enhancers is not increased. Moreover, although DNAPKcs plays a role in the acute regulation of some E2 dependent genes its recruitment in most of the tracks shown at the manuscript is not increased as the proteins binds to chromatin even in the absence of E2. Therefore, the unique role of Ku70 in the process has not been nailed. Further experiments including mutants which affect the dimerisation of Ku80/Ku70 are needed or the authors need to tone down this statement and include further discussion.

Re: We thank the Referee for the cogent comments regarding the manuscript, and appreciate the statement that we had provided a series of experiments which “convincingly” showing that

the role of TOP1cc at enhancers. Our data revealed that Ku70 was induced at the enhancers upon the acute signals, while Ku80 was not similarly induced. For the DNA-PKcs, we presented 3 examples showing a clear induction upon E₂ treatment. Actually, based on the previous report of the interaction between HP1g and Ku70 (Lomber, et al., Nature Cell Biology, 2006), once Ku70 formed a complex with HP1g, Ku80 was not present, and we now have cited this study in our final revised manuscript in Page 11 (Line 348-349). However, we agree that a fully unique role of Ku70 in these events with no absolute role for Ku80 that could be unambiguously proven, especially without evidence from Ku70/Ku80 dimerization mutants. Therefore, we now state that: “While it is now clear that Ku70 interaction with Topo1 and its function are required

for the acutely activated enhancers, a more subtle quantitative role for Ku80 cannot be excluded” in Page 11 (Line 353-355).

Reviewer #5:

Remarks to the Author:

The authors used ReCLIP and mass spectrometry to identify proteins that interact with DNAbound TOP1, yet some details are missing how the experiment was performed, and how data were analyzed. Specifically, beads were used for IP, however the antibody it was coated with (likely TOP1cc) as well as incubation conditions were not mentioned. Next, it is not clear why streptavidin was used to pull down protein complexes. Minimal information should be provided on the instruments and experimental conditions for LCMS (type of instruments, gradient length). Similarly, it should be stated how data were analyzed (search algorithm plus key settings, FDR, etc). According to information provided in suppl tables 1-3, multiple (>10 as far as I could see) post-translational modifications were allowed. This is risky since it inflates the search space and significantly increases the chance of false identifications. More seriously, modified peptides should be disregarded for protein quantification, especially when this is performed by a crude method such as peptide counting, as was done here. Hence, data should be revised to only include non-modified peptides. Finally, ReCLIP experiments were performed without replicates, limiting the robustness of the data.

Re: We highly appreciate this Referee’s comments on the ReCLIP and mass spectrometry especially on the post-translational modified peptides. The Referee is entirely correct, and we appreciate the opportunity to accordingly modify the Table; importantly, however, the conclusions remain unaltered. Streptavidin beads were commonly employed for the IP because we could use very rigorous washing conditions during the IP experiments. Further, we have updated experimental protocol, and the all the detailed experimental information requested by the Referee “instruments and experimental conditions for LCMS (type of instruments, gradient length). And how the data were analyzed (search algorithm plus key settings, FDR)” have been inserted in Page 17~18. We have expanded the Methods section describing the ReCLIP experiments. Furthermore, the Referee cautioned that we might want to include a replicate of the ReCLIP experiment leading to the identification of Ku70 as further suggestive evidence; however, we must have made it insufficiently clear that we had instead performed an orthogonal and independent experiment to confirm this putative interaction. As we showed in Figs 3b, 3e, 3f and 4c, the pull-down assays confirmed the interactions between Top1, Ku70, and HP1g, which along with the knockdown data results in Fig 4f, providing the additional critical evidence of the importance of the interaction for acute ligand-dependent acutely

enhancer activation. Thus, we went much further than initial mass spectrometry to provide evidence for interactions between Top1, Ku70 and HP1g.

Decision Letter: Third Revision

Message: Our ref: NSMB-A42739D

12th Aug 2022

Dear Dr. Rosenfeld,

Thank you for submitting your revised manuscript entitled "Signal-induced Enhancer Activation Requires Ku70 to Read Topoisomerase1-DNA Covalent Complexes" (NSMB-A42739D). It has now been seen by reviewer #5 and their comments are below. The reviewer finds that the paper has improved in revision, and therefore we'll be happy in principle to publish it in Nature Structural & Molecular Biology, pending minor revisions to comply with our editorial and formatting guidelines.

We are now performing detailed checks on your paper and will send you a checklist detailing our editorial and formatting requirements in about 2 weeks. Please do not upload the final materials and make any revisions until you receive this additional information from us.

To facilitate our work at this stage, we would appreciate if you could send us the main text as a Word file. Please make sure to copy the NSMB account (cc'ed above).

Thank you again for your interest in Nature Structural & Molecular Biology. Please do not hesitate to contact me if you have any questions.

Congratulations!

Sincerely,

Tiago

Tiago Faial, PhD
Consulting Editor
Nature Structural & Molecular Biology

Reviewer #5 (Remarks to the Author):

The authors have sufficiently addressed my concerns and I recommend publication of this work.

Decision Letter: Author Guidance

Message: Our ref: NSMB-A42739D

28th Sep 2022

Dear Dr. Rosenfeld,

Thank you for your patience as we've prepared the guidelines for final submission of your Nature Structural & Molecular Biology manuscript, "Signal-induced Enhancer Activation Requires Ku70 to Read Topoisomerase1-DNA Covalent Complexes" (NSMB-A42739D). Please carefully follow the step-by-step instructions provided in the attached file, and add a response in each row of the table to indicate the changes that you have made. Please also check and comment on any additional marked-up edits we have proposed within the text. Ensuring that each point is addressed will help to ensure that your revised manuscript can be swiftly handed over to our production team.

In recognition of the time and expertise our reviewers provide to Nature Structural & Molecular Biology's editorial process, we would like to formally acknowledge their contribution to the external peer review of your manuscript entitled "Signal-induced Enhancer Activation Requires Ku70 to Read Topoisomerase1-DNA Covalent Complexes". For those reviewers who give their assent, we will be publishing their names alongside the published article.

Nature Structural & Molecular Biology offers a Transparent Peer Review option for new original research manuscripts submitted after December 1st, 2019. As part of this initiative, we encourage our authors to support increased transparency into the peer review process by agreeing to have the reviewer comments, author rebuttal letters, and editorial decision letters published as a Supplementary item. When you submit your final files please clearly state in your cover letter whether or not you would like to participate in this initiative. Please note that failure to state your preference will result in delays in accepting your manuscript for publication.

Cover suggestions

As you prepare your final files we encourage you to consider whether you have any images or illustrations that may be appropriate for use on the cover of Nature Structural & Molecular Biology.

Nature Structural & Molecular Biology has now transitioned to a unified Rights Collection system which will allow our Author Services team to quickly and easily collect the rights and permissions required to publish your work. Approximately 10 days after your paper is formally accepted, you will receive an email in providing you with a link to complete the grant of rights. If your paper is eligible for Open Access, our Author Services team will also be in touch regarding any additional information that may be required to arrange payment for your article.

Please note that *Nature Structural & Molecular Biology* is a Transformative Journal (TJ). Authors may publish their research with us through the traditional subscription access route or make their paper immediately open access through payment of an article-processing charge (APC). Authors will not be required to make a final decision about access to their article until it has been accepted. [Find out more about Transformative Journals](https://www.springernature.com/gp/open-research/transformative-journals)

Authors may need to take specific actions to achieve [compliance with funder and institutional open access mandates](https://www.springernature.com/gp/open-research/funding/policy-compliance-faqs). If your research is supported by a funder that requires immediate open access (e.g. according to [Plan S principles](https://www.springernature.com/gp/open-research/plan-s-compliance)) then you should select the gold OA route, and we will direct you to the compliant route where possible. For authors selecting the subscription publication route, the journal's standard licensing terms will need to be accepted, including [self-archiving policies](https://www.nature.com/nature-portfolio/editorial-policies/self-archiving-and-license-to-publish). Those licensing terms will supersede any other terms that the author or any third party may assert apply to any version of the manuscript.

For information regarding our different publishing models please see our [page](https://www.springernature.com/gp/open-research/transformational-journals). If you have any questions about costs, Open Access requirements, or our legal forms, please contact ASJournals@springernature.com.

Please use the following link for uploading these materials:
[Redacted]

Best regards,

Sophia Frank
Editorial Assistant
Nature Structural & Molecular Biology
nsmb@us.nature.com

On behalf of

Florian Ullrich, Ph.D.
Associate Editor
Nature Structural & Molecular Biology
ORCID 0000-0002-1153-2040

Reviewer #5:
Remarks to the Author:
The authors have sufficiently addressed my concerns and I recommend publication of this work.

Author Rebuttal, Third revision:

Reviewer #5:
Remarks to the Author:
The authors have sufficiently addressed my concerns and I recommend publication of this work.

Response: we really appreciate the Referee for the critical suggestions that licensed an important new layer of insight in this manuscript.

Decision Letter: Final decision

Message 27th Oct 2022

:

Dear Dr. Rosenfeld,

We are now happy to accept your revised paper "Signal-induced Enhancer Activation Requires Ku70 to Read Topoisomerase1-DNA Covalent Complexes" for publication as a Article in Nature Structural & Molecular Biology.

As soon as your article is published, you can generate your shareable link by entering the DOI of your article here: <http://authors.springernature.com/share> http://authors.springernature.com/share. Corresponding authors will also receive an automated email with the shareable link

Your paper will be published online soon after we receive proof corrections and will appear in print in the next available issue. You can find out your date of online publication by contacting the production team shortly after sending your proof corrections. Content is published online weekly on Mondays and Thursdays, and the embargo is set at 16:00 London time (GMT)/11:00 am US Eastern time (EST) on the day of publication. Now is the time to inform your Public Relations or Press Office about your paper, as they might be

interested in promoting its publication. This will allow them time to prepare an accurate and satisfactory press release. Include your manuscript tracking number (NSMB-A42739E) and our journal name, which they will need when they contact our press office.

About one week before your paper is published online, we shall be distributing a press release to news organizations worldwide, which may very well include details of your work. We are happy for your institution or funding agency to prepare its own press release, but it must mention the embargo date and Nature Structural & Molecular Biology. If you or your Press Office have any enquiries in the meantime, please contact press@nature.com.

Please note that *Nature Structural & Molecular Biology* is a Transformative Journal (TJ). Authors may publish their research with us through the traditional subscription access route or make their paper immediately open access through payment of an article-processing charge (APC). Authors will not be required to make a final decision about access to their article until it has been accepted. <https://www.springernature.com/gp/open-research/transformative-journals> Find out more about Transformative Journals

Kind regards,
Florian

Dr Florian Ullrich
Associate Editor, Nature
Consulting Editor, Nature Structural & Molecular Biology
ORCID 0000-0002-1153-2040